# EF21: A New, Simpler, Theoretically Better, and Practically Faster Error Feedback

**Peter Richtárik**
KAUST*

Igor Sokolov
KAUST

Ilyas Fatkhullin
KAUST† & TU Munich

## Abstract

Error feedback (EF), also known as error compensation, is an immensely popular convergence stabilization mechanism in the context of distributed training of supervised machine learning models enhanced by the use of contractive communication compression mechanisms, such as Top-$k$. First proposed by Seide et al. [2014] as a heuristic, EF resisted any theoretical understanding until recently [Stich et al., 2018, Alistarh et al., 2018]. While these early breakthroughs were followed by a steady stream of works offering various improvements and generalizations, the current theoretical understanding of EF is still very limited. Indeed, to the best of our knowledge, all existing analyses either i) apply to the single node setting only, ii) rely on very strong and often unreasonable assumptions, such as global boundedness of the gradients, or iterate-dependent assumptions that cannot be checked a-priori and may not hold in practice, or iii) circumvent these issues via the introduction of additional unbiased compressors, which increase the communication cost. In this work we fix all these deficiencies by proposing and analyzing a new EF mechanism, which we call EF21, which consistently and substantially outperforms EF in practice. Moreover, our theoretical analysis relies on standard assumptions only, works in the distributed heterogeneous data setting, and leads to better and more meaningful rates. In particular, we prove that EF21 enjoys a fast $\mathcal{O}(1/T)$ convergence rate for smooth nonconvex problems, beating the previous bound of $\mathcal{O}(1/T^{2/3})$, which was shown under a strong bounded gradients assumption. We further improve this to a fast linear rate for Polyak-Lojasiewicz functions, which is the first linear convergence result for an error feedback method not relying on unbiased compressors. Since EF has a large number of applications where it reigns supreme, we believe that our 2021 variant, EF21, can have a large impact on the practice of communication efficient distributed learning.

## 1   Introduction

In order to obtain state-of-the-art performance, modern machine learning models rely on elaborate architectures, need to be trained on data sets of enormous sizes, and involve a very large number of parameters. Some of the most successful models are heavily over-parameterized, which means that they involve more parameters than the number of available training data points [Arora et al., 2018]. Naturally, these circumstances should inform the design of optimization methods that could be most efficient to perform the training.

First, the reliance on sophisticated model architectures, as opposed to simple linear models, generally leads to **nonconvex** optimization problems, which are more challenging than convex problems [Jain and Kar, 2017]. Second, the need for very large training data sizes necessitates the use of **distributed**

---

*King Abdullah University of Science and Technology, Thuwal, Saudi Arabia.
†This paper was written while Ilyas Fatkhullin was an intern at KAUST.

35th Conference on Neural Information Processing Systems (NeurIPS 2021).

**computing** [Verbraeken et al., 2019]. Due to its enormous size, the data needs to be partitioned across a number of machines able to work in parallel. Typically, for further efficiency gains, each such machine further parallelizes its local computations using one or more hardware accelerators. Third, the very large number of parameters describing these models exerts an extra stress on the communication links used to exchange model updates among the machines. These links are typically slow compared to the speed at which computation takes place, and communication often forms the bottleneck of distributed systems even in less extreme situations than over-parameterized training where the number of parameters, and hence the nominal size of communicated messages, can be truly staggering. For this reason, modern efficient optimization methods typically employ elaborate **lossy communication compression** techniques to reduce the size of the communicated messages.

Due to the above reasons, in this paper we are interested in solving the *nonconvex distributed optimization problem*

$$\min_{x \in \mathbb{R}^d} \left[ f(x) \stackrel{\text{def}}{=} \frac{1}{n} \sum_{i=1}^{n} f_i(x) \right], \tag{1}$$

where $x \in \mathbb{R}^d$ represents the parameters of a machine learning model we wish to train, $n$ is the number of workers/nodes/machines, and $f_i(x)$ is the loss of model $x$ on the data stored on node $i$. We specifically focus on the development of new and more efficient communication efficient first-order methods for solving (1) utilizing *biased* compression operators, with a special emphasis on *clean convergence analysis* which removes the *strong and often unrealistic assumptions*, such as the bounded gradient assumption, which are currently needed to analyze such methods (see Table 1).

The remainder of the paper is organized as follows. In Section 2 we describe the key concepts, results and open problems that form the motivation for our work, and summarize our main contributions. Our main theoretical results are presented in Section 4. In Section 4.4 we establish a connection between EF and EF21. Finally, experimental results are described in Section 5.

## 2 Background and Motivation

To better motivate our approach and contributions, we first offer a concise walk-through over the key considerations, difficulties, advances and open problems in this area.

### 2.1 Two families of compression operators

Compression is typically performed via the application of a (possibly randomized) mapping $\mathcal{C} : \mathbb{R}^d \to \mathbb{R}^d$, where $d$ is the dimension of the vector/tensor that needs to be communicated, with the property that it is much easier/quicker to transfer $\mathcal{C}(x)$ than it is to transfer the original message $x$. This can be achieved in several ways, for instance by sparsifying the input vector [Alistarh et al., 2018], or by quantizing its entries [Alistarh et al., 2017, Horváth et al., 2019a], or via a combination of these and other approaches [Horváth et al., 2019a, Beznosikov et al., 2020].

There are two large classes of compression operators $\mathcal{C}$ often studied in the literature: i) **unbiased compression operators** satisfying a variance bound proportional to the square norm of the input vector, and ii) **biased compression operators** whose square distortion is contractive with respect to the square norm of the input vector.

In particular, we say that a (possibly randomized) map $\mathcal{C} : \mathbb{R}^d \to \mathbb{R}^d$ is an *unbiased compression operator*, or simply just *unbiased compressor*, if there exists a constant $\omega \geq 0$ such that

$$\mathbb{E}\left[\mathcal{C}(x)\right] = x, \quad \mathbb{E}\left[\|\mathcal{C}(x) - x\|^2\right] \leq \omega \|x\|^2, \qquad \forall x \in \mathbb{R}^d. \tag{2}$$

The family of such operators will be denoted by $\mathbb{U}(\omega)$. Further, we say that a (possibly randomized) map $\mathcal{C} : \mathbb{R}^d \to \mathbb{R}^d$ is a *biased compression operator*, or simply just *biased compressor*, if there exists a constant $0 < \alpha \leq 1$ such that

$$\mathbb{E}\left[\|\mathcal{C}(x) - x\|^2\right] \leq (1 - \alpha) \|x\|^2, \qquad \forall x \in \mathbb{R}^d. \tag{3}$$

The family of such operators will be denoted by $\mathbb{B}(\alpha)$. It is well known that, in a certain sense, the latter class contains the former. In particular, it is easy to verify that if $\mathcal{C} \in \mathbb{U}(\omega)$, then $(1 + \omega)^{-1}\mathcal{C} \in \mathbb{B}(1/(1+\omega))$. However, the latter class is strictly larger, i.e., it contains compressors which do not arise via a scaling of an unbiased compressor. A canonical example of this is the Top-$k$ compressor, which

preserves the $k$ largest (in absolute value) entries of the input, and zeros out the remaining entries, and for which $\alpha = {}^k/d$. We refer to [Beznosikov et al., 2020, Safaryan et al., 2021, Table 1] for more examples of unbiased and biased compressors, and to Xu et al. [2020] for a systems-oriented survey.

When used in an appropriate way, greedy biased compressors, such as Top-$k$, are often empirically superior to their unbiased counterparts [Seide et al., 2014], such as Rand-$k$. Intuitively, such greedy compressors retain more of the "information" or "energy" contained within the message, and hence introduce less distortion. This is beneficial in practice, at least in the simplistic single node (i.e., non-distributed) setting, albeit even here we do not have convincing theory that would explain this. Indeed, both Top-$k$ and Rand-$k$ impart the same distortion in the worst case, which happens when the energy is distributed uniformly across all entries of the input vector, and it is not easy to capture theoretically that this worst case situation will not happen repeatedly throughout the iterations. As a result, there is currently no separation between the worst case complexity of first order methods, such as gradient descent, combined with biased vs related unbiased compressors [Beznosikov et al., 2020]. If one makes a-priori statistical assumptions on the distribution of the messages/gradients that need to be compressed, such a separation can be made [Beznosikov et al., 2020]. While insightful, this is not satisfactory. Indeed, the gradients produced by methods such as gradient descent evolve in a non-stationary way, and hence modeling them as samples coming from a fixed distribution raises questions. Further, gradient compression affects the iterates and hence also the gradients that will be produced in all subsequent iterations, which is another phenomenon not captured by the aforementioned approach.

## 2.2   Error feedback: what it is good for, and what we still do not know

The difference between what we know about unbiased and biased compressors is larger still in the distributed setting.

In particular, *unbiasedness* turns out to be a very effective tool facilitating the analysis of distributed first order methods utilizing unbiased compressors, and for this reason, the landscape of methods using such compressors is very rich and relatively well understood. For example, using unbiased compressors we know how to

    i)  analyze distributed compressed gradient descent [Khirirat et al., 2018, Mishchenko et al., 2020],

   ii)  remove the variance introduced by compression to achieve faster convergence [Mishchenko et al., 2019, Horváth et al., 2019b, Mishchenko et al., 2020],

  iii)  perform bidirectional compression at the workers and also at the master [Horváth et al., 2019a, Philippenko and Dieuleveut, 2020, Gorbunov et al., 2020b],

  iv)  develop a general theory for SGD which, besides more standard methods, also includes variants using unbiased compression of (stochastic) gradients [Gorbunov et al., 2020a, Khaled et al., 2020, Li and Richtárik, 2020],

   v)  achieve Nesterov acceleration in the strongly convex regime [Li et al., 2020],

  vi)  how to analyze these methods in the nonconvex regime [Mishchenko et al., 2019, Horváth et al., 2019a, Li and Richtárik, 2020],

 vii)  achieve acceleration in the nonconvex regime [Gorbunov et al., 2021], and even how to

viii)  apply unbiased compressors to Hessian matrices to obtain communication-efficient second-order methods [Islamov et al., 2021].

The situation with *general* biased compressors (i.e., those that do not arise from unbiased compressors via scaling) is much more challenging. The key complication comes from the fact that their naive use within first order methods, such as gradient descent, can lead to divergence. We refer the reader to [Karimireddy et al., 2019, Counterexamples 1-3] and [Beznosikov et al., 2020, Example 1] for illustrative examples. In the latter, the gradient descent "enhanced" with the Top-1 compressor leads to *exponential divergence* when applied to the problem of minimizing the average of three strongly convex quadratics in $\mathbb{R}^3$. However, divergence of gradient descent enhanced with biased compressors such as Top-$k$ was observed empirically much sooner, and a fix for this problem, known as *error feedback* (EF), or *error compensation* (EC), was suggested by Seide et al. [2014]. This fix remained a heuristic until very recently.

The first theoretical breakthroughs focused on the simpler single-node setting [Stich et al., 2018, Alistarh et al., 2018]. The first analysis in the general distributed heterogeneous data[3] setting was performed by Beznosikov et al. [2020], and was confined to the strongly convex regime. While without compression, one can expect a linear rate, the rate in [Beznosikov et al., 2020] is linear only in the special case of an over-parameterized regime (i.e., the regime in which the loss functions on all nodes share a common minimizer) with a requirement of full gradient computations on each node. These deficiencies were later fixed by Gorbunov et al. [2020b], who developed the first linearly convergent methods EC-GD-DIANA and EC-LSVRG-DIANA, and also analyzed the convex case. Further, Qian et al. [2020a,b] showed that error-compensated methods can be *accelerated* in the sense of Nesterov [Nesterov, 2004] and can be combined with *variance reduction* techniques. However, these advances were achieved through the use of additional unbiased compressors or extra communication of full vectors, and hence via an increase in communication in each round. In the orthogonal line of works, Sun et al. [2020], Magnússon et al. [2019] designed special types of biased quantization operators. However, these quantization schemes are unable to compress gradients below $\mathcal{O}(d)$[4], and the suggested methods require hyperparameters which cannot be efficiently estimated.

> *In particular, whether it is possible to obtain a linearly convergent error-compensated method in the general heterogeneous data setting, relying on general biased compressors only, is still an open problem.*

The current state-of-the-art theoretical result for error-compensated methods in the smooth non-convex regime are due to Koloskova et al. [2020, Theorem 4.1], who consider the more general problem of decentralized optimization over a network. In the case when full (as opposed to stochastic) gradients are computed on each node, they show that after $T$ communication rounds it is possible to find a random vector $\hat{x}^T$ with the guarantee

$$\mathbb{E}\left[\left\|\nabla f(\hat{x}^T)\right\|^2\right] = \mathcal{O}\left(\frac{G^{2/3}}{T^{2/3}}\right), \tag{4}$$

under the *bounded gradient* assumption which requires the existence of a constant $G > 0$ such that

$$\|\nabla f_i(x)\|^2 \leq G^2 \tag{5}$$

holds for all $x \in \mathbb{R}^d$ and all $i \in \{1,2,\ldots,n\}$. This was a slight improvement in rate over an result obtained by Lian et al. [2017], who instead use the bounded dissimilarity assumption

$$\frac{1}{n}\sum_{i=1}^n \|\nabla f_i(x) - \nabla f(x)\|^2 \leq G^2. \tag{6}$$

A summary of the limitations of known results for EF-based methods is provided in Table 1.

> *In this work we argue that the bounded gradients* (5) *and bounded dissimilarity* (6) *assumptions are too strong[5], and that the sublinear rate* (4) *is not what one should expect from a good analysis of a well designed error-compensated first-order method. Instead, one could hope for the faster $\mathcal{O}(1/T)$ rate, which is what one obtains with methods using unbiased compressors [Gorbunov et al., 2021, Theorem 2.1]. The resolution of these issues is an open problem.*

## 3 Summary of contributions

In this work we address and resolve the aforementioned challenges. Our key contributions are:

---

[3]Problem (1) is in the *heterogeneous data regime* if no similarity among the functions (and hence among the data stored across different nodes giving rise to these functions) is assumed.

[4]This is very limiting as in large scale learning, $d$ is enormous, and one usually needs to use dramatic sparsification such as Top-1.

[5]The bounded gradient (5) and bounded dissimilarity (6) assumptions are too strong as they are rarely satisfied. For example, neither hold even for simple quadratic functions. To see this, let $f_i(x) = x^\top \mathbf{A}_i x$, where $\mathbf{A}_i \in \mathbb{R}^{d \times d}$. Since $\nabla f_i(x) = \mathbf{B}_i x$, where $\mathbf{B}_i = \mathbf{A}_i + \mathbf{A}_i^\top$, the bounded gradient assumption requires the vectors $\sup_x \max_i \|\mathbf{B}_i x\|$ to be bounded, which is not the case, unless all matrices $\mathbf{B}_i$ are zero. The bounded dissimilarity assumption (6), which can be written in the form $\frac{1}{n}\sum_{i=1}^n \|(\mathbf{B}_i - \frac{1}{n}\sum_{j=1}^n \mathbf{B}_j)x\|^2 \leq G^2$, also does not hold, unless $\mathbf{B}_i = \mathbf{B}_j$ for all $i,j$, which reduces to the identical data regime, which is of limited interest.

| Algorithm | sCVX | nCVX | DIST | key limitation |
|---|:---:|:---:|:---:|---|
| EF
Stich et al. [2018], Cordonnier [2018] | ✓ | ✗ | ✓ | bounded gradients;
sublinear rate in sCVX case |
| EF-SGD
Stich and Karimireddy [2019] | ✓ | ✓ | ✗ | single node only |
| EF
Ajalloeian and Stich [2020] | ✓ | ✓ | ✗ | single node only |
| SignSGD
Karimireddy et al. [2019] | ✗ | ✓ | ✗ | moment bound;
single node only |
| EC-SGD
Beznosikov et al. [2020] | ✓ | ✗ | ✓ | linear rate only
if $\nabla f_i(x^\star) = 0 \,\forall i$ |
| EC-SGD
Gorbunov et al. [2020b] | ✓ | ✗ | ✓ | linear rate only using
an extra unbiased compressor |
| DoubleSqueeze
Tang et al. [2020] | ✗ | ✓ | ✓ | bounded compression error;
slow $\mathcal{O}(1/T^{2/3})$ rate in nCVX case |
| Qsparse-SGD, CSER
Basu et al. [2019], Xie et al. [2020] | ✓ | ✓ | ✓ | bounded gradients;
slow $\mathcal{O}(1/T^{1/2})$ rate in nCVX case |
| EC-SGD
Koloskova et al. [2020] | ✗ | ✓ | ✓[†] | bounded gradients;
slow $\mathcal{O}(1/T^{2/3})$ rate in nCVX case |

Table 1: Known results for first order methods using biased compressors. sCVX = supports strongly convex functions, nCVX= supports nonconvex functions, DIST = works in the distributed regime.
[†]decentralized method

| Assumptions | Complexity | Theorem |
|---|:---:|:---:|
| $f_i$ is $L_i$-smooth
$f$ is lower bounded by $f^{\inf}$ | $\mathbb{E}\left[\left\|\nabla f(\hat{x}^T)\right\|^2\right] \leq \frac{2\left(f(x^0) - f^{\inf}\right)}{\gamma T} + \frac{\mathbb{E}[G^0]}{\theta T}$ | 2 |
| $f_i$ is $L_i$-smooth
$f$ is lower bounded by $f^{\inf}$
$f$ satisfies PL condition | $\mathbb{E}\left[\Psi^T\right] \leq (1 - \gamma\mu)^T \mathbb{E}\left[\Psi^0\right]$ | 3 |

Table 2: Summary of complexity results obtained in this paper. Quantities: $\mu$ = PL constant; $\gamma$ = stepsize; $G^0$ = see (14); $\Psi^t$ = Lyapunov function defined in Theorem 3.

**A. New error feedback mechanism.** We propose a new error feedback (resp. error compensation) mechanism, which we call EF21 (resp. EC21) – see Algorithms 1 and 2. Unlike most results on error compensation, EF21 naturally works in the distributed heterogeneous data setting.

**B. Standard assumptions and fast rates.** Our theoretical analysis of EF21 relies on standard assumptions only, which are:

i) $L_i$-smoothness of the individual functions $f_i$, and
ii) existence of a global lower bound $f^{\inf} \in \mathbb{R}$ on $f$.

We prove that under these assumptions, EF21 enjoys the desirable $\mathcal{O}(1/T)$ convergence rate, which improves upon the previous $\mathcal{O}(1/T^{2/3})$ state-of-the-art result of Koloskova et al. [2020] both in terms of the rate, and in terms of the strength of the assumptions needed to obtain this result. These complexity results are summarized in the first row of Table 2.

**C. Linear rate for Polyak-Lojasiewicz functions.** We show that under the additional assumption that $f$ satisfies the Polyak-Lojasiewicz inequality, EF21 enjoys a linear convergence rate. This improves upon the results of Beznosikov et al. [2020], who only obtain a linear rate in the case when $\nabla f_i(x^\star) = 0$ for all $i$, where $x^\star = \arg\min f$, and provides an alternative to the linear convergence results of Gorbunov et al. [2020b], who needed to introduce additional unbiased compressors into their scheme, and hence additional communication, in order to obtain their results. Our complexity results are summarized in the second row of Table 2.

**D. Empirical superiority.** We show through extensive numerical experimentation on both synthetic problems and deep learning benchmarks that EF21 consistently and substantially outperforms EF in practice. One of the reasons behind this is the fact that our method is able to admit much larger learning rates. Since EF has a large number of applications

where it reigns supreme, we believe that EF21 will have a large impact on the practice of communication efficient distributed learning.

**E. A more aggressive variant.** We further propose a more aggressive variant, EF21+ (see Section 4.6), which has an even better empirical behavior. We show that if $\mathcal{C}$ is deterministic, the same theorems capturing the convergence of EF21 hold for EF21+ as well.

**F. Stochastic setting.** We describe an extension to the stochastic setting, i.e., when each node computes a stochastic gradient instead of the exact/full gradient, in Appendix H.

## 4 Main Results

Since we are about to re-engineer the classical error feedback technique, it will be useful to take a step back and re-examine the issues inherent to the simplest first order method which uses biased compressors but does *not* employ error feedback: distributed compressed gradient descent (DCGD).

Let $x^t$ be the $t$-th iterate, shared by all $n$ nodes. Each node $i$ first computes its local gradient $\nabla f_i(x^t)$, compresses it using some $\mathcal{C} \in \mathbb{B}(\alpha)$, and sends the compressed gradient $\mathcal{C}(\nabla f_i(x^t))$ to the master. The master aggregates all $n$ messages via averaging, and performs the optimization step

$$x^{t+1} = x^t - \frac{\gamma}{n} \sum_{i=1}^n \mathcal{C}(\nabla f_i(x^t)). \tag{7}$$

As mentioned before, this method can diverge, even in simple quadratic problems in low dimensions [Beznosikov et al., 2020]. Let us look at this problem from a different angle. Assume, for the sake of an intuitive argument, that the sequence of iterates actually converges to some $x^\dagger$. Since in general there is no reason for the gradients $\nabla f_i(x^\dagger)$ to be all zero, even if $x^\dagger$ is the minimizer of $f$, the application of $\mathcal{C}$ to the gradients $\nabla f_i(x^t)$ will introduce a nonzero distortion even if $x^t \approx x^\dagger$. Indeed, in view of (3), all that can be guaranteed is that

$$\mathbb{E}\left[\left\|\mathcal{C}(\nabla f_i(x^t)) - \nabla f_i(x^t)\right\|^2\right] \leq (1-\alpha) \left\|\nabla f_i(x^t)\right\|^2,$$

which can be large if the norm of $\nabla f_i(x^t)$ is large. So, the method is intrinsically unstable around $x^\dagger$, and hence can not converge to $x^\dagger$.

Our idea is to fix this issue by *compressing different vectors* instead of the gradients, vectors that would hopefully converge to zeros instead. Since in view of (3) the application of $\mathcal{C}$ to progressively vanishing vectors introduces progressively vanishing distortion, the stabilization problem would be solved. But what vectors should we compress? In order to answer this question, it will be useful to consider a simpler and more abstract setting first, which we shall do next.

### 4.1 Markov compressors

Assume we are given a sequence of input vectors $\{v^t\}_{t \geq 0}$ (e.g., gradients) generated by some algorithm. This sequence does not necessarily converge to zero. Our goal is to produce a sequence of "good" and "easy to communicate" (to some entity, which we shall call the "master") estimates of these vectors, making use of a compressor $\mathcal{C} \in \mathbb{B}(\alpha)$. Let us proceed through several steps of discovery.

**Naive idea.** The first and naive approach, described above, is to simply output the sequence of compressed inputs: $\{\mathcal{C}(v^t)\}_{t \geq 0}$. However, while these estimates can be communicated efficiently, they are not getting "better". That is, the distortion $\mathbb{E}\left[\left\|\mathcal{C}(v^t) - v^t\right\|^2\right]$ is not necessarily improving.

**Good but not implementable idea.** What can we do better? Consider the following idea. If we knew, hypothetically, the limit of this sequence, $v^*$, we could output $v^* + \mathcal{C}(v^t - v^*)$ at iteration $t$ instead. Since $v^t \to v^*$, the distortion between the input and the output at iteration $t$ is

$$\mathbb{E}\left[\left\|v^* + \mathcal{C}(v^t - v^*) - v^t\right\|^2\right] = \mathbb{E}\left[\left\|\mathcal{C}(v^t - v^*) - (v^t - v^*)\right\|^2\right] \overset{(3)}{\leq} (1-\alpha) \left\|v^t - v^*\right\|^2 \to 0.$$

So, the distortion issue is fixed! Moreover, if we assume the master knows $v^*$, then the output vector at each iteration can be communicated cheaply as well, since all we need to communicate is the

---
**Algorithm 1** EF21 (Single node)
---
1: **Input:** starting point $x^0 \in \mathbb{R}^d$, learning rate $\gamma > 0$, $g^0 = \mathcal{C}(\nabla f(x^0))$
2: **for** $t = 0, 1, 2, \ldots, T-1$ **do**
3: $\quad x^{t+1} = x^t - \gamma g^t$
4: $\quad g^{t+1} = g^t + \mathcal{C}(\nabla f(x^{t+1}) - g^t)$
5: **end for**
---

compressed vector $\mathcal{C}(v^t - v^*)$. It will be useful to think of this operation as a new compressor, called $\mathcal{C}_{v^*}$, one that takes $v^t$ as an input, and gives $v^* + \mathcal{C}(v^t - v^*)$ as its output. That is, we can define

$$\mathcal{C}_{v^*}(v) \stackrel{\text{def}}{=} v^* + \mathcal{C}(v - v^*). \tag{8}$$

While the compressor $\mathcal{C}_{v^*}$ satisfies all our requirements, it is not implementable, since the vector $v^*$ is not known. We will now use this intuition to construct an implementable mechanism.

**Good and implementable idea.** In the above construction, we have used the fact that $v^t - v^* \to 0$ to construct a good mechanism, but one that is not implementable. How can we fix this issue? The rescue comes from the *recursive* observation that if we indeed succeed in constructing a compressor, let's call it $\mathcal{M}$, such that the distortion between $\mathcal{M}(v^t)$ and $v^t$ vanishes as $t \to \infty$, then it must be the case that $v^t - \mathcal{M}(v^t) \to 0$. So, we can compress *this* vanishing vector instead. This idea gives rise to the following recursive definition of $\mathcal{M}$:

$$\mathcal{M}(v^0) \quad \stackrel{\text{def}}{=} \quad \mathcal{C}(v^0) \tag{9}$$
$$\mathcal{M}(v^{t+1}) \quad \stackrel{\text{def}}{=} \quad \mathcal{M}(v^t) + \mathcal{C}(v^{t+1} - \mathcal{M}(v^t)), \quad t \geq 0 \tag{10}$$

Note that (10) is similar to (8), with one key difference: we are using the previously compressed vector $\mathcal{M}(v^t)$, which is *known*, instead of the limit vector $v^*$, which is *unknown*. This property also makes our new compressor non-stationary, i.e., it has a Markov property.

It is easy to establish (see Appendix B) that under some assumptions about the speed at which the input sequence $v^t$ converges to $v^*$, it will be the case that $\mathbb{E}\left[\|\mathcal{M}(v^t) - v^t\|^2\right] \to 0$. For instance, if the convergence rate of the input sequence is linear, then the distortion will converge to 0. While this is interesting on its own, let us deploy our new tool, which we call *Markov compressor*, in the context of gradient descent, and then in the context of distributed gradient descent.

### 4.2 Compressed gradient descent using the Markov compressor

For simplicity, consider solving problem (1) in the $n = 1$ case, i.e., the problem

$$\min_{x \in \mathbb{R}^d} f(x), \tag{11}$$

using the compressed gradient descent method featuring the Markov compressor. Start with $x^0 \in \mathbb{R}^d$, stepsize $\gamma > 0$, and let

$$\mathcal{M}(\nabla f(x^0)) = \mathcal{C}(\nabla f(x^0)).$$

After this, for $t \geq 0$ iterate:

$$x^{t+1} \quad = \quad x^t - \gamma \mathcal{M}(\nabla f(x^t)) \tag{12}$$
$$\mathcal{M}(\nabla f(x^{t+1})) \quad = \quad \mathcal{M}(\nabla f(x^t)) + \mathcal{C}(\nabla f(x^{t+1}) - \mathcal{M}(\nabla f(x^t))). \tag{13}$$

Note that the situation here is more complicated than the abstract setting described earlier since now there is *interaction* between the input sequence $\{\nabla f(x^t)\}_{t \geq 0}$ of gradients and the sequence $\mathcal{M}(\nabla f(x^t))$ of compressed gradients via the Markov compressor. Indeed, the output of $\mathcal{M}$ at iteration $t$ influences the next iterate $x^{t+1}$ (via (12)), which in turn defines the next input vector $v^{t+1} = \nabla f(x^{t+1})$ in the sequence, and so on.

To lighten up the heavy notation in (12) and (13), it will be useful to write $g^t = \mathcal{M}(\nabla f(x^t))$. Using this new notation that hides the fact that $g^t$ is the application of the Markov compressor to the gradient, the method described above is formalized as Algorithm 1. This is precisely our proposed new variant of error feedback, EF21, specialized to the single node problem (11).

---

**Algorithm 2** EF21 (Multiple nodes)

---

1: **Input:** starting point $x^0 \in \mathbb{R}^d$; $g_i^0 = \mathcal{C}(\nabla f_i(x^0))$ for $i = 1, \ldots, n$ (known by nodes and the master); learning rate $\gamma > 0$; $g^0 = \frac{1}{n} \sum_{i=1}^n g_i^0$ (known by master)
2: **for** $t = 0, 1, 2, \ldots, T - 1$ **do**
3:     Master computes $x^{t+1} = x^t - \gamma g^t$ and broadcasts $x^{t+1}$ to all nodes
4:     **for all nodes** $i = 1, \ldots, n$ **in parallel do**
5:         Compress $c_i^t = \mathcal{C}(\nabla f_i(x^{t+1}) - g_i^t)$ and send $c_i^t$ to the master
6:         Update local state $g_i^{t+1} = g_i^t + \mathcal{C}(\nabla f_i(x^{t+1}) - g_i^t)$
7:     **end for**
8:     Master computes $g^{t+1} = \frac{1}{n} \sum_{i=1}^n g_i^{t+1}$ via $g^{t+1} = g^t + \frac{1}{n} \sum_{i=1}^n c_i^t$
9: **end for**

---

### 4.3 Distributed variant of EF21

The main method of this paper, which we now present as Algorithm 2, is an extension of Algorithm 1 to the general finite-sum problem (1). In particular, we apply the Markov compressor individually on each node to the local gradients $\nabla f_i(x^t)$, and communicate the compressed gradients to the master. Recall that we only need to communicate the vectors $\mathcal{C}(\nabla f_i(x^{t+1}) - g_i^t)$ since the additive terms $g_i^t$ appearing in the Markov compressor were communicated in the previous round. Master then averages all gradient estimators, obtaining $g^{t+1} = \frac{1}{n} \sum_{i=1}^n g_i^{t+1}$, which can be done by performing the calculation $g^{t+1} = g^t + \frac{1}{n} \sum_{i=1}^n c_i^t$, where $g^t$ is the average from the previous round which the master maintains, and $c_i^t = \mathcal{C}(\nabla f_i(x^{t+1}) - g_i^t)$ are the compressed messages. After this, the master takes a gradient-like step, and broadcasts the new model to all nodes.

### 4.4 Relationship between EF and EF21

While this is not at all apparent at first sight, it turns out that EF and EF21 are related. In particular, under certain conditions on the compressor $\mathcal{C}$, which are not met in practice, they are identical.

**Theorem 1.** *Assume that $\mathcal{C}$ is deterministic, positive homogeneous and additive. Then* EF *(Algorithm 3; see appendix) and* EF21 *produce the same sequences of iterates* $\{x^t\}_{t \geq 0}$.

Note that while the Top-$k$ compressor is deterministic and positively homogeneous, it is not additive. Likewise, compressors arising via rescaling of unbiased compressors are randomized, and hence do not satisfy the first condition. Still, the above theorem sheds some (at least to us) unexpected light on the close connection between EF and our new variant, EF21. This connection is also what justifies our naming decision: EF21 – error feedback mechanism from the year 2021.

### 4.5 Theory

We make the following assumption throughout:

**Assumption 1** (Smoothness and lower boundedness). *Every $f_i$ has $L_i$-Lipschitz gradient, i.e.,*

$$\|\nabla f_i(x) - \nabla f_i(y)\| \leq L_i \|x - y\|$$

*for all $x, y \in \mathbb{R}^d$, and $f^{\inf} \overset{def}{=} \inf_{x \in \mathbb{R}^d} f(x) > -\infty$.*

If each $f_i$ has $L_i$-Lipschitz gradient, then it is straightforward to check by Jensen's inequality that $f$ is $L$-Lipschitz, with $L$ satisfying the inequality $L \leq \frac{1}{n} \sum_i L_i$. It will be also useful to define $\widetilde{L} \overset{def}{=} (\frac{1}{n} \sum_{i=1}^n L_i^2)^{1/2}$. By the arithmetic-quadratic mean inequality, we have $\frac{1}{n} \sum_i L_i \leq \widetilde{L}$. Let

$$G^t \overset{def}{=} \frac{1}{n} \sum_{i=1}^n \left\| g_i^t - \nabla f_i(x^t) \right\|^2, \tag{14}$$

a quantity which will appear in both our theorems. In EF21 we use $\mathcal{C} \in \mathbb{B}(\alpha)$, where $0 < \alpha \leq 1$, and define $\theta = 1 - \sqrt{1 - \alpha}$ and $\beta = \frac{1 - \alpha}{1 - \sqrt{1 - \alpha}}$. We now formulate our first complexity result.

**Theorem 2.** *Let Assumption 1 hold, and let the stepsize in Algorithm 2 be set as*

$$0 < \gamma \leq \left( L + \widetilde{L}\sqrt{\frac{\beta}{\theta}} \right)^{-1}. \tag{15}$$

*Fix $T \geq 1$ and let $\hat{x}^T$ be chosen from the iterates $x^0, x^1, \ldots, x^{T-1}$ uniformly at random. Then*

$$\mathbb{E}\left[ \left\| \nabla f(\hat{x}^T) \right\|^2 \right] \leq \frac{2\left( f(x^0) - f^{inf} \right)}{\gamma T} + \frac{\mathbb{E}\left[ G^0 \right]}{\theta T}. \tag{16}$$

Note that $\sqrt{\beta/\theta} = \frac{(1+\sqrt{1-\alpha})}{\alpha} - 1 \leq \frac{2}{\alpha} - 1$ is decreasing in $\alpha$. This makes sense since larger $\alpha$ means less dramatic compression, which leads to smaller $\sqrt{\beta/\theta}$, and this through (15) allows for larger stepsize, and hence fewer communication rounds. We now introduce the PL assumption, which enables us to obtain a linear convergence result.

**Assumption 2** (Polyak-Lojasiewicz). *There exists $\mu > 0$ such that $f(x) - f(x^\star) \leq \frac{1}{2\mu} \left\| \nabla f(x) \right\|^2$ for all $x \in \mathbb{R}^d$, where $x^\star = \arg\min_{x \in \mathbb{R}^d} f(x)$.*

**Theorem 3.** *Let Assumptions 1 and 2 hold, and let the stepsize in Algorithm 2 be set as*

$$0 < \gamma \leq \min\left\{ \left( L + \widetilde{L}\sqrt{\frac{2\beta}{\theta}} \right)^{-1}, \frac{\theta}{2\mu} \right\}. \tag{17}$$

*Let $\Psi^t \overset{def}{=} f(x^t) - f(x^\star) + \frac{\gamma}{\theta} G^t$. Then for any $T \geq 0$, we have*

$$\mathbb{E}\left[ \Psi^T \right] \leq (1 - \gamma\mu)^T \mathbb{E}\left[ \Psi^0 \right]. \tag{18}$$

Our theorems hold for an arbitrary choice of the initial vectors $\{g_i^0\}$, and not just for $g_i^0 = \mathcal{C}(\nabla f_i(x^0))$. For instance, if $g_i^0 = \nabla f_i(x^0)$ is used, then $\mathbb{E}\left[ G^0 \right] = 0$, and the second term in (16) and the last term in $\mathbb{E}\left[ \Psi^0 \right]$ vanish.

### 4.6  EF21+: Use $\mathcal{C}$ or the Markov compressor, whichever is better

We now briefly describe a new hybrid method, called EF21+, which often performs particularly well in practice. Full description and the analysis of EF21+ are deferred to Section G. In every communication round, EF21+ allows each node to compress using the "best" of $\mathcal{C}$ and the Markov compressor generated. So, EF21+ can be thought of as a hybrid between DCGD (see (7)) and EF21. The decision about which compressor to use is made by each node $i$ individually, based on which of the distortions $\|\mathcal{C}(s) - s\|$ and $\|\mathcal{M}(s) - s\|$ is smaller, where $s = \nabla f_i(x^{t+1})$.

### 4.7  Dealing with stochastic gradients

In Section H (see Algorithm 5) we describe a natural extension of EF21 to the setting where full gradient computations are replaced by stochastic gradient estimators, i.e., we use a random vector $\hat{g}_i^t \approx \nabla f_i(x^t)$. We also outline how convergence analysis is performed in this regime.

## 5  Experiments

We first consider solving a logistic regression problem with a non-convex regularizer,

$$f(x) = \frac{1}{N} \sum_{i=1}^{N} \log\left( 1 + \exp\left( -y_i a_i^\top x \right) \right) + \lambda \sum_{j=1}^{d} \frac{x_j^2}{1 + x_j^2}, \tag{19}$$

where $a_i \in \mathbb{R}^d, y_i \in \{-1,1\}$ are the training data, and $\lambda > 0$ is the regularizer parameter. We set $\lambda = 0.1$ in all experiments. At each set of experiments we use Top-$k$ with $k = 1$ [Alistarh et al., 2017] as a canonical example of biased compressor $\mathcal{C}$.

| Dataset | $n$ | $N$ (total # of datapoints) | $d$ (# of features) | $N_i$ (# of datapoints per client) |
|---|---|---|---|---|
| phishing | 20 | 11,055 | 68 | 552 |
| mushrooms | 20 | 8,120 | 112 | 406 |
| a9a | 20 | 32,560 | 123 | 1,628 |
| w8a | 20 | 49,749 | 300 | 2,487 |

Table 3: Summary of the datasets and splitting of the data among clients.

**Datasets.** The datasets were taken from LibSVM [Chang and Lin, 2011], and were split into $n = 20$ equal parts, each associated with one of 20 clients. The last part, of size $N - 20 \cdot \lfloor N/20 \rfloor$, was assigned to the last worker. That is, we consider the distributed regime with heterogeneous data. A summary can be found in Table 3. The details on hardware and implementation can be found in Section A.

**Experiment 1: Stepsize tolerance.** In our first experiment (see Figure 1) we test the robustness/tolerance of EF, EF21, and EF21+ to large stepsizes. Note that while for all stepsize choices, EF gets stuck at a certain accuracy level, EF21 and EF21+ do not suffer from this issue, and are hence able to work with larger or even much larger stepsizes.

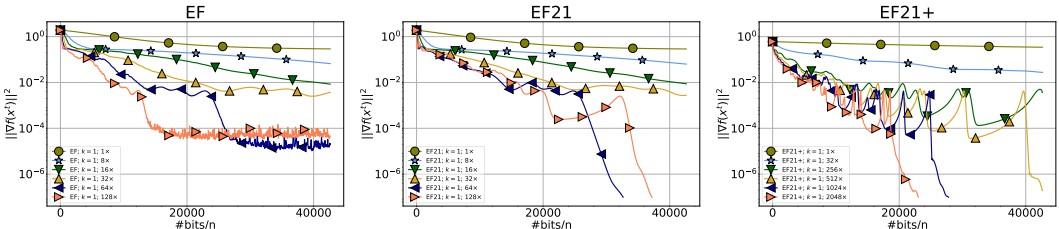

Figure 1: The performance of EF, EF21, and EF21+ with Top-1 compressor, and for increasing stepsizes. Representative dataset used: a9a. By $1\times, 2\times, 4\times$ (and so on) we indicate that the stepsize was set to a multiple of the largest stepsize predicted by our theory.

**Experiment 2: Fine-tuning $k$ and the stepsizes.** We now showcase the superior communication efficiency of EF21 and EF21+ over classical EF. In this set of experiments, we fine-tuned $k$ (for Top-$k$) and stepsizes individually for each method (details are given in Appendix A). For comparison, we also included distributed gradient descent (GD), which can be seen as EF21 with $k = d$ (no compression), into the mix.

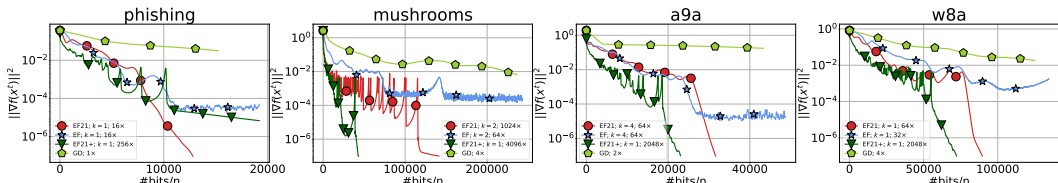

Figure 2: Comparison of EF21, EF21+ to EF with Top-$k$ for individually fine-tuned $k$ and fine-tuned stepsizes for all methods.

In Figure 2 we can see that in all cases, the proposed methods outperform EF in terms of the the number of bits sent to the server per client ($\text{bits}/n$), and rapidly converge to the desired accuracy, whereas EF is stuck at some accuracy levels in all cases. Moreover, in all experiments, classical GD shows the worst convergence rate. Note that EF21 tolerates larger, and EF21+ much larger, stepsizes than EF.

**Further experiments.** Further experiments, including deep learning experiments, are presented in Appendix A.

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
