# Contents

# Appendix

## A   Extra Experiments

We now present several additional experiments. First, in Section A.1 we comment on experiments with nonconvex logistic regression (see (19)), in Section A.2 we perform experiments on least-squares problem (as an example of a function that is not strongly convex but satisfies the PL inequality), and finally, in Section A.3 we conduct several deep learning experiments. The code is available at `github.com/IgorSokoloff/ef21_experiments_source_code`.

### A.1   Experiments with nonconvex logistic regression

#### A.1.1   Experiment 1: Stepsize tolerance (extension)

This sequence of experiments extends the results presented in the corresponding paragraph of Section 5. For each dataset, we select the parameter $k$ (varied by rows) within the powers of 2. For each plot, we vary the stepsize within the powers of 2 starting from the largest theoretically accepted $\gamma$.

For example, for $k = 2$ and `EF21+` with `mushrooms` dataset we consider factors from the set

$$\{1, 2, 4, 8, 16, 32, 64, 128, 256, 512, 1024, 2048\}$$

and select the stepsize as a multiple of the upper bound stated in Theorem 2.

Red diamond markers indicate the iterations at which `EF21+` method uses mostly DCGD steps. Precisely, the red diamond marker appears on the plot if the distortion $\|\mathcal{C}(s) - s\|$ is smaller that $\|\mathcal{M}(s) - s\|$ for at least half of the workers, where $s = \nabla f_i(x^{t+1})$. For more details, see figures below, where parameter $k$ is fixed within each row and each column corresponds to a particular method.

All of the figures above illustrate that `EF21` and `EF21+` tolerates much larger stepsizes, which makes them more efficient in practice. Moreover, in all experiments with large stepsizes ($16\times$–$128\times$), EF starts oscillating, which hinders the convergence to the desired tolerance

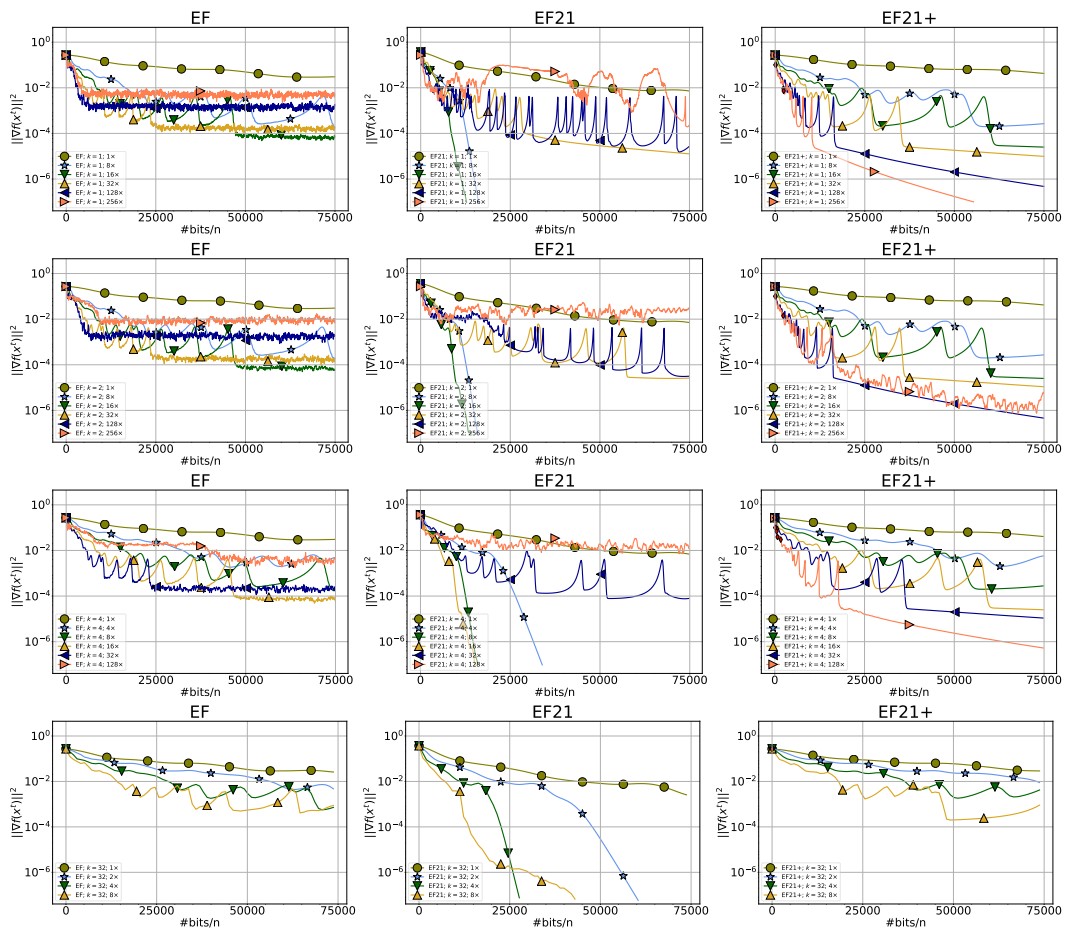

Figure 3: The performance of EF, EF21, and EF21+ with Top-$k$ compressor, and for increasing stepsizes. Each row corresponds to a different value of $k \in \{1, 2, 4, 32\}$. The dataset used: phishing. By $1\times, 2\times, 4\times$ (and so on) we indicate that the stepsize was set to a multiple of the largest stepsize predicted by our theory.

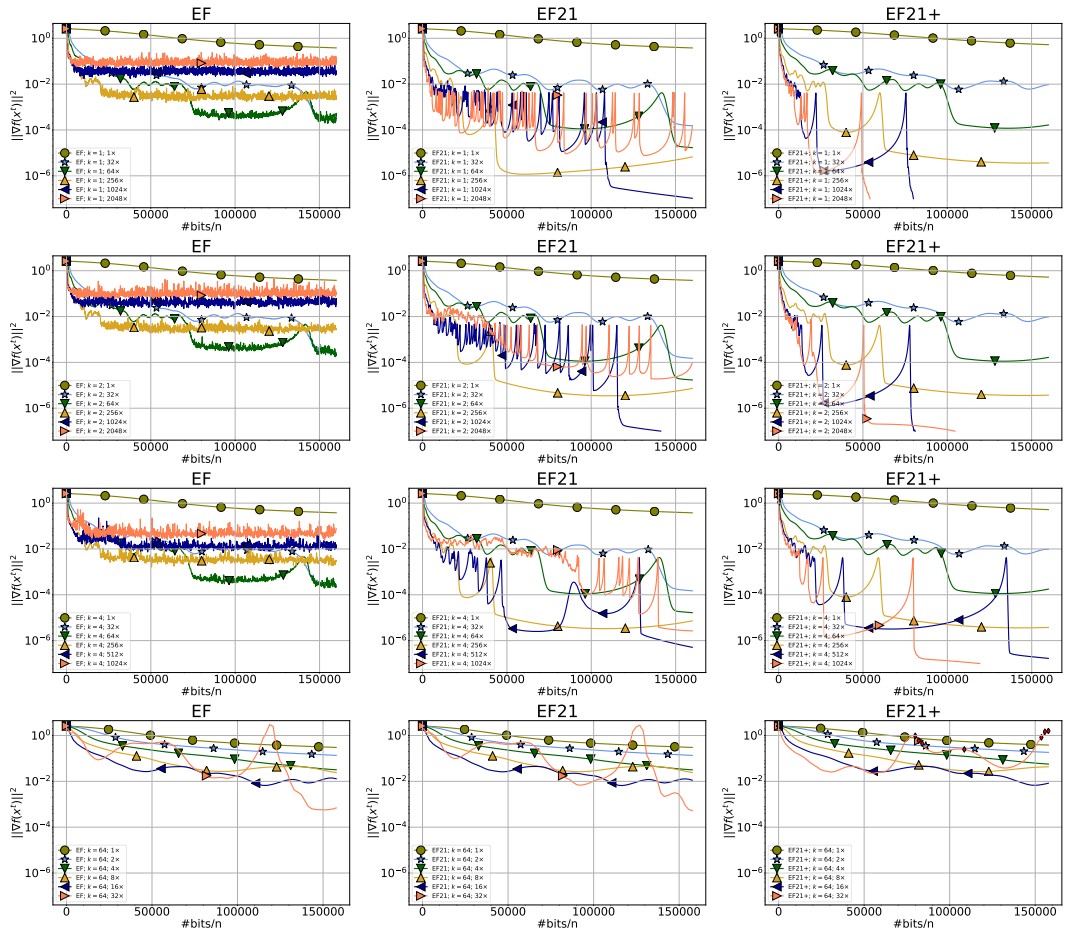

Figure 4: The performance of EF, EF21, and EF21+ with Top-$k$ compressor, and for increasing step-sizes. Each row corresponds to a different value of $k \in \{1, 2, 4, 64\}$. The dataset used: mushrooms. By $1\times, 2\times, 4\times$ (and so on) we indicate that the stepsize was set to a multiple of the largest stepsize predicted by our theory.

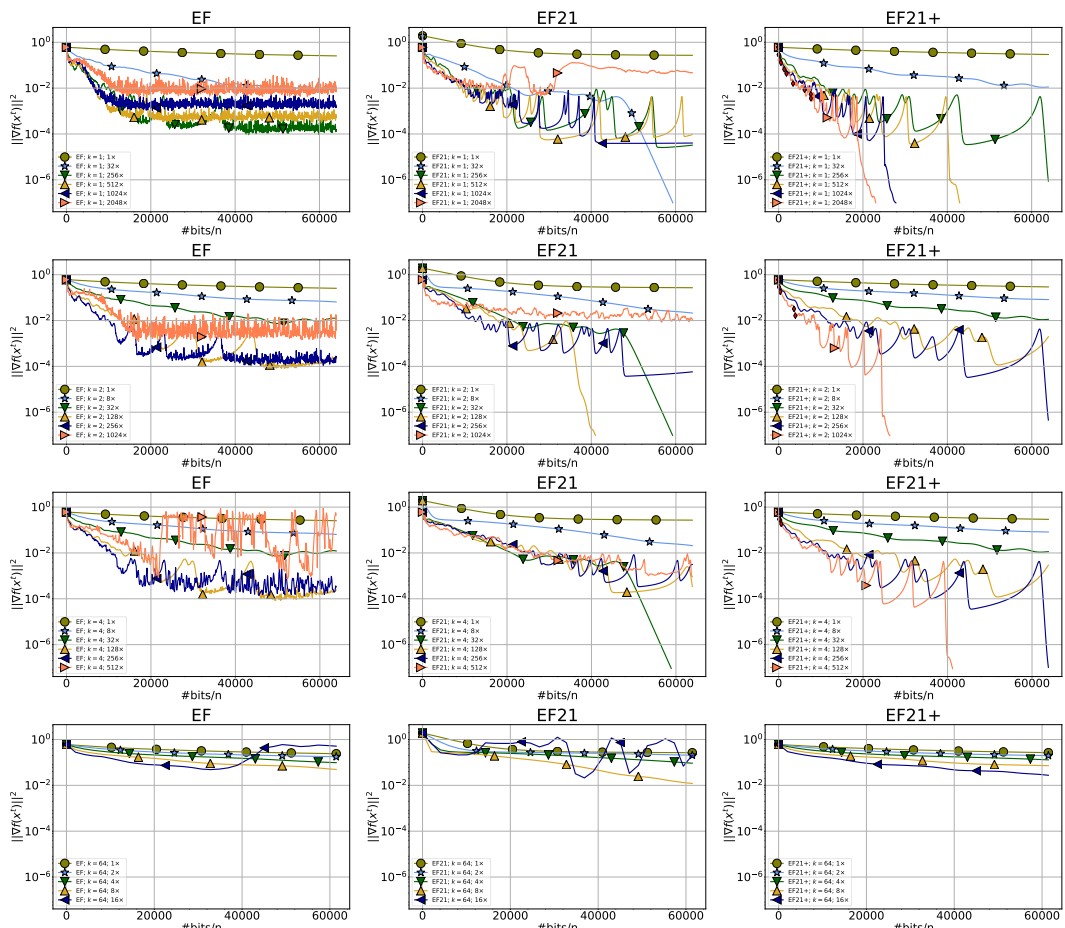

Figure 5: The performance of EF, EF21, and EF21+ with Top-$k$ compressor, and for increasing stepsizes. Each row corresponds to a different value of $k \in \{1, 2, 4, 64\}$. The dataset used: a9a. By $1\times, 2\times, 4\times$ (and so on) we indicate that the stepsize was set to a multiple of the largest stepsize predicted by our theory.

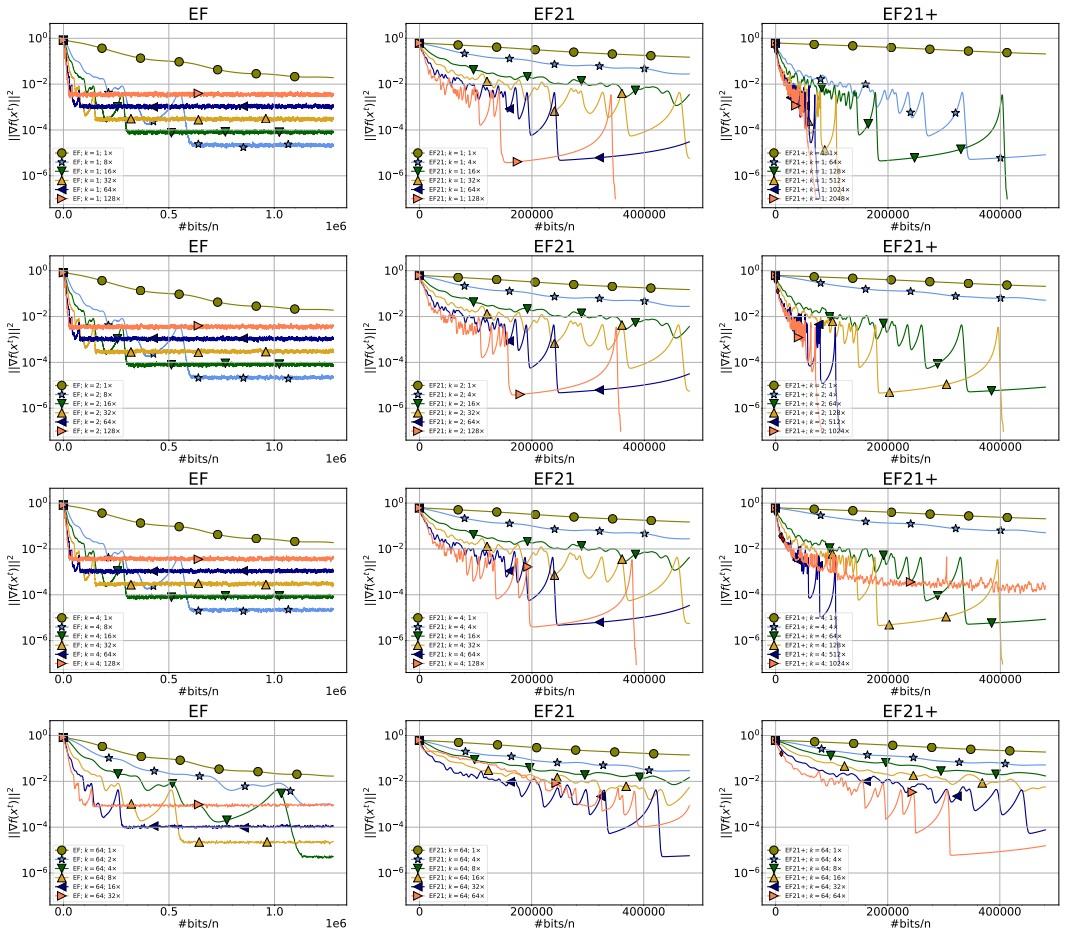

Figure 6: The performance of EF, EF21, and EF21+ with Top-$k$ compressor, and for increasing stepsizes. Each row corresponds to a different value of $k \in \{1, 2, 4, 64\}$. The dataset used: w8a. By $1\times, 2\times, 4\times$ (and so on) we indicate that the stepsize was set to a multiple of the largest stepsize predicted by our theory.

### A.1.2 Experiment 2: Fine-tuning $k$ and the stepsizes (extension)

This sequence of experiments extends the results presented in the similar paragraph of Section 5. In these plots we focus on the effect of the parameter $k$ on convergence. For each method, dataset, and $k$, the stepsize is fine-tuned (based on the fine-tuning results from Section A.1.1). Note that the theoretical stepsize allowed by Theorem 2 increases with the increase of $k$.

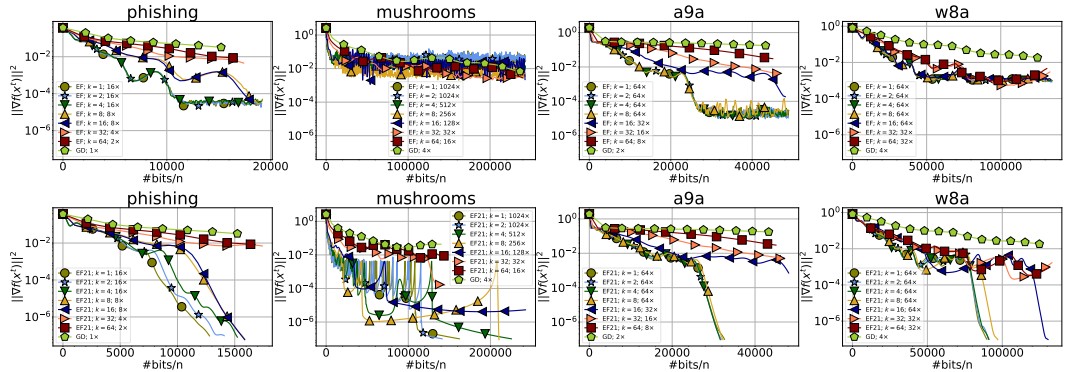

Figure 7: Effect of the parameter $k$ on convergence. For each method, dataset and $k$ the stepsize is fine-tuned. By $1\times, 2\times, 4\times$ (and so on) we indicate that the stepsize was set to a multiple of the largest stepsize predicted by our theory.

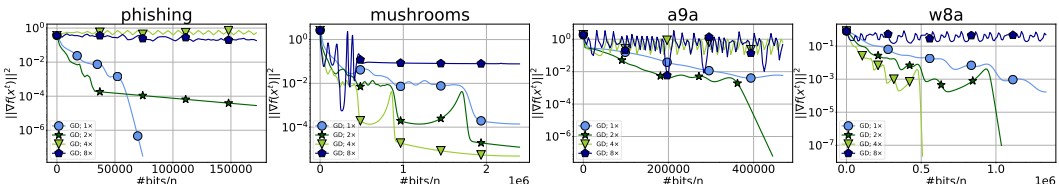

Figure 8: GD tuning.

We see that the best choice of $k$ relates to $1, 2$ or $4$, which confirms that both EF21 and EF are more communication efficient compared to GD.

## A.2 Experiments with least squares

In this section we will test on a function satisfying the PL condition (see Assumption 2). In particular, we consider the function

$$f(x) = \frac{1}{N} \sum_{i=1}^{N} (a_i^\top x - y_i)^2,$$

where $a_i \in \mathbb{R}^d, y_i \in \{-1, 1\}$ are the training data, and $N$ is the total number of datapoints. We consider the same datasets as for the logistic regression problem (see Table 3).

### A.2.1 Experiment 1: Stepsize tolerance

In this set of experiments we test the robustness/tolerance of EF, EF21, and EF21+ to large stepsizes, using Top-$k$ [Alistarh et al., 2017] as a canonical example of biased compressor $\mathcal{C}$. For each plot, we vary the stepsize within the powers of 2 starting from the largest theoretically accepted $\gamma$. For example, for $k = 2$ and EF21+ with mushrooms dataset we consider factors from the set $\{1, 4, 64, 256, 1024\}$ and select the stepsize as a multiple of the upper bound stated in Theorem 2. Red diamond markers indicate the iterations at which EF21+ method uses mostly DCGD steps. More precisely, the red diamond marker appears on the plot if the distortion $\|\mathcal{C}(s) - s\|$ is smaller that $\|\mathcal{M}(s) - s\|$ for at least half of the workers, where $s = \nabla f_i(x^{t+1})$. For more details, see Figures 9–12, where parameter $k$ is fixed within each row and each column correspond to a particular method.

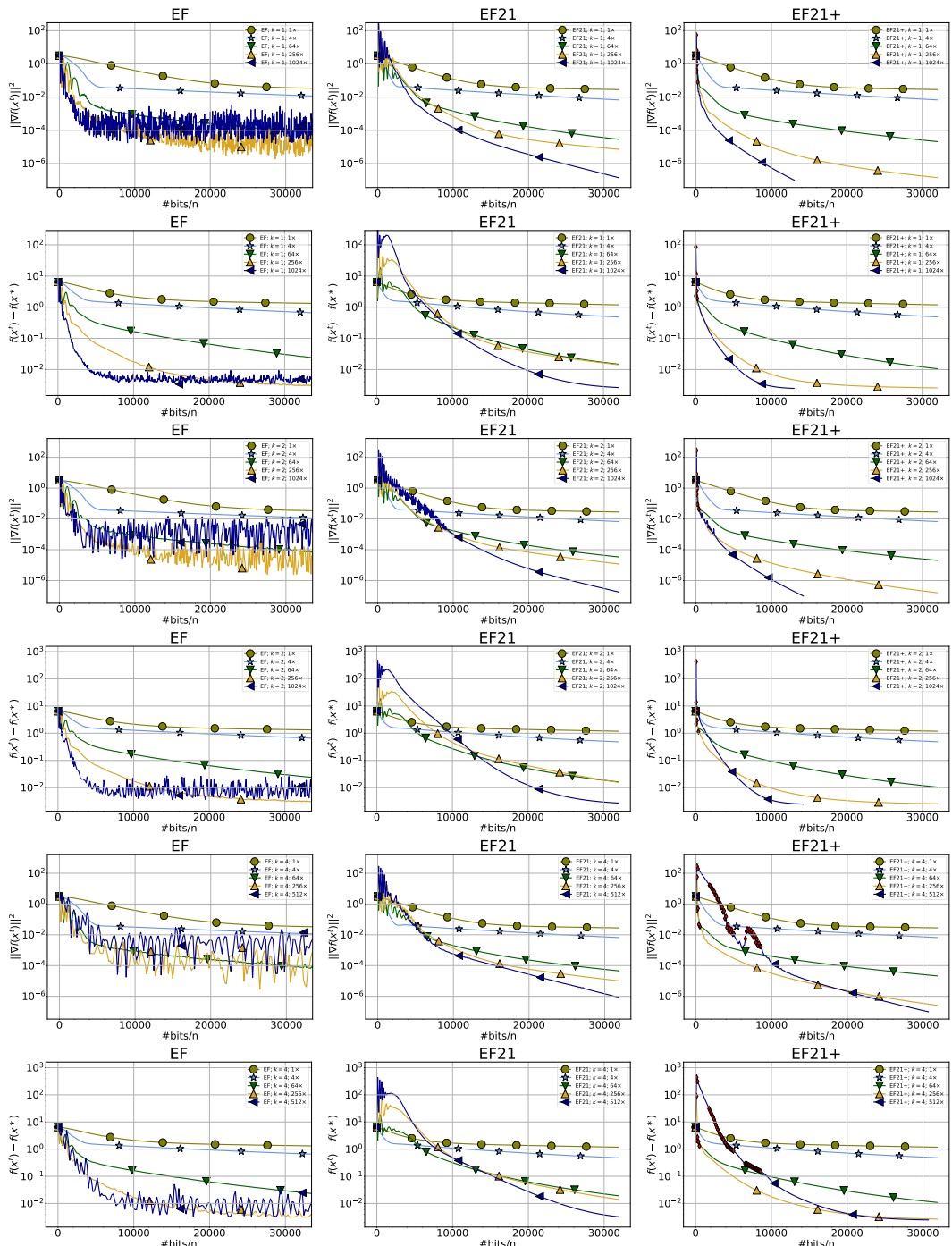

Figure 9: The performance of EF, EF21, and EF21+ with Top-$k$ compressor, and for increasing stepsizes. The dataset used: phishing. By $1\times, 2\times, 4\times$ (and so on) we indicate that the stepsize was set to a multiple of the largest stepsize predicted by our theory.

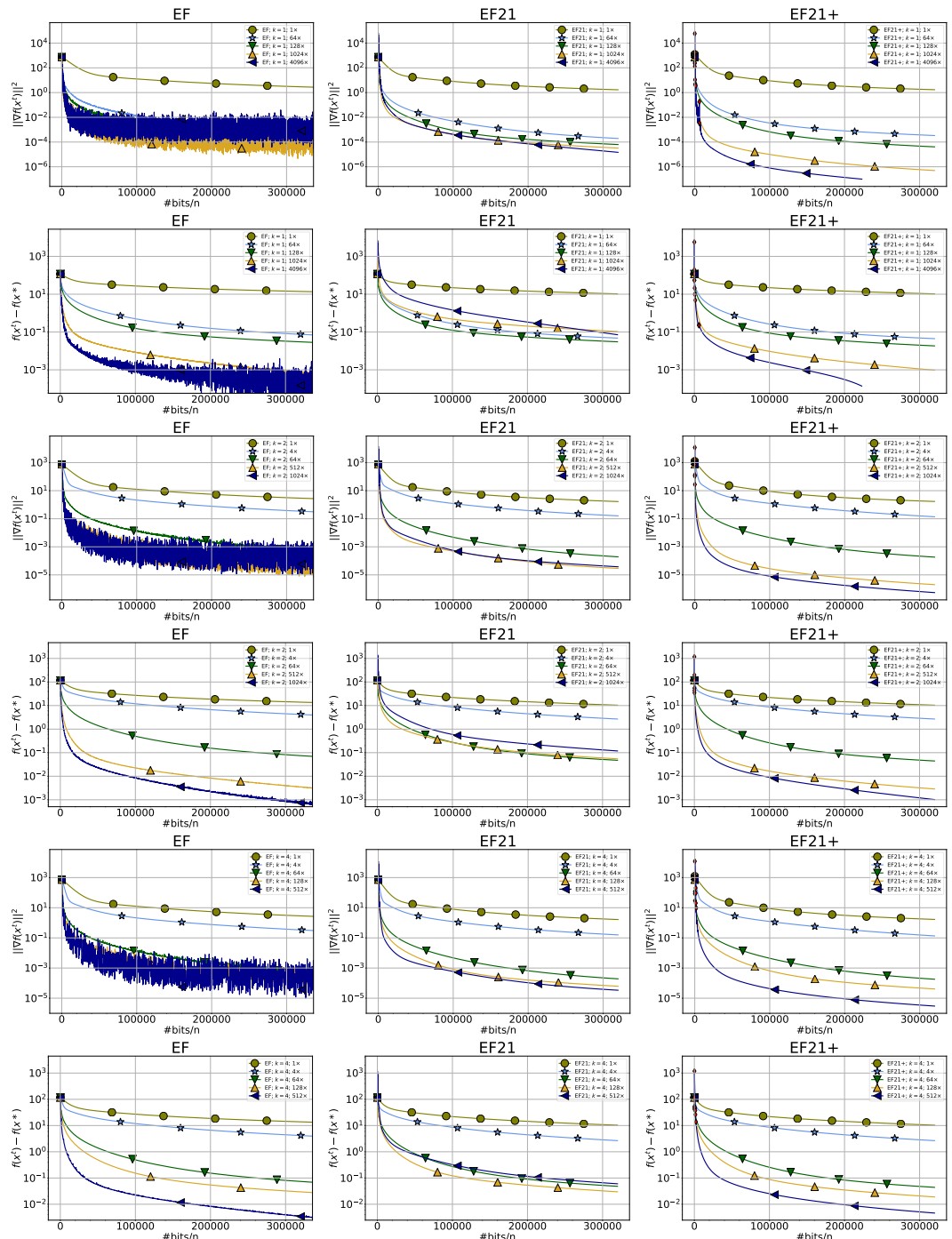

Figure 10: The performance of EF, EF21, and EF21+ with Top-$k$ compressor, and for increasing stepsizes. The dataset used: mushrooms. By $1\times, 2\times, 4\times$ (and so on) we indicate that the stepsize was set to a multiple of the largest stepsize predicted by our theory.

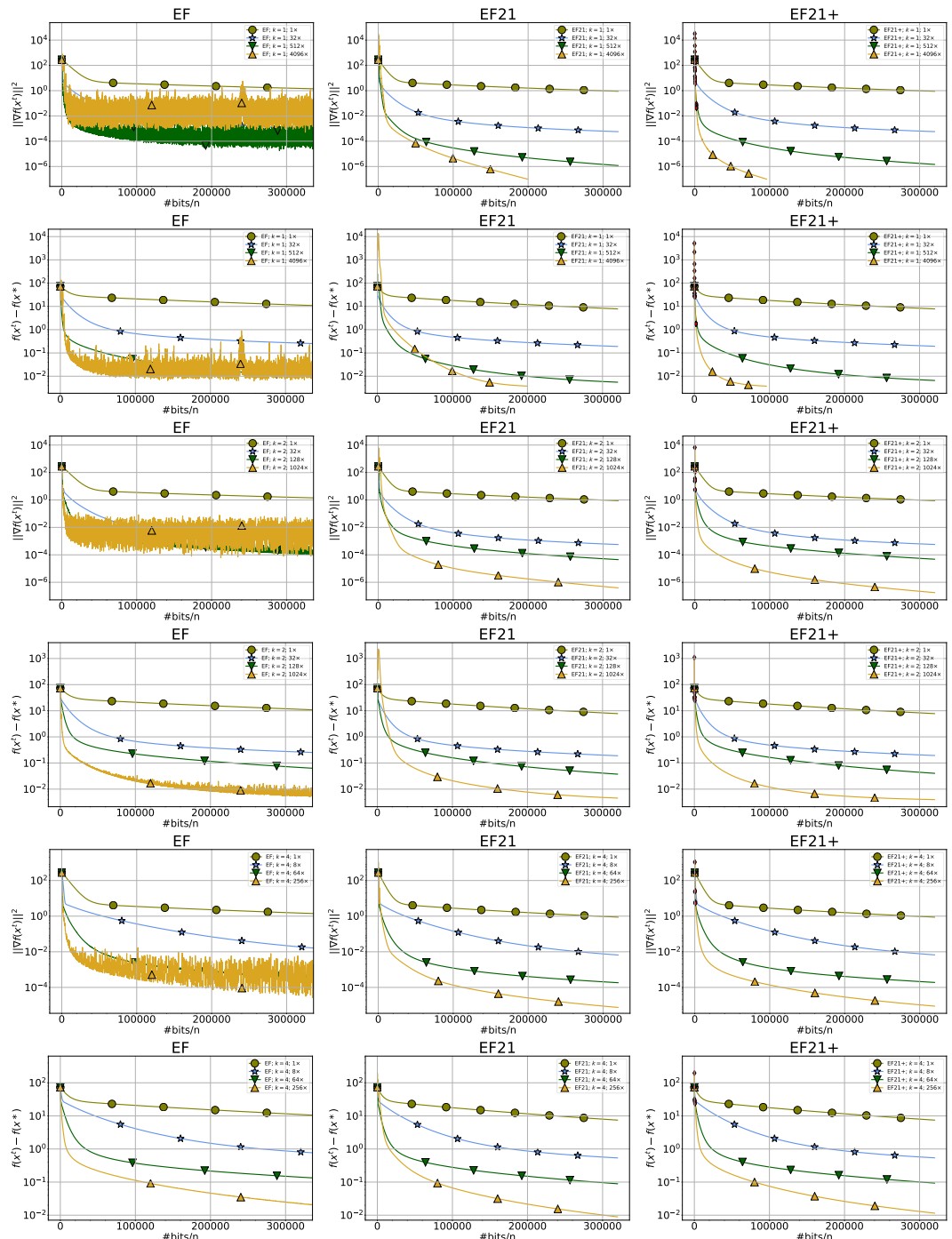

Figure 11: The performance of EF, EF21, and EF21+ with Top-$k$ compressor, and for increasing stepsizes. The dataset used: a9a. By $1\times, 2\times, 4\times$ (and so on) we indicate that the stepsize was set to a multiple of the largest stepsize predicted by our theory.

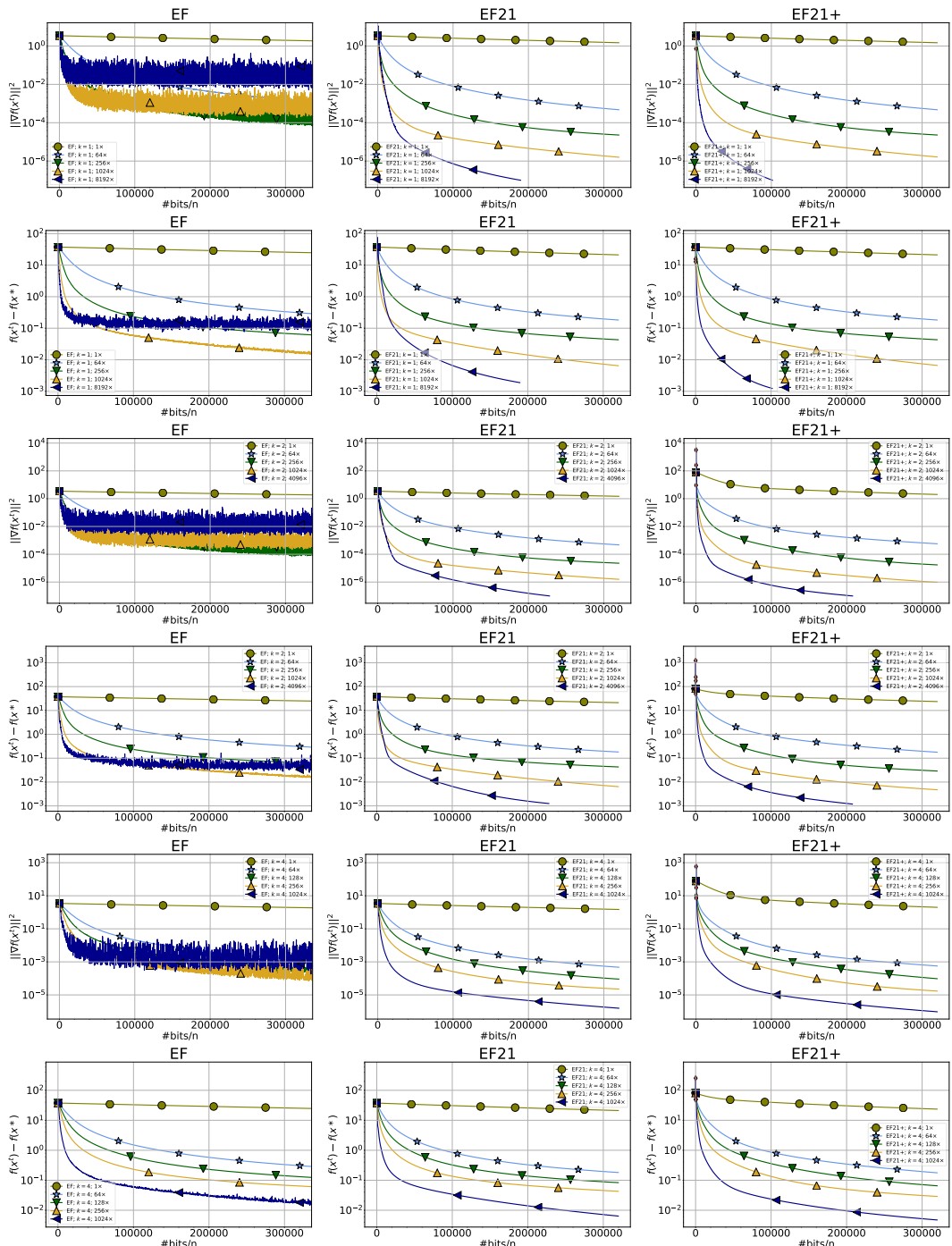

Figure 12: The performance of EF, EF21, and EF21+ with Top-$k$ compressor, and for increasing stepsizes. The dataset used: w8a. By $1\times, 2\times, 4\times$ (and so on) we indicate that the stepsize was set to a multiple of the largest stepsize predicted by our theory.

All of the figures above illustrate that in the PL setting, EF21 and EF21+ tolerate much larger stepsizes than EF, which makes them more efficient in practice. Moreover, in all experiments with large stepsizes ($512\times$–$4096\times$), EF starts oscillating, which hinders the convergence to the desired tolerance.

### A.3 Deep learning experiments

In this section, we replace full gradient $\nabla f_i(x^{k+1})$ in the algorithms EF21 and EF by its stochastic estimator (minibatch without replacement), and conduct several deep learning experiments for multi-class image classification. In particular, we compare our EF21 method to EF by running ResNet18 [He et al., 2016] and VGG11 models on the CIFAR-10 [Krizhevsky et al., 2009] dataset.

We implement the algorithms in PyTorch [Paszke et al., 2019] and run the experiments on several GPUs. We used 3 different GPU cluster node types in total within all experiments:

1. NVIDIA GeForce GTX 1080 Ti;

2. NVIDIA GeForce RTX 2080 Ti;

3. NVIDIA Tesla V100.

The dataset is split into $n = 5$ equal parts. Total train set size for CIFAR-10 is 50,000. The test set for evaluation has 10,000 data points. The train set is split into batches of size $\tau \in \{128, 1024\}$. The first four workers own equal number of batches of data, while the last worker has the rest.

#### A.3.1 Tuned stepsizes

In our first experiments, summarized in Figures 13 and 14, we fix $k \approx 0.05D$ and $\tau = 1024$ for ResNet18, and $\tau = 128$ for VGG11.[6] We tune the stepsize starting from $10^{-3}$ as a baseline, and progressively increase it by a factor of 2. In Figure 13 we compare EF, EF21, EF21+, and SGD with the best tuned stepsizes. The experiment shows that during the training, both EF and EF21 (EF21+) perform similarly with a slight improvement in the new EF21 method. Moreover, EF21 achieves better test accuracy for both NN architectures.

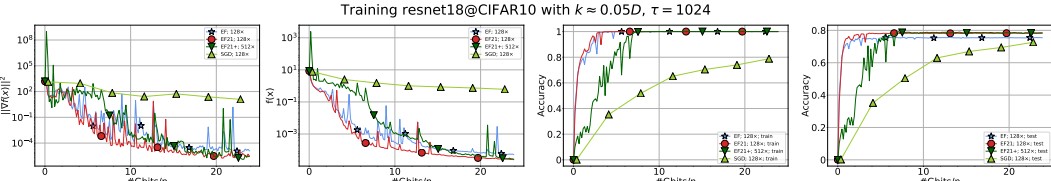

Figure 13: ResNet18 on CIFAR-10.

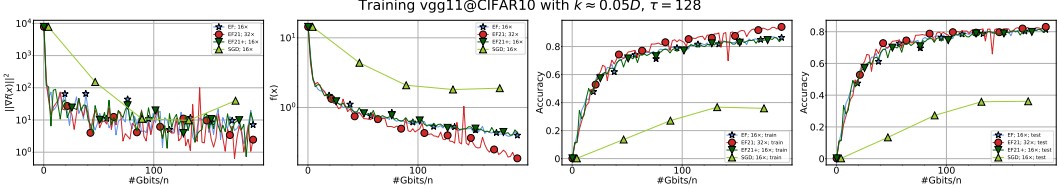

Figure 14: VGG11 on CIFAR-10.

#### A.3.2 Dependence on $k$

In this experiment, we fix the batch size $\tau = 1024$ and a medium stepsize $\gamma = 1.6 \cdot 10^{-2}$. We demonstrate that choosing smaller $k$ in the Markov compressor makes the method more communication efficient, and helps it to achieve higher test accuracy more quickly.

---

[6] $D$ is the number of model parameters. For ResNet18, $D = 11,511,784$, and for VGG11, $D = 132,863,336$.

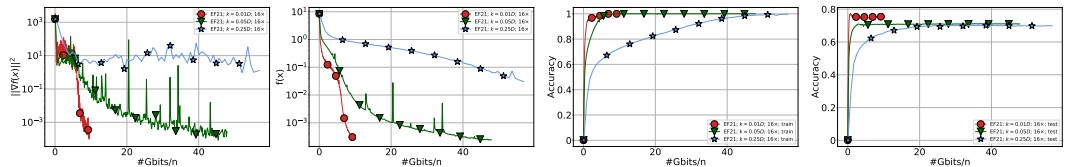

Figure 15: ResNet18 on CIFAR-10, minibatch size $\tau = 1024$.

# B  Proofs for Section 4.1: Distortion of Markov Compressor

We have made a couple statements, without proof, at the end of Section 4.1 which were not critical to the development of our results. Here we provide the justification.

**Lemma 1.** *Let $\{v^t\}_{t\geq 0}$ be any sequence of vectors in $\mathbb{R}^d$. Let*

$$D^t \overset{def}{=} \left\| \mathcal{M}(v^t) - v^t \right\|^2 \tag{20}$$

*be the distortion of the Markov compressor $\mathcal{M}$ on input $v^t$. Then*

$$\mathbb{E}\left[ D^t \right] \leq (1-\theta)^t \mathbb{E}\left[ D^0 \right] + \beta \sum_{i=0}^{t-1} (1-\theta)^i \Delta^{t-i}, \tag{21}$$

*where $\Delta^t \overset{def}{=} \left\| v^{t+1} - v^t \right\|^2$.*

*Proof.* By conditioning on $\mathcal{M}(v^t)$, we get

$$
\begin{aligned}
\mathbb{E}\left[ D^{t+1} \mid \mathcal{M}(v^t) \right] &= \mathbb{E}\left[ \left\| \mathcal{M}(v^{t+1}) - v^{t+1} \right\|^2 \mid \mathcal{M}(v^t) \right] \\
&= \mathbb{E}\left[ \left\| \mathcal{M}(v^t) + \mathcal{C}\left( v^{t+1} - \mathcal{M}(v^t) \right) - v^{t+1} \right\|^2 \mid \mathcal{M}(v^t) \right] \\
&\overset{(3)}{\leq} (1-\alpha) \left\| v^{t+1} - \mathcal{M}(v^t) \right\|^2 \\
&\leq (1-\alpha)\left[ (1+s) \left\| v^t - \mathcal{M}(v^t) \right\|^2 + (1+s^{-1}) \left\| v^{t+1} - v^t \right\|^2 \right] \\
&= (1-\theta) \left\| v^t - \mathcal{M}(v^t) \right\|^2 + \beta \Delta^t,
\end{aligned} \tag{22}
$$

where $s > 0$ is small enough so that that $1 - \theta = (1-\alpha)(1+s) < 1$, and we define $\beta = (1-\alpha)(1+s^{-1})$.

By applying the tower property, we get

$$
\begin{aligned}
\mathbb{E}\left[ D^{t+1} \right] &= \mathbb{E}\left[ \mathbb{E}\left[ D^{t+1} \mid \mathcal{M}(v^t) \right] \right] \\
&\overset{(22)}{=} (1-\theta)\mathbb{E}\left[ \left\| v^t - \mathcal{M}(v^t) \right\|^2 \right] + \beta \Delta^t \\
&\overset{(20)}{=} (1-\theta)\mathbb{E}\left[ D^t \right] + \beta \Delta^t.
\end{aligned}
$$

It remains to unroll this recurrence.

$\square$

**Corollary 1.** *Assume that $\Delta^t \leq (1-\phi)^t \Delta^0$ for all $t \geq 0$ and some $\phi > 0$. Then*

$$\lim_{t\to\infty} \mathbb{E}\left[ D^t \right] = 0.$$

*Proof.* Using Lemma 1, we get

$$\mathbb{E}\left[D^t\right] \overset{(21)}{\leq} (1-\theta)^t \mathbb{E}\left[D^0\right] + \beta \sum_{i=0}^{t-1}(1-\theta)^i \Delta^{t-i}$$

$$\leq (1-\theta)^t \mathbb{E}\left[D^0\right] + \beta \Delta^0 \sum_{i=0}^{t-1}(1-\theta)^i (1-\phi)^{t-i}$$

$$\leq (1-\theta)^t \mathbb{E}\left[D^0\right] + \beta \Delta^0 \sum_{i=0}^{t-1}(1-\min\{\theta,\phi\})^t$$

$$= (1-\theta)^t \mathbb{E}\left[D^0\right] + t(1-\min\{\theta,\phi\})^t \beta \Delta^0.$$

Clearly, the right hand side converges to 0 as $t \to \infty$. $\qquad\square$

# C   Proofs for Section 4.4: Theorem 1

In this section we describe the original error feedback (EF) method, restate the EF–EF21 equivalence theorem (Theorem 1), and prove it.

## C.1   The original error feedback method

The EF method is described in Algorithm 3. We write it in a slightly non-conventional but equivalent form which facilitates comparison with EF21.

EF works as follows. In iteration $t = 0$, each node $i$ computes its local gradient $\nabla f_i(x^0)$, and "would like" to communicate the vector $\gamma \nabla f_i(x^0)$ to the master, which is supposed to perform an aggregation of these vectors via averaging, and perform the gradient-type step

$$x^1 = x^0 - \frac{1}{n}\sum_{i=1}^{n} \gamma \nabla f_i(x^0).$$

This, in fact, is one step of gradient descent. However, the vector $\gamma \nabla f_i(x^0)$ is hard to communicate. For this reason, this vector needs to be compressed, and the compressed version needs to be communicated instead. This would lead to the iteration

$$x^1 = x^0 - \frac{1}{n}\sum_{i=1}^{n} w_i^0, \qquad \text{where} \qquad w_i^0 = \mathcal{C}(\gamma \nabla f_i(x^0)),$$

which is a variant[7] of distributed CGD (DCGD).

However, it is well known that DCGD may diverge. The key idea of error feedback is to compute the *error*

$$e_i^1 = \gamma \nabla f_i(x^0) - \mathcal{C}(\gamma \nabla f_i(x^0)) = \gamma \nabla f_i(x^0) - w_i^0,$$

which is the difference between the message $\gamma \nabla f_i(x^0)$ we *want* to communicate, and the compressed message $w_i^0$ we *actually* communicate. This error is then *added* to the message $\gamma \nabla f_i(x^1)$ we would normally want to communicate in the *next* iteration, providing feedback/compensation for the error incurred. That is, in the next iteration, node $i$ communicates the compressed vector

$$w_i^1 = \mathcal{C}(e_i^1 + \gamma \nabla f_i(x^1))$$

instead. Note that since in iteration 1 we wanted to communicate the vector $e_i^1 + \gamma \nabla f_i(x^1)$, the error in the next iteration becomes

$$e_i^2 = e_i^1 + \gamma \nabla f_i(x^1) - \mathcal{C}(e_i^1 + \gamma \nabla f_i(x^1)) = e_i^1 + \gamma \nabla f_i(x^1) - w_i^1.$$

This process is repeated, leading to Algorithm 3.

---

[7]This method *is* DCGD if $\mathcal{C}$ is positively homogeneous, i.e., of $\mathcal{C}(\gamma g) = \gamma \mathcal{C}(g)$ for every $\gamma > 0$ and $g \in \mathbb{R}^d$. However, even without positive homogeneity, this variant has the same theoretical properties as standard DCGD.

---

**Algorithm 3** EF (Original error feedback)

---

1: Each node $i = 1, \ldots, n$ sets the initial error to zero: $e_i^0 = 0$
2: Each node $i = 1, \ldots, n$ computes $w_i^0 = \mathcal{C}(\gamma \nabla f_i(x^0))$ and sends this to the master
3: **for** $t = 0, 1, 2, \ldots, T - 1$ **do**
4:     Master computes $x^{t+1} = x^t - \frac{1}{n} \sum_{i=1}^n w_i^t$
5:     **for all nodes** $i = 1, \ldots, n$ **in parallel do**
6:         Compute current error: $e_i^{t+1} = e_i^t + \gamma \nabla f_i(x^t) - w_i^t$
7:         Compute new local gradient $\nabla f_i(x^{t+1})$
8:         Compute error-compensated (stepsize-scaled) gradient $w_i^{t+1} = \mathcal{C}(e_i^{t+1} + \gamma \nabla f_i(x^{t+1}))$
9:         Send $w_i^{t+1}$ to the master
10:     **end for**
11: **end for**

---

### C.2 The proof of Theorem 1

**Theorem 1.** *Assume that $\mathcal{C}$ is deterministic, positive homogeneous and additive. Then* EF *(Algorithm 3) and* EF21 *(Algorithm 2) produce the same sequences of iterates $\{x^t\}_{t \geq 0}$.*

*Proof.* To prove this result, it suffices to show that $w_i^t = \gamma g_i^t$ for all $t \geq 0$. We perform this proof by induction.

**Base case** ($t = 0$): Recall that $w_i^0 = \mathcal{C}(\gamma \nabla f_i(x^0))$ and $g_i^0 = \mathcal{C}(\nabla f_i(x^0))$. By positive homogeneity of $\mathcal{C}$, we have

$$w_i^0 = \mathcal{C}(\gamma \nabla f_i(x^0)) = \gamma \mathcal{C}(\nabla f_i(x^0)) = \gamma g_i^0.$$

**Inductive step:** Assume that $w_i^t = \gamma g_i^t$ holds for some $t \geq 0$. Note that in view of how EF operates, we have

$$w_i^{t+1} = \mathcal{C}\left(e_i^{t+1} + \gamma \nabla f_i(x^{t+1})\right) = \mathcal{C}\left(e_i^t + \gamma \nabla f_i(x^t) - w_i^t + \gamma \nabla f_i(x^{t+1})\right).$$

Since we assume that $\mathcal{C}$ is additive, and because $w_i^t = \mathcal{C}(e_i^t + \gamma \nabla f_i(x^t))$, we can write

$$
\begin{aligned}
w_i^{t+1} &= \mathcal{C}\left(e_i^t + \gamma \nabla f_i(x^t)\right) + \mathcal{C}\left(\gamma \nabla f_i(x^{t+1}) - w_i^t\right) \\
&= w_i^t + \mathcal{C}\left(\gamma \nabla f_i(x^{t+1}) - w_i^t\right).
\end{aligned}
$$

Finally, using positive homogeneity, our inductive hypothesis, and the way $g_i^t$ is updated in EF21, we can write

$$
\begin{aligned}
w_i^{t+1} &= \gamma \left(\frac{1}{\gamma} w_i^t + \mathcal{C}\left(\nabla f_i(x^{t+1}) - \frac{1}{\gamma} w_i^t\right)\right) \\
&= \gamma \left(g_i^t + \mathcal{C}\left(\nabla f_i(x^{t+1}) - g_i^t\right)\right) \\
&= \gamma g_i^{t+1},
\end{aligned}
$$

which concludes our proof. $\qquad\square$

## D Four Lemmas Needed in the Proofs of Theorems 2 and 3

We first state several auxiliary results we need for the proofs of our main theorems.

### D.1 Compression distortion bound

The following lemma play a key role in our analysis. It characterizes the change of the distortion imparted by the Markov compressor in a single iteration.

**Lemma 2.** *Let $\mathcal{C} \in \mathbb{B}(\alpha)$ for $0 < \alpha \leq 1$. Define $G_i^t \overset{def}{=} \|g_i^t - \nabla f_i(x^t)\|^2$ and $W^t \overset{def}{=} \{g_1^t, \ldots, g_n^t, x^t, x^{t+1}\}$. For any $s > 0$ we have*

$$\mathbb{E}\left[G_i^{t+1} \mid W^t\right] \leq (1 - \theta(s))G_i^t + \beta(s)\left\|\nabla f_i(x^{t+1}) - \nabla f_i(x^t)\right\|^2, \tag{23}$$

*where*

$$\theta(s) \stackrel{def}{=} 1 - (1 - \alpha)(1 + s), \qquad and \qquad \beta(s) \stackrel{def}{=} (1 - \alpha)\left(1 + s^{-1}\right). \tag{24}$$

*Proof.*

$$
\begin{aligned}
\mathbb{E}\left[G_i^{t+1} \mid W^t\right] &= \mathbb{E}\left[\left\|g_i^{t+1} - \nabla f_i(x^{t+1})\right\|^2 \mid W^t\right] \\
&= \mathbb{E}\left[\left\|g_i^t + \mathcal{C}(\nabla f_i(x^{t+1}) - g_i^t) - \nabla f_i(x^{t+1})\right\|^2 \mid W^t\right] \\
&\stackrel{(3)}{\leq} (1 - \alpha)\left\|\nabla f_i(x^{t+1}) - g_i^t\right\|^2 \\
&\leq (1 - \alpha)(1 + s)\left\|\nabla f_i(x^t) - g_i^t\right\|^2 \\
&\quad + (1 - \alpha)\left(1 + s^{-1}\right)\left\|\nabla f_i(x^{t+1}) - \nabla f_i(x^t)\right\|^2,
\end{aligned}
$$

where the last inequality follows from Young's inequality, which states that for any $a, b \in \mathbb{R}^d$ and any $s > 0$ we have $\|a + b\|^2 \leq (1 + s)\|a\|^2 + \left(1 + s^{-1}\right)\|b\|^2$. $\qquad\square$

In particular, consider node $i$ and iteration $t$. Applying Markov compressor specific to node $i$ (let us call it $\mathcal{M}_i$) to $v_i^t = \nabla f_i(x^t)$, we get $g_i^t = \mathcal{M}_i(v_i^t)$. In the next iteration, we apply Markov compressor to the new gradient, $v_i^{t+1} = \nabla f_i(x^{t+1})$, and the compressed vector is $g_i^{t+1} = \mathcal{M}_i(v_i^{t+1})$. Note that $G_i^t$ is the distortion of Markov compressor at iteration $t$, and that (23) describes how this distortion changes from iteration $t$ to iteration $t+1$. The expectation on the left hand side is over the randomness inherent in $\mathcal{C}$ (and so, for example, if $\mathcal{C}$ is the Top-$k$ compressor, expectation is not needed).

Note that since the distortion of the Markov compressor at iteration $t$ is equal to

$$G_i^t \stackrel{def}{=} \left\|g_i^t - \nabla f_i(x^t)\right\|^2,$$

(23) says that, provided that $\theta(s) > 0$, the distortion decreases by the factor of $1 - \theta(s)$, subject to the additive error

$$\varepsilon_i^t(s) \stackrel{def}{=} \beta(s)\left\|\nabla f_i(x^{t+1}) - \nabla f_i(x^t)\right\|^2.$$

That is, (23) can be written in the form

$$\mathbb{E}\left[\left\|\mathcal{M}_i(\nabla f_i(x^{t+1})) - \nabla f_i(x^{t+1})\right\|^2 \mid W^t\right] \leq (1 - \theta(s))\left\|\mathcal{M}_i(\nabla f_i(x^t)) - \nabla f_i(x^t)\right\|^2 + \varepsilon_i^t(s).$$

Note that since our method converges, the difference $\nabla f_i(x^{t+1}) - \nabla f_i(x^t)$ decreases to zero, and hence the additive error $\varepsilon_i^t(s)$ decreases to zero, too.

Note that the distortion evolution mechanism described by Lemma 2 is fundamentally different from the distortion evolution mechanism behind the vanilla biased compressor $\mathcal{C}$. Indeed, for this compressor we instead have

$$\mathbb{E}\left[\left\|\mathcal{C}(\nabla f_i(x^{t+1})) - \nabla f_i(x^{t+1})\right\|^2 \mid W^t\right] \leq (1 - \alpha)\left\|\nabla f_i(x^{t+1})\right\|^2.$$

This inequality bounds the distortion, but does not provide a *recursion* characterizing how the distortion changes from one iteration to another.

### D.2   Optimal choice of $s$ in Lemma 2

Notice that in Lemma 2 we have some freedom in how to choose $s$. It turns out, and this will be apparent from the proofs of Theorems 2 and 3, that the optimal way of choosing $s$ is to minimize the ratio $\frac{\beta(s)}{\theta(s)}$. The next lemma characterizes the optimal choice of $s$. Note that the upper bound on $s$ is equivalent to requiring that $\theta(s) > 0$, i.e., that the first term on the right hand side in (23) results in a contraction.

**Lemma 3.** *Let $0 < \alpha < 1$ and for $s > 0$ let $\theta(s)$ and $\beta(s)$ be as in (24). Then the solution of the optimization problem*

$$\min_s \left\{ \frac{\beta(s)}{\theta(s)} \; : \; 0 < s < \frac{\alpha}{1 - \alpha} \right\} \tag{25}$$

*is given by* $s^* = \frac{1}{\sqrt{1-\alpha}} - 1$. *Furthermore,* $\theta(s^*) = 1 - \sqrt{1-\alpha}$, $\beta(s^*) = \frac{1-\alpha}{1-\sqrt{1-\alpha}}$ *and*

$$\sqrt{\frac{\beta(s^*)}{\theta(s^*)}} = \frac{1}{\sqrt{1-\alpha}} - 1 = \frac{1}{\alpha} + \frac{\sqrt{1-\alpha}}{\alpha} - 1 \leq \frac{2}{\alpha} - 1. \tag{26}$$

*Proof.* After simple algebraic manipulation, it is easy to see that

$$\frac{\beta(s)}{\theta(s)} = \left( \frac{1}{1-\alpha} - \frac{1}{(1+s)(1-\alpha)} - s \right)^{-1},$$

and hence the optimization problem (25) is equivalent to the problem

$$\min_s \left\{ \varphi(s) \overset{\text{def}}{=} \frac{1}{(1+s)(1-\alpha)} + s \ : \ 0 < s < \frac{\alpha}{1-\alpha} \right\}.$$

Note that $\varphi$ is convex, and that $\varphi(0) = \varphi(\frac{\alpha}{1-\alpha}) = \frac{1}{1-\alpha}$. Hence, the global minimum of $\varphi$ must lie in the interval $0 < s < \frac{\alpha}{1-\alpha}$. Thus, we can drop the constraints, and find the solution by looking for a stationary point (i.e., for $s^*$ satisfying $\varphi'(s^*) = 0$), which leads to $s^* = 1 - \sqrt{1-\alpha}$. The rest follows by substituting the value $s = s^*$ to the expressions for $\theta(s)$, $\beta(s)$ and $\sqrt{\frac{\beta(s)}{\theta(s)}}$. $\qquad \square$

### D.3 A descent lemma

The next lemma, due to Li et al. [2021], gives a bound on the function value after one step of a method of the type

$$x^{t+1} \overset{\text{def}}{=} x^t - \gamma g^t,$$

where $g^t \in \mathbb{R}^d$ is any vector, and $\gamma > 0$ any scalar. The only assumption we need for it to hold is for $f$ to have $L$-Lipschitz gradient.

**Lemma 4** ([Li et al., 2021]). *Suppose that function $f$ is $L$-smooth and let $x^{t+1} \overset{\text{def}}{=} x^t - \gamma g^t$, where $g^t \in \mathbb{R}^d$ is any vector, and $\gamma > 0$ any scalar. Then we have*

$$f(x^{t+1}) \leq f(x^t) - \frac{\gamma}{2} \left\| \nabla f(x^t) \right\|^2 - \left( \frac{1}{2\gamma} - \frac{L}{2} \right) \left\| x^{t+1} - x^t \right\|^2 + \frac{\gamma}{2} \left\| g^t - \nabla f(x^t) \right\|^2. \tag{27}$$

### D.4 Stepsize selection

The only purpose of our final lemma is to get an easy-to-write bound on the stepsize. We achieve this at the cost of a slightly worse theoretical result, by at most a factor of two. In particular, in the proof of our main theorems, the stepsize needs to satisfy an inequality of the type

$$a\gamma^2 + b\gamma \leq 1 \tag{28}$$

where $a, b$ are positive scalars. Instead of writing an algebraic expression for the largest $\gamma$ satisfying this inequality (let's call this optimal stepsize $\gamma^*$), we first observe that, necessarily,

$$\gamma^* \leq \min \left\{ \frac{1}{\sqrt{a}}, \frac{1}{b} \right\}.$$

Further, it is easy to verify that $\gamma^- \overset{\text{def}}{=} \frac{1}{\sqrt{a}+b}$ satisfies the quadratic inequality (28), and that $\gamma^+ \overset{\text{def}}{=} \frac{2}{\sqrt{a}+b}$ does not. So, any $0 \leq \gamma \leq \gamma^-$ satisfies (28), and the upper bound is at most a factor of 2 worse than $\gamma^*$.

We now formalize the above observations.

**Lemma 5.** *Let $a, b > 0$. If $0 \leq \gamma \leq \frac{1}{\sqrt{a}+b}$, then $a\gamma^2 + b\gamma \leq 1$. Moreover, the bound is tight up to the factor of 2 since $\frac{1}{\sqrt{a}+b} \leq \min \left\{ \frac{1}{\sqrt{a}}, \frac{1}{b} \right\} \leq \frac{2}{\sqrt{a}+b}$*

# E   Proof of Theorem 2

*Proof.* **STEP 1.** Recall that Lemma 2 says that

$$\mathbb{E}\left[\left\|g_i^{t+1} - \nabla f_i(x^{t+1})\right\|^2 \mid W^t\right] \overset{(23)}{\leq} (1-\theta)\left\|g_i^t - \nabla f_i(x^t)\right\|^2 + \beta\left\|\nabla f_i(x^{t+1}) - \nabla f_i(x^t)\right\|^2, \quad (29)$$

where $\theta = \theta(s^*)$ and $\beta = \beta(s^*)$ are given by Lemma 3. Averaging inequalities (29) over $i \in \{1,2,\ldots,n\}$ gives

$$
\begin{aligned}
\mathbb{E}\left[G^{t+1} \mid W^t\right] \quad &\overset{(14)}{=} \quad \frac{1}{n}\sum_{i=1}^{n}\mathbb{E}\left[\left\|g_i^{t+1} - \nabla f_i(x^{t+1})\right\|^2 \mid W^t\right] \\
&\overset{(29)}{\leq} \quad (1-\theta)\frac{1}{n}\sum_{i=1}^{n}\left\|g_i^t - \nabla f_i(x^t)\right\|^2 + \beta\frac{1}{n}\sum_{i=1}^{n}\left\|\nabla f_i(x^{t+1}) - \nabla f_i(x^t)\right\|^2 \\
&\overset{(14)}{=} \quad (1-\theta)\,G^t + \beta\frac{1}{n}\sum_{i=1}^{n}\left\|\nabla f_i(x^{t+1}) - \nabla f_i(x^t)\right\|^2 \\
&\leq \quad (1-\theta)\,G^t + \beta\left(\frac{1}{n}\sum_{i=1}^{n}L_i^2\right)\left\|x^{t+1} - x^t\right\|^2, \quad (30)
\end{aligned}
$$

where in the last step we have applied $L_i$-smoothness of functions $f_i$ for $i = 1, 2, \ldots, n$. Using Tower property in (30), we proceed to

$$\mathbb{E}\left[G^{t+1}\right] = \mathbb{E}\left[\mathbb{E}\left[G^{t+1} \mid W^t\right]\right] \overset{(30)}{\leq} (1-\theta)\,\mathbb{E}\left[G^t\right] + \beta\widetilde{L}^2\mathbb{E}\left[\left\|x^{t+1} - x^t\right\|^2\right]. \quad (31)$$

**STEP 2.** Next, using Lemma 4 and Jensen's inequality applied to the function $x \mapsto \|x\|^2$, we obtain the bound

$$
\begin{aligned}
f(x^{t+1}) \quad &\overset{(27)}{\leq} \quad f(x^t) - \frac{\gamma}{2}\left\|\nabla f(x^t)\right\|^2 - \left(\frac{1}{2\gamma} - \frac{L}{2}\right)\left\|x^{t+1} - x^t\right\|^2 + \frac{\gamma}{2}\left\|\frac{1}{n}\sum_{i=1}^{n}\left(g_i^t - \nabla f_i(x^t)\right)\right\|^2 \\
&\overset{(14)}{\leq} \quad f(x^t) - \frac{\gamma}{2}\left\|\nabla f(x^t)\right\|^2 - \left(\frac{1}{2\gamma} - \frac{L}{2}\right)\left\|x^{t+1} - x^t\right\|^2 + \frac{\gamma}{2}G^t. \quad (32)
\end{aligned}
$$

Subtracting $f^{\text{inf}}$ from both sides of (32) and taking expectation, we get

$$
\begin{aligned}
\mathbb{E}\left[f(x^{t+1}) - f^{\text{inf}}\right] \quad \leq \quad &\mathbb{E}\left[f(x^t) - f^{\text{inf}}\right] - \frac{\gamma}{2}\mathbb{E}\left[\left\|\nabla f(x^t)\right\|^2\right] \\
&-\left(\frac{1}{2\gamma} - \frac{L}{2}\right)\mathbb{E}\left[\left\|x^{t+1} - x^t\right\|^2\right] + \frac{\gamma}{2}\mathbb{E}\left[G^t\right]. \quad (33)
\end{aligned}
$$

**COMBINING STEP 1 AND STEP 2.** Let $\delta^t \overset{\text{def}}{=} \mathbb{E}\left[f(x^t) - f^{\text{inf}}\right]$, $s^t \overset{\text{def}}{=} \mathbb{E}\left[G^t\right]$ and $r^t \overset{\text{def}}{=} \mathbb{E}\left[\left\|x^{t+1} - x^t\right\|^2\right]$. Then by adding (33) with a $\frac{\gamma}{2\theta}$ multiple of (31) we obtain

$$
\begin{aligned}
\delta^{t+1} + \frac{\gamma}{2\theta}s^{t+1} \quad &\leq \quad \delta^t - \frac{\gamma}{2}\left\|\nabla f(x^t)\right\|^2 - \left(\frac{1}{2\gamma} - \frac{L}{2}\right)r^t + \frac{\gamma}{2}s^t + \frac{\gamma}{2\theta}\left(\beta\widetilde{L}^2 r^t + (1-\theta)s^t\right) \\
&= \quad \delta^t + \frac{\gamma}{2\theta}s^t - \frac{\gamma}{2}\left\|\nabla f(x^t)\right\|^2 - \left(\frac{1}{2\gamma} - \frac{L}{2} - \frac{\gamma}{2\theta}\beta\widetilde{L}^2\right)r^t \\
&\leq \quad \delta^t + \frac{\gamma}{2\theta}s^t - \frac{\gamma}{2}\left\|\nabla f(x^t)\right\|^2.
\end{aligned}
$$

The last inequality follows from the bound $\gamma^2\frac{\beta\widetilde{L}^2}{\theta} + L\gamma \leq 1$, which holds because of Lemma 5 and our assumption on the stepsize. By summing up inequalities for $t = 0, \ldots, T-1$, we get

$$0 \leq \delta^T + \frac{\gamma}{2\theta}s^T \leq \delta^0 + \frac{\gamma}{2\theta}s^0 - \frac{\gamma}{2}\sum_{t=0}^{T-1}\mathbb{E}\left[\left\|\nabla f(x^t)\right\|^2\right].$$

Multiplying both sides by $\frac{2}{\gamma T}$, after rearranging we get

$$\sum_{t=0}^{T-1} \frac{1}{T} \mathbb{E}\left[\left\|\nabla f(x^t)\right\|^2\right] \leq \frac{2\delta^0}{\gamma T} + \frac{s^0}{\theta T}.$$

It remains to notice that the left hand side can be interpreted as $\mathbb{E}\left[\left\|\nabla f(\hat{x}^T)\right\|^2\right]$, where $\hat{x}^T$ is chosen from $x^0, x^1, \ldots, x^{T-1}$ uniformly at random. $\qquad\square$

## F   Proof of Theorem 3

*Proof.* We proceed as in the previous proof, but use the PL inequality and subtract $f(x^\star)$ from both sides of (32) to get

$$\mathbb{E}\left[f(x^{t+1}) - f(x^\star)\right] \overset{(32)}{\leq} \mathbb{E}\left[f(x^t) - f(x^\star)\right] - \frac{\gamma}{2}\left\|\nabla f(x^t)\right\|^2 - \left(\frac{1}{2\gamma} - \frac{L}{2}\right)\left\|x^{t+1} - x^t\right\|^2 + \frac{\gamma}{2}G^t$$

$$\leq (1 - \gamma\mu)\mathbb{E}\left[f(x^t) - f(x^\star)\right] - \left(\frac{1}{2\gamma} - \frac{L}{2}\right)\left\|x^{t+1} - x^t\right\|^2 + \frac{\gamma}{2}G^t.$$

Let $\delta^t \overset{\text{def}}{=} \mathbb{E}\left[f(x^t) - f(x^\star)\right]$, $s^t \overset{\text{def}}{=} \mathbb{E}\left[G^t\right]$ and $r^t \overset{\text{def}}{=} \mathbb{E}\left[\left\|x^{t+1} - x^t\right\|^2\right]$. Then by adding the above inequality with a $\frac{\gamma}{\theta}$ multiple of (31), we obtain

$$\delta^{t+1} + \frac{\gamma}{\theta}s^{t+1} \leq (1 - \gamma\mu)\delta^t - \left(\frac{1}{2\gamma} - \frac{L}{2}\right)r^t + \frac{\gamma}{2}s^t + \frac{\gamma}{\theta}\left((1 - \theta)s^t + \beta\widetilde{L}^2 r^t\right)$$

$$= (1 - \gamma\mu)\delta^t + \frac{\gamma}{\theta}\left(1 - \frac{\theta}{2}\right)s^t - \left(\frac{1}{2\gamma} - \frac{L}{2} - \frac{\beta\widetilde{L}^2\gamma}{\theta}\right)r^t.$$

Note that our assumption on the stepsize implies that $1 - \frac{\theta}{2} \leq 1 - \gamma\mu$ and $\frac{1}{2\gamma} - \frac{L}{2} - \frac{\beta\widetilde{L}^2\gamma}{\theta} \geq 0$. The last inequality follows from the bound $\gamma^2 \frac{2\beta\widetilde{L}^2}{\theta} + \gamma L \leq 1$, which holds because of Lemma 5 and our assumption on the stepsize. Thus,

$$\delta^{t+1} + \frac{\gamma}{\theta}s^{t+1} \leq (1 - \gamma\mu)\left(\delta^t + \frac{\gamma}{\theta}s^t\right).$$

It remains to unroll the recurrence. $\qquad\square$

## G   `EF21+`: The Algorithm and its Analysis

### G.1   The `EF21+` Algorithm

In this section we formally present the `EF21+` algorithm (see Algorithm 4), and show that Theorems 2 and 3 still apply.

### G.2   Analysis of `EF21+`

It is easy to see that both Theorem 2 and Theorem 3 apply for `EF21+` as well, under the additional assumption that $\mathcal{C}$ is deterministic, such as Top-$k$. Note that the properties of $\mathcal{C}$ appear in the proofs only through Lemma 2, which in the language of Algorithm 4 says that

$$\mathbb{E}\left[M_i^{t+1} \mid W^t\right] \leq (1 - \theta)G_i^t + \beta\left\|\nabla f_i(x^{t+1}) - \nabla f_i(x^t)\right\|^2,$$

where $G_i^t = \left\|g_i^t - \nabla f_i(x^t)\right\|^2$. On the other hand, due to Step 8 in Algorithm 4, we know that

$$G_i^{t+1} \leq \min\{B_i^{t+1}, M_i^{t+1}\} \leq M_i^{t+1}.$$

Now, due to the assumption that $\mathcal{C}$ is a deterministic compressor, we have $\mathbb{E}\left[G_i^{t+1} \mid W^t\right] \leq G_i^{t+1}$. By stringing these three inequalities together, we arrive at

$$\mathbb{E}\left[G_i^{t+1} \mid W^t\right] \leq (1 - \theta)G_i^t + \beta\left\|\nabla f_i(x^{t+1}) - \nabla f_i(x^t)\right\|^2,$$

and this inequality can be used in the proofs instead. The rest of the proof is identical.

---

**Algorithm 4** EF21+ (Multiple nodes)

---

1: **Input:** starting point $x^0 \in \mathbb{R}^d$; $g_i^0 = \mathcal{C}(\nabla f_i(x^0))$ for $i = 1, \ldots, n$ (known by nodes and the master); learning rate $\gamma > 0$; $g^0 = \frac{1}{n} \sum_{i=1}^n g_i^0$ (known by master)
2: **for** $t = 0, 1, 2, \ldots, T - 1$ **do**
3:     Master computes $x^{t+1} = x^t - \gamma g^t$ and broadcasts $x^{t+1}$ to all nodes
4:     **for all nodes** $i = 1, \ldots, n$ **in parallel do**
5:         Compute gradient compressed by biased compressor $b_i^{t+1} = \mathcal{C}(\nabla f_i(x^{t+1}))$
6:         Compute gradient compressed my Markov compressor $m_i^{t+1} = g_i^t + \mathcal{C}(\nabla f_i(x^{t+1}) - g_i^t)$
7:         Compute distortions: $B_i^{t+1} = \left\| b_i^{t+1} - \nabla f_i(x^{t+1}) \right\|^2$; $M_i^{t+1} = \left\| m_i^{t+1} - \nabla f_i(x^{t+1}) \right\|^2$
8:         Set $g_i^{t+1} = \begin{cases} m_i^{t+1} & \text{if} \quad M_i^{t+1} \leq B_i^{t+1} \\ b_i^{t+1} & \text{if} \quad M_i^{t+1} > B_i^{t+1} \end{cases}$
9:     **end for**
10:     Master computes $g^{t+1} = \frac{1}{n} \sum_{i=1}^n g_i^{t+1}$
11: **end for**

---

## H   Dealing with Stochastic Gradients (Details for Section 4.7)

We now describe a natural extension of EF21 to the setting where full gradient computations are replaced by stochastic gradient estimators, i.e., we use a random vector

$$\hat{g}_i^t \approx \nabla f_i(x^t)$$

instead of $\nabla f_i(x^t)$. This simple change leads to Algorithm 5, where we highlight in red the parts that differ from the exact/full gradient version of EF21.

---

**Algorithm 5** EF21 (Multiple nodes + Stochastic regime)

---

1: **Input:** starting point $x^0 \in \mathbb{R}^d$; $g_i^0 = \mathcal{C}(\hat{g}_i^0)$, where $\hat{g}_i^0 \approx \nabla f_i(x^0)$ for $i = 1, \ldots, n$ (known by nodes and the master); learning rate $\gamma > 0$; $g^0 = \frac{1}{n} \sum_{i=1}^n g_i^0$ (known by master)
2: **for** $t = 0, 1, 2, \ldots, T - 1$ **do**
3:     Master computes $x^{t+1} = x^t - \gamma g^t$ and broadcasts $x^{t+1}$ to all nodes
4:     **for all nodes** $i = 1, \ldots, n$ **in parallel do**
5:         Compute a stochastic gradient $\hat{g}_i^{t+1} \approx \nabla f_i(x^{t+1})$
6:         Compress $c_i^t = \mathcal{C}(\hat{g}_i^{t+1} - g_i^t)$ and send $c_i^t$ to the master
7:         Update local state $g_i^{t+1} = g_i^t + \mathcal{C}(\hat{g}_i^{t+1} - g_i^t)$
8:     **end for**
9:     Master computes $g^{t+1} = \frac{1}{n} \sum_{i=1}^n g_i^{t+1}$ via $g^{t+1} = g^t + \frac{1}{n} \sum_{i=1}^n c_i^t$
10: **end for**

---

An analysis of this extension/generalization can be done in a similar manner. The key change is the replacement of Lemma 2 in the proofs of the two complexity theorems, and then accounting for this change in the proof. However, this is easy to do. We now describe what Lemma 2 should be replaced with.

We first start with a technical lemma.

**Lemma 6.** *Let $\mathcal{C} \in \mathbb{B}(\alpha)$, and let $\xi \in \mathbb{R}^d$ be a random vector independent of $\mathcal{C}$, with zero mean and variance bounded as $\mathbb{E}\left[ \|\xi\|^2 \right] \leq \sigma^2$. Then for any $s > 0$, we have*

$$\mathbb{E}\left[ \|\mathcal{C}(x + \xi) - x\|^2 \right] \leq (1 - \alpha)(1 + s) \|x\|^2 + \left( (1 - \alpha)(1 + s) + 1 + s^{-1} \right) \sigma^2, \qquad \forall x \in \mathbb{R}^d.$$

*Proof.* First, due to Young's inequality, for any $s > 0$ we have

$$\|\mathcal{C}(x + \xi) - x\|^2 \leq (1 + t) \|\mathcal{C}(x + \xi) - (x + \xi)\|^2 + (1 + s^{-1}) \|\xi\|^2. \tag{34}$$

By taking conditional expectation, we get

$$
\mathbb{E}\left[\left\|\mathcal{C}(x+\xi)-x\right\|^2 \mid \xi\right] \overset{(34)}{\leq} (1+s)\mathbb{E}\left[\left\|\mathcal{C}(x+\xi)-(x+\xi)\right\|^2 \mid \xi\right] + (1+s^{-1})\left\|\xi\right\|^2
$$

$$
\begin{aligned}
&\overset{(3)}{\leq} (1+s)(1-\alpha)\left\|x+\xi\right\|^2 + (1+s^{-1})\left\|\xi\right\|^2 \\
&= (1-\alpha)(1+s)\left\|x\right\|^2 + 2(1-\alpha)(1+s)\langle x,\xi\rangle \\
&\quad + \left((1-\alpha)(1+s)+1+s^{-1}\right)\left\|\xi\right\|^2.
\end{aligned} \tag{35}
$$

Taking expectation again, applying the tower property, and using the fact that $\mathbb{E}[\xi] = 0$ and $\mathbb{E}\left[\left\|\xi\right\|^2\right] \leq \sigma^2$, we finally get

$$
\begin{aligned}
\mathbb{E}\left[\left\|\mathcal{C}(x+\xi)-x\right\|^2\right] &= \mathbb{E}\left[\mathbb{E}\left[\left\|\mathcal{C}(x+\xi)-x\right\|^2 \mid \xi\right]\right] \\
&\overset{(35)}{\leq} (1-\alpha)(1+s)\left\|x\right\|^2 + \left((1-\alpha)(1+s)+1+s^{-1}\right)\sigma^2.
\end{aligned}
$$

$\square$

We will choose $s < \frac{\alpha}{1-\alpha}$, so that $1-\hat{\alpha} \overset{\text{def}}{=} (1-\alpha)(1+s) < 1$. The above lemma postulates that for $\mathcal{C} \in \mathbb{B}(\alpha)$, and under certain assumptions on the noise $\xi$, there exist constants $\hat{\alpha} > 0$ and $\hat{\sigma} > 0$ such that

$$
\mathbb{E}\left[\left\|\mathcal{C}(x+\xi)-x\right\|^2\right] \leq (1-\hat{\alpha})\left\|x\right\|^2 + \hat{\sigma}^2, \qquad \forall x \in \mathbb{R}^d. \tag{36}
$$

We will elevate this inequality into an assumption because the particular values for $\hat{\alpha}$ and $\hat{\sigma}$ given by the lemma will not be tight for every compressor $\mathcal{C}$, and we want to formulate our complexity results with as tight constants as possible.

**Assumption 3.** *Let $\mathcal{C} : \mathbb{R}^d \to \mathbb{R}^d$ be a (possibly randomized) mapping and let $\xi \in \mathbb{R}^d$ be a random vector independent of $\mathcal{C}$. We assume that there exist constants $\hat{\alpha} > 0$ and $\hat{\sigma} > 0$ such that (36) holds for all $x \in \mathbb{R}^d$.*

We now present an analogue of Lemma 2 in the stochastic regime.

**Lemma 7.** *Consider Algorithm 5 and let the the stochastic estimator $\hat{g}_i^t$ be given by*

$$
\hat{g}_i^t = \nabla f_i(x^t) + \xi_i^t,
$$

*where $\xi_i^t$ is a random vector. Assume that for $\xi = \xi_i^t$, inequality (36) holds[8]. Let $G_i^t \overset{\text{def}}{=} \left\|g_i^t - \nabla f_i(x^t)\right\|^2$ and $W^t \overset{\text{def}}{=} \{g_1^t, \ldots, g_n^t, x^t, x^{t+1}\}$. For any $t > 0$ we have*

$$
\mathbb{E}\left[G_i^{t+1} \mid W^t\right] \leq \underbrace{(1-\hat{\alpha})(1+s)}_{1-\hat{\theta}(s)} G_i^t + \underbrace{(1-\hat{\alpha})\left(1+s^{-1}\right)}_{\hat{\beta}(s)}\left\|\nabla f_i(x^{t+1})-\nabla f_i(x^t)\right\|^2 + \hat{\sigma}^2. \tag{37}
$$

*Proof.*

$$
\begin{aligned}
\mathbb{E}\left[G_i^{t+1} \mid W^t\right] &= \mathbb{E}\left[\left\|g_i^{t+1}-\nabla f_i(x^{t+1})\right\|^2 \mid W^t\right] \\
&= \mathbb{E}\left[\left\|g_i^t + \mathcal{C}(\nabla f_i(x^{t+1})+\xi_i^{t+1}-g_i^t)-\nabla f_i(x^{t+1})\right\|^2 \mid W^t\right] \\
&\overset{(36)}{\leq} (1-\hat{\alpha})\left\|\nabla f_i(x^{t+1})-g_i^t\right\|^2 + \hat{\sigma}^2 \\
&\leq (1-\hat{\alpha})(1+s)\left\|\nabla f_i(x^t)-g_i^t\right\|^2 \\
&\quad + (1-\hat{\alpha})\left(1+s^{-1}\right)\left\|\nabla f_i(x^{t+1})-\nabla f_i(x^t)\right\|^2 + \hat{\sigma}^2.
\end{aligned}
$$

$\square$

It is straightforward to use this inequality in the proofs of Theorems 2 and 3 to establish complexity results for our stochastic variant of EF21.

---

[8]Recall that by Lemma 6, it holds if $\xi = \xi_i^t$ is a zero mean vector with variance bounded by $\sigma^2$.

# I Computation of $\sqrt{\frac{\beta(s^*)}{\theta(s^*)}}$ for some Compressors

## I.1 From unbiased to biased compressors

We start by proving the simple and very well known result about the relationship between the classes $\mathbb{U}(\omega)$ and $\mathbb{B}(\alpha)$ we mentioned in Section 2.

**Lemma 8.** *If $\mathcal{C} \in \mathbb{U}(\omega)$, then $\frac{1}{1+\omega}\mathcal{C} \in \mathbb{B}\left(\frac{1}{1+\omega}\right)$.*

*Proof.* Fix $x \in \mathbb{R}^d$. Note that for $\mathcal{C} \in \mathbb{U}(\omega)$ we have

$$\mathbb{E}\left[\mathcal{C}(x)\right] = x, \tag{38}$$

$$\mathbb{E}\left[\|\mathcal{C}(x)\|^2\right] \leq (1+\omega)\|x\|^2. \tag{39}$$

Then

$$
\begin{aligned}
\mathbb{E}\left[\left\|\frac{1}{1+\omega}\mathcal{C}(x) - x\right\|^2\right] &= \frac{1}{(1+\omega)^2}\mathbb{E}\left[\|\mathcal{C}(x)\|^2\right] - \frac{2}{1+\omega}\mathbb{E}\left[\langle\mathcal{C}(x),x\rangle\right] + \|x\|^2 \\
&\overset{(38)}{\leq} \frac{1}{(1+\omega)^2}\|x\|^2 + \frac{\omega-1}{\omega+1}\|x\|^2 \\
&\overset{(39)}{\leq} \frac{1}{(1+\omega)}\|x\|^2 + \frac{\omega-1}{\omega+1}\|x\|^2 \\
&= \frac{\omega}{1+\omega}\|x\|^2 \\
&= \left(1 - \frac{1}{1+\omega}\right)\|x\|^2.
\end{aligned}
$$

$\square$

## I.2 Top-$k$ and a scaled version of Rand-$k$

We now compute the value $\sqrt{\frac{\beta(s^*)}{\theta(s^*)}}$ appearing in pour complexity theorems for two well known compressors belonging to the class $\mathbb{B}(\alpha)$.

**Example 1.** *Let $\mathcal{C}$ be the Top-k compressor. Then $\mathcal{C} \in \mathbb{B}(\alpha)$ with $\alpha = \frac{k}{d}$ and*

$$\sqrt{\frac{\beta(s^*)}{\theta(s^*)}} = \frac{\sqrt{1 - k/d}}{1 - \sqrt{1 - k/d}}.$$

*Proof.* It is well known that $\mathcal{C} \in \mathbb{B}(\alpha)$ with $\alpha = \frac{k}{d}$ (e.g., see [Beznosikov et al., 2020]). Then according to Lemma 3, we have

$$\sqrt{\frac{\beta(s^*)}{\theta(s^*)}} = \frac{\sqrt{1-\alpha}}{1-\sqrt{1-\alpha}} = \frac{\sqrt{1 - k/d}}{1 - \sqrt{1 - k/d}}.$$

$\square$

**Example 2.** *Let $\mathcal{C} = \left(\frac{1}{1+\omega}\right)\mathcal{C}'$, where $\mathcal{C}'$ is the Rand-k compressor. Then $\mathcal{C} \in \mathbb{B}(\alpha)$ with $\alpha = \frac{k}{d}$ and*

$$\sqrt{\frac{\beta(s^*)}{\theta(s^*)}} = \frac{\sqrt{1 - k/d}}{1 - \sqrt{1 - k/d}}.$$

*Proof.* It is well known that $\mathcal{C}' \in \mathbb{B}(\omega)$ with $\omega = \frac{d}{k} - 1$ (e.g., see [Beznosikov et al., 2020]). Moreover, using the Lemma 8, we get $\left(\frac{1}{1+\omega}\right)\mathcal{C}' \in \mathbb{B}\left(\frac{k}{d}\right)$. Finally, according to Lemma 3, we have

$$\sqrt{\frac{\beta(s^*)}{\theta(s^*)}} = \frac{\sqrt{1-\alpha}}{1-\sqrt{1-\alpha}} = \frac{\sqrt{1-k/d}}{1-\sqrt{1-k/d}}.$$

$\square$