# OpenReview forum: "EF21: A New, Simpler, Theoretically Better, and Practically Faster Error Feedback"
_NeurIPS.cc/2021/Conference — NeurIPS 2021 Oral_

### Official Review · Reviewer_EY42 · 2021-06-26

**Rating:** 5
**Confidence:** 5

**Summary:**

This paper studies the Error Feedback (EF) technique in distributed optimization. The paper proposes a new version called EF21 and claims this is theoretically and practically better. A new scheme for EF compression named Markov compressor is proposed. The writing quality is great and the paper is easy to follow. However, some of the statements are flawed which substantially compromise the quality of this paper.

**Limitations And Societal Impact:**

The paper discusses a new variant on a technique in distributed training. There is, from my perspective, no serious issue or limitation that would impact society.

**Main Review:**

1. For starters, one of the main contributions of EF21 is to converge at a rate of $1/T$ instead of $1/T^{2/3}$. Here the author makes a mistake in line 133, saying Koloskova et al. [2020, Theorem 4.1] assumes a bound on the full gradient on worker i. However, the actual assumption in that paper was a bound on the *stochastic* gradient on worker i. These two are notably different as the latter incorporates the sampling noise. If the sample complexity is not considered (which seems to be the case of EF21), it's not surprising that a $1/T$ rate is achieved (this is like saying GD improves upon SGD). In fact, if we remove the sampling noise in Koloskova et al. [2020, Theorem 4.1], a $1/T$ rate is straightforward to obtain. The paper claims in the abstract EF21 "beats" the previous bound of $1/T^{2/3}$, which is confusing. Also Koloskova et al. [2020] works in the decentralized case, which is more challenging than the setting in this paper.

2. Could you provide a data-parallel setting where we need communication compression but no sampling? I cannot think of any since communication overhead is usually observed in deep learning tasks and sampling is a must rather than an option. Having said that, the application of EF21 is very limited. The authors provide empirical results on deep learning tasks in the appendix, but mini-batching is used there, which is not aligned with the theory or logistic regression task. Also the data heterogeneity issue is confusing: In data center training where we are able to shuffle the dataset, we can usually reduce the intra-worker variance. Heterogeneous data usually takes place in the federated learning, where the execution of EF21 shown in the paper clearly does not cover.

3. Based on 1 and 2, the main improvement of Markov compressor upon EF is data heterogeneity where $||\nabla f_i||$ can be large when the algorithm converges. However, the only case the Markov compressor can address this seems to be in non-stochastic case. If we use stochastic gradient in the Markov compressor, the error bound does not approach zero either. How EF21 handles this issue remains unclear.

4. The authors conduct different experiments  but the results are not convincing. First, # Gbits/n is used as X-axis for all figures. However, for compression algorithms the convergence speed with respect to iterations or time are also important since when communication is reduced, the compression can become a new overhead. If the paper claims EF21 is practically better, other X-axis should be provided. Second, some of the plots for baselines are omitted for no reason, which is serious. Specifically, in the Resnet18 on CIFAR-10 experiment, EF and EF21 seems to perform similar under 128X, however only EF21 is shown under 512X while EF at 512X is not shown. Note that validation performance is not informative here, since the design of EF21 does not consider generalization.

Clarification on my review after reading the author response
===
While I appreciate the author answering my questions, but it appears some of my concerns were misunderstood. Specifically:

**On the problem with improved rates**

I did not state anywhere in my review that $G=0$, as author claimed. Let me rephrase: this is the problem with the baseline comparison. Take Koloskova et al. [2020] as example (other baselines such as Tang et al. [2020] has similar issue), in their Theorem 4.1, the convergence bound is shown as
$O\left(\frac{\bar{\sigma}}{\sqrt{T}} + \frac{G^{2/3}}{T^{2/3}} + \frac{1}{T}\right)$, where $\bar{\sigma}^2$ and $G$ are assumed to be the bound on the variance and magnitude of the *stochastic* gradients, respectively. Now, since EF21's rate is shown in non-stochastic case, for comparison, the authors convert the baseline bound as follows: drop the first term (which makes sense since $\bar{\sigma}=0$); and keep the second term. That's where my confusion comes from, as $G$ is clearly correlated with the stochastic gradient in their analysis, so how do you justify the removal of term $\frac{1}{T^{2/3}}$ in EF21 is due to EF21 being better algorithm/analysis, but not because the fact that two settings are different so that the conversion of baseline results is loosen? This happens because the authors converted a result from a more general setting to a special case, which could implicitly worsen the baseline result. In fact, there is no obvious way to detach the stochastic noise from $\frac{G^{2/3}}{T^{2/3}}$ term without reproving the baselines in the non-stochastic case, so I don't fully understand how is it an improved rate.

To make this even clearer, consider this example (not relevant to the paper, just for illustration): suppose we prove algorithm A having rate $O(1/T)$ on smooth functions while algorithm B having rate $O(\rho^T)$ on smooth and strongly convex functions. Clearly, algorithm A is analyzed in a more general setting (analogous to Koloskova et al. [2020]). Can we say B is better than A by directly converting these bound in any way? Apparently not. In fact, the only way to make conclusion on rate comparison is to reprove A or B in the other setting.

(**In response to author's follow-up question on why Gower et al (2019) can do the conversion correctly**): I thank the authors for bringing up this paper as it is a perfect example for illustrating the problem. In a nutshell, you can convert a result from a general case to a special case by setting zero to variables, but not in every case. It depends on how the baseline result is shown. Gower et al (2019) does an analysis for SGD, take their Theorem 3.1 as example, they show the convergence rate by $\mathbb{E}||x^k-x^*||^2 \leq (1-\gamma\mu)^k\mathbb{E}||x^0-x^*||^2 + 2\gamma\sigma^2/\mu$ in their Equation (10). Now this result can be converted to GD with valid conversion since the term relates to stochastic sampling is the second one, and they can easily convert by setting $\sigma=0$. However, in this paper the conversion is problematic (I acknowledge setting $\sigma=0$ is correct in my review, it's the $G$ term that has problem) because you cannot detach the effect of stochastic gradient from $G$. Put it another way, if the baseline result looks like $O(\sigma/\sqrt{T}+{1}/{T^{2/3}} + 1/T)$, then your conversion would be correct since now only the first term relates to the sampling and you can easily convert the result by setting $\sigma=0$ but that's not the case here.

**On the problem with application**

In my original review, I raised the question of naming an application setting where we need communication compression but no sampling, to which EF21 can be applicable. The author states in their response that one application of EF21 is federated learning, which seems a little off to me. As we know, federated learning usually involves a large number of nodes, so a typical algorithm (such as FedAvg, SCAFFOLD) requires client sampling. However, in the multiple node version of EF21, it requires all the nodes to participate in each iteration and need to be synchronized at the end of each iteration. So if I understand it correctly, EF21 can be used in federated learning only under the following two conditions:

(1) The number of nodes cannot be too large so that global synchronization is possible and negligible.

(2) The number of data samples on each node (client) cannot be too large so that full gradient computation is not an overhead.

I don't really understand how this is applicable, please correct me if anything in my statement is incorrect.

**On the problem of extension to stochastic case**

Maybe I did not state it well in the original review. My claim is that stochastic gradient actually breaks the intuition of EF21 which makes it not easy to extend. Here's why: the intuition of EF21 is illustrated in line 271: $\mathbb{E}||\mathcal{M}(v^t)-v^t||^2\rightarrow 0$, while this is true for full gradient because as the algorithm converges, the magnitude of full gradient will approach zero. However, it's not clear to me why it still holds for stochastic gradient, especially the case where all the data samples have different local minimas (not quite the heterogeneity as mentioned in the paper). Specifically, could you explain why such quantity still approaches zero if $v^t$ contains sampling noise?

The authors state in their response that error feedback is orthogonal to stochastic approximation, and the latter should be handled by other variance reduction algorithms like SVRG, I'm confused. Because the baseline algorithms like DoubleSqueeze, Q-local-Sparse clearly works with stochastic approximation without involving variance reduction parts.

The author pointed out a stochastic analysis is included in the appendix I, however, no rate is shown in that analysis and that analysis ends with "It is straightforward...to establish complexity results for our stochastic variant of EF21 ." If the authors believe the statement of my original review "$O(1/T)$ rate is straightforward to obtain in baseline" is unjustified because "such a rate is not included in their paper", I politely ask the authors to apply the same standard to the paper, and avoid using such statement to justify your results. Because, the rate of EF21+ too, is not shown in the paper or appendix. As pointed out by authors in their response: "Simply claiming this does not make it true". Please correct me if I'm wrong.

**Summary**

I believe all my concerns are specific on the technical issues. The idea of Markov compressor is interesting, so I've decided to raise my score slightly to give some more credit. However, at this point, I still don't think this paper meets the acceptance bar of NeurIPS as the issues I raised are serious and closely related to the main contribution of this paper, unless the authors can prove I misunderstood them all.





**Time Spent Reviewing:**

4

---

> ### Author Response · Authors · 2021-08-04
> **Response to Issue 1: improved rate**
>
> a) We did not make a mistake, you made a mistake. In Assumption 2 on page 4 in Koloskova et al (https://arxiv.org/pdf/1907.09356.pdf), the following assumption is made: $$ \mathbb{E}_{\zeta_i} ||  \nabla f_i(x,\zeta_i) ||^2 \leq G^2, \quad \forall x\in \mathbb{R}^d, \quad \forall i.$$
>
> Specialized to the deterministic setting which we consider, this assumption reads as follows:
> $$||  \nabla f_i(x) ||^2 \leq G^2, \quad \forall x\in \mathbb{R}^d, \quad \forall i.$$
>
> This is a bounded gradient assumption, exactly as we claim in Line 133 and summarize in Table 1 of our paper. Note that $G$ is positive!
>
> Perhaps this is where your confusion comes from: you probably think $G$ comes only from the stochastic noise. No! One can't set $G$ to zero, since this would mean that
> $$||  \nabla f_i(x) ||^2 =0, \quad \forall x\in \mathbb{R}^d, \quad \forall i,$$
>
> which would mean that $\nabla f_i(x)=0$ for all $x$ and $i$, which means that each function $f_i$ is linear. This setting is too limited, and is in conflict with other assumptions made in their paper. So, in their paper, $G$ is strictly positive. In the deterministic case they instead have $\sigma_i^2=0$ for all $i$, which means that the $O(1/T^{1/2})$ term disappears from their rate. However, the $O(1/T^{2/3})$ term  is still there!
>
> b) You say that $O(1/T)$ rate is straightforward to obtain in the deterministic case. First, such a rate is not included in their paper. Second, please substantiate your claim by providing a straightforward proof as a trivial change in their approach. Simply claiming this does not make it true. This is unjustified criticism.
>
> c) Yes, Koloskova et al work in the more general decentralized case in that particular paper. But this reduces to our case for a fully connected network. Even then, their rate is $O(1/T^{2/3})$. Please note that the same $O(1/T^{2/3})$ rate is obtained in two other papers we list in Table 1 - these papers do not consider the decentralized case. We mention the decentralized setting of Koloskova et al as their approach there is an improvement on their previous approach, despite the generalization to arbitrary network topology.

---

> ### Author Response · Authors · 2021-08-05
> **Response to Issue 4: experiments - A**
>
> Thanks for reviewing our experiments. However, we disagree they are not convincing. Let us explain. Please ask any questions if anything is unclear.
>
> A) "First, # Gbits/n is used as X-axis for all figures. However, for compression algorithms the convergence speed with respect to iterations or time are also important since when communication is reduced, the compression can become a new overhead. If the paper claims EF21 is practically better, other X-axis should be provided. "
>
> - Note that EF performs the same amount of local computation and server computation as EF21. While this is apparent from the description of the methods, and hence we thought this was obvious, we did not explicitly state this in the paper. We will add a couple sentences to the paper to make this clear - this is easy to do. Since we do not have any computation overhead in EF21 compared to EF, even if one compresses more in both methods (say, by choosing Top-$k$ with a smaller value of $k$), so that local computation becomes more expensive relative to communication, both EF and EF21 are affected in the same way. Therefore, it is not interesting to look into this effect computationally. There is no difference. This is why we did not feel the need to include any experiments of this type. And we still do not feel the need to do so. But we can include them in the supplementary material; this will certainly to hurt the paper, and may shed light on this question to people who moss the comments that we promised to add. So, thanks for your comments.
>
> - Note also that both EF and EF21 have the same per-iteration communication cost. Indeed, in both methods, all nodes first apply $\cal C$ to some vector (a different one in each method!), and then send this compressed vector to the master. So, EF and EF21 do not differ in the cost of communication per iteration. So, comparing this would also not make sense.
>
> - What does make sense to compare is: sensitivity of both methods to the stepsize choice $\gamma$, and we perform experiments in this direction. It is important to do so for several reasons. First, standard EF convergence theory is not only weak relatively to our EF21 theory, but it is also based on strong or even unreasonable assumptions - and we elaborated on this in the paper. A byproduct of the latter is that the EF stepsize depends on quantities which are not known, or even which can't be known before the start of the method, such as a bound on the norm of the gradients the method will see throughout the iterates. So, it is more difficult to choose the EF stepsize according to theory. Since EF21 theory does not rely on any such quantities, it is easier to set according to theory. Our theoretical stepsizes, and the rates they lead to, are clearly better. But what about empirical/tuned stepsizes? We do not know. It is conceivable that perhaps while EF21 theory predicts better stepsizes and a better rate, EF might actually allow even better stepsizes in simulations, albeit not supported by theory (yet). So, we investigated this in the experiments. Our conclusions are: EF21 allows much larger stepsizes than EF without this leading to divergence! Now, since EF and EF1 perform the same amount of computation in each iteration, and communicate the same amount of bits in each iteration, the key difference remaining is: stepsize, and its effect on the number of iterations (=communications) it has. This is precisely what we do in our first set of experiments.
>
> Due to the above reasons, we ignore the amount of computation that is performed by both methods (it is the same), and only focus on the number of bits transferred in order to drive the gradient norm to a small enough value. Since both methods communicate the same number of bits if used with the same compressor $\cal C$, focusing on the amount of bits transferred is exactly what is needed.
>
> Also note that there is a one-to-one mapping between the number of communicated bits and number of iterations. So, the iteration plots, which you suggest we include, would look exactly the same!

---

> ### Author Response · Authors · 2021-08-05
> **Response to Issue 4: experiments - B**
>
> B) "Second, some of the plots for baselines are omitted for no reason, which is serious. Specifically, in the Resnet18 on CIFAR-10 experiment, EF and EF21 seems to perform similar under 128X, however only EF21 is shown under 512X while EF at 512X is not shown."
>
> We explain in lines 543-544 of the paper that "We tune stepsize starting from $10^{-3}$ as a baseline and increase it by a factor of 2. In Figure 13 we compare EF, EF21, EF21+, and SGD with the best tuned stepsizes." This resolves your concern - there is no issue here. For EF21+ we use 512x stepsize because that's the best stepsize for EF21+! So, EF21+ allows for larger stepsizes in practice (in this experiment). Note that in Fig 13, the best stepsize for EF and EF21 is 128, but still EF21 does better. Even if the stepsizes are the same, the methods are different, and EF21 does better.

---

> ### Author Response · Authors · 2021-08-05
> **Response to Issue 4: experiments - C**
>
> C) "Note that validation performance is not informative here, since the design of EF21 does not consider generalization."
>
> Indeed, our theory does not focus on generalization, but instead focuses on optimization. In this sense, and in this sense only, we agree that it "is not informative" to perform testing error comparisons (which is what we believe you mean by "validation"). However, note that almost no method used in practice to perform training, including the methods used by everyone, such as ADAM, are supported by generalization guarantees, and still, generalization/testing experiments are performed. This is obviously because generalization is what we are ultimately interested in in machine learning.
>
> The fact that EF21 is able to achieve comparable generalization to EF should obviously be seen as a plus. We included these experiments even though our paper would be OK without them.
>
> So, we suggest this should not be seen as criticism which in any way could affect our score.

---

> ### Author Response · Authors · 2021-08-05
> **Response to Issue 2: sampling/minibatching**
>
> Issue: "Could you provide a data-parallel setting where we need communication compression but no sampling? I cannot think of any since communication overhead is usually observed in deep learning tasks and sampling is a must rather than an option. Having said that, the application of EF21 is very limited. The authors provide empirical results on deep learning tasks in the appendix, but mini-batching is used there, which is not aligned with the theory or logistic regression task."
>
> - Error feedback is a mechanism which is orthogonal to stochastic approximation at the nodes/devices. In other words, the use of minibatching/subsampling at the nodes is a different/independent/orthogonal/additional algorithm-enhancing "trick", in the same way that momentum, adaptive stepsizes, asynchronous communication (mentioned by another reviewer as something we are missing - and we disagree there as well), the use of second order information, use of sampling with our replacement (random reshuffling), and so on, are tricks which can all be potentially combined with EF21 for a more pronounced aggregate effect. Our key goal in this paper is to study error feedback in depth, and to be able to do so, we do not focus on any of these additional tricks. We are fully convinced that these should be studied in subsequent papers instead, each in depth. This is how we believe good science should be done. Consider, as an example, Nesterov's momentum. Nesterov studied it in isolation, without considering additional tricks such as parallelism, communication compression, adaptive stepsizes and so on. Still, his paper is a seminal piece of work. Additional development came later (and indeed, parallel, communication efficient and adaptive method were developed later by others). Now let's turn back to our paper. Once we decided to focus on error feedback, and unbothered by these extra considerations, which we argue are not central to our key object of study, we created space in the paper that enabled us to explain our algorithmic, methodological and theoretical contributions more properly, intuitively, and completely, for the benefit of the community. We are convinced that the paper is *better* as a result, not weaker.
>
> - We completely agree that additional tricks, such as stochastic approximation (i.e., being able to work with stochastic gradients at the nodes instead of full/exact gradients) is very useful in many applications (but of course we disagree that this is a must in all). However, we argue that stochastic approximation in conjunction with EF21 should best be studied in a follow-up paper if this is to be done in sufficient depth. Having said that, you perhaps missed that in Appendix I we did in fact describe a simple stochastic extension of EF21, and outlined how the analysis can be done (see Lemmas 7 and 8). The approach outlined there can be formalized into a theorem - but we did not do so on purpose. We can easily add a formal theorem as well if you strongly suggest that we do. We did not want to focus the attention of the reader to this too much though, due to the reasons mentioned above. Further, we already obtained much stronger results for stochastic approximation, and are writing a separate paper on this (and some other tricks as well). These additional results go far beyond the simplistic analysis outlined in Section I of our paper since they consider variance reduction as well, and combining this with EF21 is not trivial. Designing, describing and establishing the theory for a successful variance reduction strategies for EF21 takes a lot of space, and for this reason we have decided to write a separate paper. If you want, we can mention some of these results in more detail here - please just ask. At this point we want to reiterate that any extensions of EF21 should be seen as such - that is, extensions not central to the core theory of EF21.
>
> - There are scenarios where distributed training/optimization is done with (as you mentioned) but also *without* stochastic approximation at the nodes. In federated learning, for example, it does happen that most or all devices have very limited amount of data. This precisely why they need to collaborate to build a more powerful model. At the same time, communication is typically very expensive in such settings. In such a regime, a local pass over data can be cheap enough relative to communication cost. This is well known in the federated learning literature.

---

> ### Author Response · Authors · 2021-08-05
> **Response to Issue 2: data heterogeneity**
>
> Issue: "Also the data heterogeneity issue is confusing: In data center training where we are able to shuffle the dataset, we can usually reduce the intra-worker variance. Heterogeneous data usually takes place in the federated learning, where the execution of EF21 shown in the paper clearly does not cover."
>
> - Indeed, data heterogeneity is one of the key issues in distributed learning/optimization in general, and federated learning in particular. See, for example, Section 3.1 entitled "Non-IID Data in Federated Learning" in the survey paper "Advances and Open Problems in Federated Learning": https://arxiv.org/pdf/1912.04977.pdf. There are tens or even hundreds of papers trying to address, in one way or another, data heterogeneity in federated learning.
>
> - We challenge your view that error feedback is unrelated to federated learning. In fact, error feedback is a key trick often used in federated learning. On page 33 of the above survey there is this text: "...Consequently, a wider set of ideas related to error-correction such as [311, 405, 463, 444, 263, 435] are relevant in this setting, many of which could address both (a) and (b)..." Error correction refers to error feedback. See also "Section 4.2.1. Gradient quantization" in the survey "A survey of federated learning for edge computing: Research problems and solutions" which prominently mentions error feedback: https://www.sciencedirect.com/science/article/pii/S266729522100009X
>
> - We agree that in centralized training it is possible to reshuffle data across workers to achieve a more balanced or less heterogeneous data regime. However, this is often not done due to other considerations. For instance, the problem of partitioning data so as to achieve a more homogeneous setup is not easy, and may be computationally expensive. Strategies such as random partitioning work to a certain degree only, and while the result is often a better balanced distribution, a lot of heterogeneity still remains, and this poses issues to algorithms that are not design to be robust to heterogeneous data.
>
> In summary, it is clearly much much better to design methods with data heterogeneity in mind than to rely on methods that work for homogeneous data only, hoping that we will not need to apply them to heterogeneous data, or that some pre-processing will help.

---

> ### Author Response · Authors · 2021-08-05
> **Response to Issue 3**
>
> Issue: "Based on 1 and 2, the main improvement of Markov compressor upon EF is data heterogeneity where $||\nabla f_i||$ can be large when the algorithm converges. However, the only case the Markov compressor can address this seems to be in non-stochastic case. If we use stochastic gradient in the Markov compressor, the error bound does not approach zero either. How EF21 handles this issue remains unclear."
>
> There are serious issues with this thinking which relies in the validity of the comments raised in issues 1 and 2. However, we have explained that the comments are not valid criticism. In short, EF21 *is* combinable with stochastic approximation and we have outlined a simple approach to this in the appendix which the reviewer missed, and heterogeneous data regime *is* key to several important regimes of huge interest in practice (e.g., federated learning). The extra noise coming from stochastic approximation should *not* be handled by error feedback - error feedback is *not* the right mechanism to handle this. Instead, this can be handled by variance reduction techniques such as SVRG, Spider, SARAH, PAGE and so on. EF21 *can* be combined with such techniques, and we have obtained such methods and have proved convergence. In this settings, EF21 works as it should, the error does diminish progressively in a similar way as this happens in the full gradient case we consider in this paper. However, such extensions are left to future work/papers as it is not possible nor desirable to include all in a single paper and still do a good job in explaining the key results properly.

---

> ### Author Response · Authors · 2021-08-12
> **Re clarification: On the problem with improved rates**
>
> Thanks for the response and clarification.
>
> What you say here is **not reasonable**. Of course, the stochastic regime of Koloskova et al includes the deterministic regime as a special case! So, if we want to see what their results says in the deterministic regime, then what we did is exactly mathematically correct! We need to specialize it to that case, and their rate would be $G^{2/3}/T^{2/3}$. This is a mathematically correct statement about the rate of their EF method in the case of deterministic gradients!
>
> Do you indeed claim this is not true? If this is so, I really need the help of the AC and the other reviewers here as we do not know how to explain such a basic derivation beyond what we already did.
>
> This is the standard way any analysis that covers a family of functions works! For instance, look at the SGD analysis of Gower et al (2019): http://proceedings.mlr.press/v97/qian19b.html Their SGD analysis reduces to the standard GD rate in the non-stochastic case. They explicitly say so. This is a strength of their analysis.
>
> We can't compare to results that they do not have. If they had a better result in the deterministic case, we would have compared with that. We claim that an $O(1/T^{2/3})$ rate **is** the best rate, prior to our work, for error compensation in the deterministic regime for smooth nonconvex optimization without PL. We provided a table with known results in our paper.
>
> If you know of any work that proves a better rate in this deterministic regime, then let us know. If you do not know any, it is not possible to raise the criticism you do. It is not valid.
>
> ---
>
> With EF21, we can obtain a $O(1/T)$ rate also in the stochastic case. We already proved it. But this is the subject of a follow up paper (currently an 80p long private research report we are working on). Our method is based on a combination of EF21 and the optimal nonconvex SGD method for finite sum problems called PAGE [Li et al, 2020] for removing the noise introduced by the stochastic gradient. This shows that EF21 is fundamentally superior to EF in theory even in the stochastic case. If the AC allows this, we can even share the method, theorem and proof. But this would need to be confidential.
>
> One can push EF21 to many extensions (including partial participation, for example), but we decided to keep our first paper on EF21 as simple as possible. We view this as a feature of our paper!

---

> ### Author Response · Authors · 2021-08-12
> **Re clarification: On the problem with application**
>
> What you fail to see is that EF21 is the *bare bones* error feedback mechanism, without any additional bells and whistles, which orthogonal tricks unrelated to error feedback.
>
> As presented, EF21 is the simplest possible EF mechanism. As such, it has a fundamental value in the same way Nesterov accelerated gradient descent has a fundamental value without allowing for stochastic gradients, partial participation and so on. So, we believe the readers will **appreciate** he simplicity rather than criticize lack of applications.
>
> having said that, EF21 **is** combinable with both stochastic approximation and partial participation. We have obtained these results and have the proofs. So, EF21 **should not be rejected** just because we decided not to include these results into the paper. Some papers are better when they are simple, and this is exactly what we wanted to achieve.
>
> If the AC allows, we can share our part participation extension of EF21, with the proof of convergence.
>
> The simple version has application in cross silo FL, where there is a small number of nodes and each node is powerful enough to route full gradients. But yes, the applicability of EF21 can be extended substantially by allowing partial participation and stochastic approximation.
>
>
> **Are you really ready to recommend rejection for a breakthrough paper just because we did not include these extensions in the paper?**

---

> ### Author Response · Authors · 2021-08-12
> **Re clarification: On the problem of extension to stochastic case**
>
> The intuition does not break if one applies the PAGE variance reduction mechanism to reduce the variance coming from stochastic gradients. This is because variance reduced gradients get better over time, and their variance diminishes to zero. So, the stochastic gradient estimators will become better and better approximations of the true gradient. Combined with EF21, the combined EF21-PAGE method works and has an $O(1/T)$ rate. Yes, we merely claim this as we have a proof in a separate document that we are preparing for submission as a follow up paper. Perhaps the AC can check this? We are happy to share it.
>
> This is precisely why we separated stochasticity from error feedback. We realized that error feedback is an orthogonal trick. Stochasticity is not what error feedback handles well. For this we need dedicated variance reduction mechanisms. If such mechanisms are not applied, the rate gets worse. But it still reduces to our $O(1/T)$ rate in the deterministic special case (unlike the results of Koloskova at al).
>
> EF21+ theory: the proof is **exactly the same** as for EF. We explained where exactly the difference arises in the proof: it happens in just one step. So, we do not need a formal proof - the would be just repeating everything verbatim.
>
> Re the rudimentary stochastic case: Indeed, we have only outlined the proof in this case and the reviewer is justified in criticizing this if indeed he/she views this result as important. We hesitated to include it in the first place. But we can and will include in detail in the camera ready version of the paper. This is easy for us to do.

---

> ### Author Response · Authors · 2021-08-12
> **Re clarification: increase by 1 point**
>
> Thanks for the increased score. You said:
>
> > "as the issues I raised are serious and closely related to the main contribution of this paper, unless the authors can prove I misunderstood them all."
>
> Yes, we know for a fact you misunderstood these issues. They are not issues. And they are not serious at all.
>
> Let us continue the discussion so that we can convince you.
>
> Perhaps the AC can step in and verify that we indeed have obtained the claimed rates for EF21 combined with stochastic gradients and partial participation? This should remove your concerns as it shows that EF21 is indeed the correct baseline error feedback method.

---

> ### Author Response · Authors · 2021-08-13
> **Re: "it's the $G$ term that has problem because you cannot detach the effect of stochastic gradient from $G$"**
>
> **Your claim is mathematically incorrect.**
>
> **While we believe that our original rebuttal entitled "Response to Issue 1: improved rate" was convincing and should have settled the issue (certainly for any expert!), we now see that the level of confusion you have is deeper than we expected. However, we believe that your latest response gave us a clue about what exactly you might be confused about. Because of this, we will now attempt a more detailed explanation, aimed at clarifying the point we think you are confused about (we know for sure you are wrong, all that we do not know for sure is why you think you are right despite this being a very simple issue).**
>
> Let us begin.
>
> **1. The noise**
>
> First, in Koloskova et al (2020), $\nabla F_i(x,\xi_i)$ is a random vector which in some sense approximates the gradient $\nabla f_i(x)$. In particular, they assume that there exists $\sigma_i^2\geq 0$ such that
>
> $$ \mathbb{E}_{\xi_i \sim {\cal D}_i }  || \nabla F_i(x,\xi_i) - \nabla f_i(x) ||^2 \leq \sigma_i^2, \quad \forall x
> \in \mathbb{R}^d,$$
>
> where ${\cal D}_i$ is the distribution of $\xi_i$. It will be useful to use a bit more detailed notation.  Note that the left hand side depends on the properties of $f_i$ and also on the properties of the noise $\zeta_i$, i.e., on ${\cal D}_i$. So, obviously, the right hand side $\sigma_i^2$ *also* depends on both. However, Koloskova et al suppress this dependence in their notation for simplicity. This is normally not an issue, as virtually all papers in the field do so, and no confusion arises since people *know* that this dependence *is* there, and when this is important, they can invoke it. Let us make this dependence explicit:
>
> $$\sigma_i^2  = \sigma_i^2 (f_i, {\cal D}_i).$$
>
> Now, let ${\cal D}^*_i$ correspond to the distribution for which we have $ \sigma_i^2 (f_i, {\cal D}^*_i) = 0$; that is, ${\cal D}^*_i$ is the distribution of the noise which leads to perfect reconstruction of the gradient. In other words, this means that there is "no noise". (Typically, this is formalized by setting $\xi_i\equiv 0$  with probability 1. However, Koloskova et al do not specify it as this level of formalism is not needed in their analysis. But this is not important for our purposes here. )
>
> Let us repeat that the no-noise scenario corresponds to $$ \sigma_i^2 = \sigma_i^2 (f_i, {\cal D}^*_i) = 0,$$
>
> in which case $\nabla F_i(x,\xi_i) = \nabla f_i(x)$ for all $x$ with probability one. We do not have any disagreement on this point.
>
> **2. The "problematic" assumption**
>
> We agree that the "problematic" assumption in the paper of Koloskova et al is this one:
>
> $$ \mathbb{E}_{\xi_i \sim {\cal D}_i} || \nabla F_i(x,\xi_i) ||^2 \leq G^2 ,  \quad \forall x\in \mathbb{R}^d,$$
>
> Let us first clarify a bit what this notation actually means in their paper since we are guessing that your confusion comes from misunderstanding this. We will use the same approach as we used in the case of the noise assumption above. That is, first note that the left hand side depends on the properties of $f_i$ and also on the properties of the noise $\xi_i$. So, obviously, the right hand side *also* depends on both. That is, $G^2$ is a function of both the function and the noise. The notation used by Koloskova et al does not make this explicit, but this dependence is implicitly clear. Let us make it explicit as we believe your misunderstanding has its root here. That is, we will write
>
> $$G^2=G^2(f_i, {\cal D}_i).$$
>
> Now, you say that
>
> > "...you cannot detach the effect of stochastic gradient from $G$"
>
> Of course you can! This is how it is done: we plug ${\cal D}_i = {\cal D}^*_i$ into $G^2(f_i, {\cal D}_i)$ to obtain
>
> $$G^2=G^2(f_i, {\cal D}^*_i)$$
>
> in the same way we plugged in ${\cal D}_i = {\cal D}^*_i$ into $\sigma_i^2 (f_i, {\cal D}_i)$ to obtain
>
> $$\sigma_i^2 = \sigma_i^2 (f_i, {\cal D}^*_i) = 0.$$
>
> We claim that if one *can* use Koloskova et al analysis to capture the no-noise case, then it *must* be the case that
>
> $$G^2=G^2(f_i, {\cal D}^*_i) \neq 0.$$
>
> - Now, $G^2$ can't possibly be zero, and we already explained this in our original rebuttal. Indeed, since in the no-noise case we have $\nabla F_i(x,\xi_i) \equiv \nabla f_i(x)$, if $G^2$ was zero, this would mean that
>
> $$ || \nabla f_i(x) ||^2 = \mathbb{E}_{\xi_i \sim {\cal D}_i}  || \nabla F_i(x,\xi_i) ||^2  \leq G^2(f_i, {\cal D}^*_i) =G^2 = 0 \quad \forall x\in \mathbb{R}^d.$$
>
> But this is not an interesting case as that would mean that all functions $f_i$ are linear. But then $f$ would be linear, and the only way for it to be bounded below (and hence to have a point with arbitrarily small gradient) is for it to be zero. So, this case would be trivial.
>
> - The other option is that $G^2=+\infty$. But in this case, the analysis in Koloskova et al does not make sense, i.e., it does not apply to this case. In fact, Koloskova et al do *not* argue that it is possible for $G^2(f_i, {\cal D}^*_i)$ to be positive and finite. That is, they do not give examples of problems where the inequality
>
> $$ || \nabla f_i(x) ||^2 \leq G^2, \quad \forall x\in \mathbb{R}^d$$
>
> holds for some $0< G^2 < +\infty$. **This is a key limitation in their analysis.** There are not many smooth functions used in practice whose gradients are bounded. For instance, quadratic functions already do not satisfy this. Strongly convex functions do not satisfy this. Logistic loss does not satisfy this. **Note that our analysis does not have this limitation at all.** However, this point is orthogonal to our discussion here.
>
>
>
>
>
>
> ---
>
> **Summary:**
>
> So in the no-noise case, in the "problematic" assumption we have
>
> $$G^2 = G^2(f_i, {\cal D}^*_i) >0,$$
>
> and in this case it has the following form:
>
> $$ || \nabla f_i(x) ||^2 = \mathbb{E}_{\xi_i \sim {\cal D}_i} || \nabla F_i(x,\xi_i) ||^2 \leq G^2(f_i, {\cal D}^*_i) =G^2 ,  \quad \forall x\in \mathbb{R}^d.$$
>
> But then the rate of their method in the no-noise case indeed is $O(G^{2/3}/T^{2/3})$, as we claim in our paper, and in this discussion with you since the beginning.
>
> Moreover, **all of this is obvious to any expert.** **This is not a complicated discussion; this is a trivial point.**

---

> > ### Author Response · Authors · 2021-08-13
> > **Message for Reviewer EY42**
> >
> > Please respond by adding an official comment to whatever comment you are responding to rather than updating your original review. Our conversation will be less confusing that way.

---

> > ### Comment · Reviewer_EY42 · 2021-08-14
> > **The definition of full gradient**
> >
> > I thank the authors for the explanation.
> >
> > Based on their conversion detail, I think the problem seems to come from the definition of full gradient on worker $i$. Specifically, based on  the Equation (1) of baseline paper (koloskova et al), the loss on worker $i$ is defined as
> >
> > $f_i(x) = \mathbb{E}_{\xi}F_i(x;\xi)$
> >
> > which, upon definition, should be invariant to whether sampling is used in the optimization. However, the author says in their response,
> >
> > "in the no-noise case, $\nabla F_i(x;\xi) = \nabla f_i(x)$"
> >
> > which seems to be diverged from the definition Let's say if the distribution of $\xi$ is discrete uniform, i.e., a finite sum setting, then $\nabla F_i(x;\xi) = \nabla f_i(x)$ seems to imply each worker only possesses one data sample. However, if no noise is introduces, the full gradient should always be the $\nabla \mathbb{E}_{\xi_i}F(x;\xi_i)$ based on the definition.

---

> > > ### Author Response · Authors · 2021-08-14
> > > **Re: The definition of full gradient**
> > >
> > > Dear reviewer,
> > >
> > > First, thanks for responding by adding an official comment to our post; this will indeed make the discussion much more transparent and easy to parse. Second, thanks for continuing to engage with us. We appreciate it as this way we feel we have a chance to clear up these confusions.
> > >
> > > **No, there is no problem here either - you are now exhibiting yet another confusion about some basic concepts in stochastic approximation.**
> > >
> > > Yes, Koloskova et al (2020) define
> > >
> > > $$f_i(x) = \mathbb{E}_{\xi_i} F_i(x, \xi_i). \qquad (1)$$
> > >
> > > Let us illustrate the source of your new confusion on a concrete and very simple example. Starting with $f_i$ given, let $\xi_i$ be Gaussian with mean zero and variance $\sigma_i^2$, and define
> > >
> > > $$ F_i(x,\xi_i) = f_i(x) + \xi_i^\top x.$$
> > >
> > > This is not the particular application they have in mind, of course, but their notion, assumptions and theory apply to this simple case as well. And it will serve us well as a simple scenario to show where your misunderstanding comes from. Clearly,
> > >
> > > $$ \mathbb{E}_{\xi_i} F_i(x, \xi_i) = $$
> > >
> > > $$ = \mathbb{E}_{\xi_i} [ f_i(x) + \xi_i^\top x ]  $$
> > >
> > > $$ = f_i(x)  +  \mathbb{E}_{\xi_i} [ \xi_i^\top x ] $$
> > >
> > > $$ = f_i(x)  + \left( \mathbb{E}_{\xi_i}  \xi_i\right)^\top x  $$
> > >
> > > $$ = f_i(x)  + 0^\top x $$
> > >
> > > $$ = f_i(x). $$
> > >
> > > So, (1) holds. Moreover,
> > >
> > > $$ \mathbb{E}_{\xi_i} || \nabla F_i(x, \xi_i) - \nabla f_i(x) ||^2 = $$
> > >
> > > $$ = \mathbb{E}_{\xi_i} || \nabla f_i(x) + \xi_i - \nabla f_i(x) ||^2 $$
> > >
> > > $$ = \mathbb{E}_{\xi_i} || \xi_i ||^2 $$
> > >
> > > $$ = \sigma_i^2, $$
> > >
> > > and hence the first part of their Assumption 2 holds as well.
> > >
> > > In this simple scenario, the "problematic" assumption (in the eyes of the reviewer) of Koloskova et al (2020), i.e., the second part of their Assumption 2, takes the following form:
> > >
> > > $$  \mathbb{E}_{\xi_i} || \nabla f_i(x) + \xi_i  ||^2  = $$
> > >
> > > $$ = \mathbb{E}_{\xi_i} || \nabla F_i(x, \xi_i)  ||^2 \leq G^2.$$
> > >
> > > Now, the no noise case in this example corresponds to setting $\xi_i\equiv 0$ (one can also relax this a bit and say that $\xi_i\equiv 0$ with probability 1 and this will have no effect on the validity of anything Koloskova et al talk about). In this case, we have $\sigma_i^2=0$, and
> > >
> > > $$ \nabla f_i(x) = \nabla f_i(x) + \xi_i = \nabla F_i(x,\xi_i) \quad \text{with probability} \quad 1, \qquad (2)$$
> > >
> > > as we claimed. The reviewer says that (1) is incorrect, and that the gradient of $f_i$ should instead look like this:
> > >
> > > $$ \nabla f_i(x) = \nabla  \mathbb{E}_{\xi_i} F_i(x, \xi_i) . \qquad (3)$$
> > >
> > > We wish to make two (trivial) points here:
> > >
> > > - Identity (2) is *not* incorrect, as we have shown above. It is correct.
> > > - Identity (3) suggested by the reviewer is *also* correct, obviously so, since it follows by directly differentiating $f_i$ on both sides in (1).
> > >
> > > **The obvious fact that (3) is correct does not in any way imply that (2) is incorrect. Both are correct. What the reviewer wrote in no way challenges our claims.**
> > >
> > > **Above we point to the trivial mistake the reviewer committed in his/her thinking in his/her "the definition of full gradient" post using a very simple illustrating example. We chose this example for pedagogical reasons: we feel we need to get to a very elementary discussion here as the misunderstanding by this reviewer goes very deep.**
> > >
> > > ---
> > >
> > > Further, note that the above example enables us to show, in a **third** way (we already shown this twice, first time in "Response to Issue 1: improved rate" and second time in "Re: it's the $G$ term that has problem because you cannot detach the effect of stochastic gradient from $G$") and on a simple example, that our interpretation of the $G$ terms is correct:
> > >
> > > Indeed, as shown above the problematic assumption of Koloskova et al (2020) in this simple example takes the form
> > >
> > > $$  \mathbb{E}_{\xi_i} || \nabla f_i(x) + \xi_i  ||^2 = $$
> > >
> > > $$ = \mathbb{E}_{\xi_i} || \nabla F_i(x, \xi_i)  ||^2 \leq G^2.$$
> > >
> > > In the no-noise case (i.e., for $\xi_i=0$), it specializes to
> > >
> > > $$  \mathbb{E}_{\xi_i} || \nabla f_i(x) ||^2 \leq G^2. $$
> > >
> > > This is the assumption they make when one specializes their result to the no-noise / deterministic / full-gradient case. One can't have $G=0$, as we explained before, and so one either has $G^2=+\infty$, in which case their theory does not apply at all, or $G^2$ is positive and finite, in which case their theory implies the slow $O(G^{2/3}/T^{2/3})$ rate, as we have claimed since the beginning.

---

> > > > ### Comment · Reviewer_EY42 · 2021-08-14
> > > > **Re: Re: The definition of full gradient**
> > > >
> > > > I thank the authors for clarifying their conversion on the full gradient. Based on the discussion with authors in recent posts, it comes clear to me we are little diverged from my original question, as the fact of $1/T^{2/3}$ to $1/T$ potentially implies two possible things:
> > > >
> > > > (1) The presentation of baseline result is loose in a special case. It's possible that their algorithm is good enough to get $1/T$ in no-noise case and no further design is needed. But it needs reproving a polishing on the demonstration.
> > > >
> > > > (2) The baseline algorithm is not good, and $1/T^{2/3}$ is the best rate it can get in the no-noise case. EF21 gives a better design with Markov compressor and is able to improve the limitation of $1/T^{2/3}$.
> > > >
> > > > I raised concern 1 is because I was convinced (2) is the case. My question has always been: how do you rule out (1) when you are dealing with a result now analyzed in the same setting?
> > > >
> > > > As a side note, I politely ask the authors to avoid calling a reviewer's question "trivial" "basic", and implying the expertise of a reviewer repeatedly in all your response with bold font. These are unnecessary and unrelated to the discussion. The reason why a reviewer actively discusses these questions with you (basically taking his/her time for free), especially with such quick response, is that he/she is willing to resolve the questions with you and to make sure your paper won't be rejected due to unfair factors.

---

> > > > > ### Author Response · Authors · 2021-08-14
> > > > > **(1) vs (2) is a new issue**
> > > > >
> > > > > We are glad that you now agree about our points! We hope this part of our discussion is settled, and we are thankful that you have engaged in the discussion. We made it quite clear that we have found it frustrating as indeed, we do believe these points are trivial. And we used bold to make sure this message gets across. But we are glad we managed to convince you in the end! We will honor your request and will not use bold in this way from now on. We will use bold for other things.
> > > > >
> > > > > Having read your last response, we noticed that
> > > > > **you have now raised yet another issue which is related but fundamentally different from what we have been discussing so far.**
> > > > >
> > > > > Re your point (1): The proof in Koloskova et al fundamentally depends on the gradient bound, even in the no-noise case. We do not see any way to fix it - and we tried. The authors did not present a better rate in that regime. We know that many people tried to get a better rate and failed. This does not mean, of course, that it is *not* possible to prove that the original EF method converges at a $1/T$ rate.  However, we are not aware of *any* result establishing such a $O(1/T)$ rate in *any* paper on error feedback, not just the Koloskova et al paper (which was the theoretical SOTA to the best of our knowledge before our results). So, this is a major open problem in the theory of error feedback. Our solution to this problem is a fundamental redesign of error feedback as a method, and this leads to EF21. And for this redesigned error feedback method, we were able to prove a $O(1/T)$ rate. Moreover, EF and EF21 are identical for deterministic, positively homogeneous and additive compressors - and hence our proof actually *is* a proof that EF converges at the $O(1/T)$ rate; albeit not for all contractive compressors.
> > > > >
> > > > > (2) This is at least partially incorrect due to our Theorem 1 which shows restricted equivalence of EF and EF21. However, the problem whether EF obtains $O(1/T)$ rate also for contractive compressors not covered by Theorem 1 is open. It may be the case that EF can't be better than $O(1/T^{2/3})$ for compressors not covered by Theorem 1, and it can be case that this is possible. No one knows. This is an open problem that needs to be investigated. However, in our mind, this is not a very important open problem anymore since we believe EF21 is a more natural error feedback mechanism than EF, and it also works better in practice, as we have shown.
> > > > >
> > > > >
> > > > > ---
> > > > >
> > > > > In any case, we have constructed the first error feedback mechanism that can achieve the fast $O(1/T)$ rate of gradient descent. And our rate is tight in that if we plug the identity compressor to our theorem, we obtain the rate of gradient descent exactly. We obtain the best rate for error feedback in the PL case as well. **We believe this is of the highest theoretical interest in the area of communication efficient distributed training.** And our method and proof techniques extend to many additional tricks, including partial participation and stochastic approximation (we have these results, but are writing a separate paper for this).
> > > > >
> > > > > ---
> > > > >
> > > > > We would be glad if you could respond to our rebuttal for other issues as well. Thanks!

---

> ### Comment · Reviewer_EY42 · 2021-08-14
> **updated score for the concern 1**
>
> I thank the authors for the clarification and addressing my concern 1 of improved rates. Although we've had many posts on this concern, I believe it is the latest post from authors titled "(1) vs (2) is a new issue" that answers what I asked in the first place. I recommend the authors to include the details of this post into the papers so as to show clearly what the contribution is. Since the authors clarified this, I've updated my score accordingly.
>
> Regarding other issues, I notice the authors mentioned they send some updated results to AC due to confidential issues. Since I cannot see those results, I believe only AC can decide on those.

---

> > ### Author Response · Authors · 2021-08-14
> > **Re: Updated score for the concern 1**
> >
> > Thanks for adjusting your score based on our discussions about one of the points raised.
> >
> > You wrote:
> > > Regarding other issues, I notice the authors mentioned they send some updated results to AC due to confidential issues. Since I cannot see those results, I believe only AC can decide on those.
> >
> > We see several ways how this can be handled:
> >
> > A) We send our manuscript to the AC who will verify that we have the partial participation and EF21-PAGE results we claimed we have.
> >
> > B) We write the proofs here on OpenReview.
> >
> > We prefer (A) for two reasons: i) because our results on partial participation and EF21-PAGE (and several other results) are subject of ongoing follow-up work that we plan to submit to a conference soon, ii) because it is not easy to reproduce the proofs here in OpenReview (our proofs are in LaTeX, using many macros, and MathJax used by OpenReview is much less powerful...)
> >
> > However, we will proceed with (B) if either the AC does not want or can't go the first route, or if you prefer (B).
> >
> > **Please let us know which route do you prefer. We stand behind our claims and are ready to provide the results you requested. Will this settle all your remaining issues?**
> >
> > Authors

---

> > > ### Comment · Reviewer_EY42 · 2021-08-16
> > > **Thanks the authors for providing two options**
> > >
> > > I thank the authors for providing two options. I agree with the authors for them to take (A) to further justify their results with AC.

---

> ### Author Response · Authors · 2021-08-23
> **Stochastic approximation, partial participation and variance reduction**
>
> Dear Reviewer EY42,
>
> You have raised some concerns regarding the applicability of EF21. You mentioned/hinted/noticed that
> - the simple **stochastic approximation** extension we outlined in the appendix is not formalized,
> - we do not consider **partial participation** that is important in federated learning, and that
> - we will likely not get the fast $1/T$ rate in the stochastic case.
>
> We claimed it is easy to formalize the stochastic approach outlined in the appendix, and that EF21 allows for various extensions (which we claimed underlined the importance of this method), including partial participation and variance reduction for stochastic approximation via the PAGE estimator, yielding the fast $O(1/T)$ rate.
>
> **We will now handle all three concerns, and hope this settles your remaining issues.**
>
> **1. Dealing with Stochastic Gradients (formal proof of the simple approach outlined in the appendix)**
>
> Assume, as we do in the appendix (see for justification there), that
> $$
> \mathbb{E}\left[||\mathcal{C}(x+\xi)-x||^{2}\right] \leq(1-\hat{\alpha})||x||^{2}+\hat{\sigma}^{2}, \quad \forall x \in \mathbb{R}^{d} . \qquad (36)
> $$
>
> We now state a formal theorem for Algorithm 5 (EF21 in the stochastic case), and include its proof. We will add this to the appendix.
>
> **Theorem 4.** Let Assumption 1 hold and let there exist $\hat{\alpha} \in (0, 1]$, $\hat{\sigma} > 0$ such that (36) is satisfied. Set the stepsize in Algorithm 5 as
> $$
> 0<\gamma \leq \frac{1}{ L+\widetilde{L} \sqrt{\hat{\beta }/ \hat{\theta}}  }.
> $$
> Fix $T \geq 1$ and let $\hat{x}^{T}$ be chosen from the iterates $x^{0}, x^{1}, \ldots, x^{T-1}$ uniformly at random. Then
> $$
> \mathbb{E}\left[ || \nabla f\left(\hat{x}^{T}\right) ||^{2}\right] \leq \frac{2\left(f\left(x^{0}\right)-f^{\operatorname{inf}}\right)}{\gamma T}+\frac{\mathbb{E}\left[G^{0}\right]}{\hat{\theta } T}+ \frac{\hat{\sigma}^{2}}{ \hat{\theta}}
> $$
> with $\hat{\theta} = \hat{\theta}(s) = 1 - (1-\hat{\alpha})(1+s)$, $\hat{\beta} = \hat{\beta}(s) = (1-\hat{\alpha})\left(1+s^{-1}\right)$, and $0 < s < \frac{1}{1 - \hat{\alpha}}$.
>
> *Proof.* Averaging inequalities given by Lemma 8 over $i \in \\{1,2, \ldots, n \\}$ yields
>
> $$
> \mathbb{E}\left[G^{t+1} \mid W^{t}\right] = (1-\hat{\theta}) G^{t}+\hat{\beta} \frac{1}{n} \sum_{i=1}^{n} || \nabla f_{i}\left(x^{t+1}\right)-\nabla f_{i}\left(x^{t}\right) ||^{2} +\hat{\sigma}^{2} .
> $$
> Using Tower property and $L_i$-smoothness of $f_i(\cdot)$, we proceed with
> $$
> \mathbb{E}\left[G^{t+1}\right]=\mathbb{E}\left[\mathbb{E}\left[G^{t+1} \mid W^{t}\right]\right]  \leq (1-\hat{\theta}) \mathbb{E}\left[ G^{t} \right]+\hat{\beta }\widetilde{L}^2  \mathbb{E}\left[ || x^{t+1}-x^{t} ||^{2} \right] +\hat{\sigma}^{2}, \qquad (A)
> $$
> where $\widetilde{L}^2 = \frac{1}{n} \sum_{i=1}^{n} L_{i}^{2}$.
>
> By inequality (31) (obtained in the proof of Theorem 2), we have
> $$
> \mathbb{E}\left[f\left(x^{t+1}\right)-f^{\mathrm{inf}}\right] \leq \mathbb{E}\left[f\left(x^{t}\right)-f^{\mathrm{inf}}\right]-\frac{\gamma}{2} \mathbb{E}\left[|| \nabla f\left(x^{t}\right) ||^{2}\right]
> -\left(\frac{1}{2 \gamma}-\frac{L}{2}\right) \mathbb{E}\left[ || x^{t+1}-x^{t} ||^{2}\right]+\frac{\gamma}{2} \mathbb{E}\left[G^{t}\right] .
> $$
>
> Let $\delta^{t} \stackrel{\text { def }}{=} \mathbb{E}\left[f\left(x^{t}\right)-f^{\mathrm{inf}}\right], s^{t} \stackrel{\text { def }}{=} \mathbb{E}\left[G^{t}\right]$ and $r^{t} \stackrel{\text { def }}{=} \mathbb{E}\left[\left\\|x^{t+1}-x^{t}\right\\|^{2}\right]$. Then by adding the above inequality with a $\frac{\gamma }{\hat{2 \hat{\theta}}}$ multiple of (A), we obtain
>
>
> $$
> \begin{aligned}
> 	\delta^{t+1}+\frac{\gamma}{2 \hat{\theta}} s^{t+1} & \leq \delta^{t}-\frac{\gamma}{2}\left\\|\nabla f\left(x^{t}\right)\right\\|^{2}-\left(\frac{1}{2 \gamma}-\frac{L}{2}\right) r^{t}+\frac{\gamma}{2} s^{t}+\frac{\gamma}{2 \hat{\theta}}\left(\hat{\beta} \widetilde{L}^{2} r^{t}+(1-\hat{\theta}) s^{t} + \hat{\sigma}^{2}\right) \\\\
> 	&=\delta^{t}+\frac{\gamma}{2 \hat{\theta}} s^{t}-\frac{\gamma}{2}\left\\|\nabla f\left(x^{t}\right)\right\\|^{2}-\left(\frac{1}{2 \gamma}-\frac{L}{2}-\frac{\gamma}{2 \hat{\theta}} \hat{\beta} \widetilde{L}^{2}\right) r^{t} + \frac{\gamma}{2 \hat{\theta}}  \hat{\sigma}^{2} \\\\
> 	& \leq \delta^{t}+\frac{\gamma}{2 \hat{\theta}} s^{t}-\frac{\gamma}{2}\left\\|\nabla f\left(x^{t}\right)\right\\|^{2} + \frac{\gamma}{2 \hat{\theta}}  \hat{\sigma}^{2}.
> \end{aligned}
> $$
>
>
> The last inequality follows from the bound $\gamma^{2} \frac{\hat{\beta} \widetilde{L}^{2}}{\hat{\theta}}+L \gamma \leq 1$, which holds because of Lemma 5 and our assumption on the stepsize. By summing up inequalities for $t=0, \ldots, T-1$, we get
> $$
> 0 \leq \delta^{T}+\frac{\gamma}{2 \hat{\theta}} s^{T} \leq \delta^{0}+\frac{\gamma}{2 \hat{\theta}} s^{0}-\frac{\gamma}{2} \sum_{t=0}^{T-1} \mathbb{E}\left[ || \nabla f\left(x^{t}\right) ||^{2}\right] + \frac{\gamma}{2 \hat{\theta}}  \hat{\sigma}^{2}.
> $$
> Multiplying both sides by $\frac{2}{\gamma T}$, after rearranging we get
> $$
> \sum_{t=0}^{T-1} \frac{1}{T} \mathbb{E}\left[ || \nabla f\left(x^{t}\right) ||^{2}\right] \leq \frac{2 \delta^{0}}{\gamma T}+\frac{s^{0}}{\hat{\theta }T} + \frac{\hat{\sigma}^{2}}{ \hat{\theta}}  .
> $$
> It remains to notice that the left hand side can be interpreted as $\mathbb{E}\left[ || \nabla f\left(\hat{x}^{T}\right) ||^{2}\right]$, where $\hat{x}^{T}$ is chosen from $x^{0}, x^{1}, \ldots, x^{T-1}$ uniformly at random.
>
>
> **2. Partial Participation**
>
> We now show how to handle partial participation with EF21. We include the method and convergence theorems, but omit the proofs.
>
> We will make the following change to Algorithm 2 in the paper: in line 3, instead of "[the master] broadcasts $x^{t+1}$ to all nodes", we have "Master samples a subset $S_t$ of nodes ($|S_t| \leq n$) by choosing each node $i$ with probability $p_1$ independently from others, and broadcasts $x^{t+1}$ to nodes in $S_t$." We call this modification EF21-PP (EF21 with partial participation of clients).
>
> The general nonconvex regime is handled by the following theorem:
>
> **Theorem 5.** Let Assumption 1 hold, and let the stepsize in EF21-PP be set as
> $$
> 0<\gamma \leq\left(L+\frac{4 \widetilde{L}}{p_{1} \alpha}\right)^{-1}.
> $$
> Fix $T \geq 1$ and let $\hat{x}^{T}$ be chosen from the iterates $x^{0}, x^{1}, \ldots, x^{T-1}$ uniformly at random. Then
> $$
> \mathbb{E}\left[ || \nabla f\left(\hat{x}^{T}\right) ||^{2}\right] \leq \frac{2\left(f\left(x^{0}\right)-f^{\mathrm{inf}}\right)}{\gamma T}.
> $$
>
> The PL nonconvex  regime is handled by the following theorem:
>
> **Theorem 6.** Let Assumptions 1 and 2 hold, and let the stepsize in EF21-PP be set as
> $$
> 0<\gamma \leq \min \left\\{\left(L+\frac{4 \sqrt{2} \widetilde{L}}{p_{1} \alpha}\right)^{-1}, \frac{\alpha p_{1}}{4 \mu}\right\\} .
> $$
> Let $\Psi^{t} \stackrel{\text { def }}{=} f\left(x^{t}\right)-f\left(x^{\star}\right)+\frac{2 \gamma}{p_{1} \alpha} G^{t} .$ Then for any $T \geq 0$, we have
> $$
> \mathbb{E}\left[\Psi^{T}\right] \leq(1-\gamma \mu)^{T} \mathbb{E}\left[\Psi^{0}\right].
> $$
>
>
> **3. Fast $O(1/T)$ rate for stochastic approximation via variance reduction: EF21-PAGE (in single node)**
>
> The simple approach to stochastic approximation described in Step 1 above led to a $O(1/T)$ rate up to a certain fixed neighborhood. Here we fix this issue in the setting where each $f_i$ is of a finite sum form. Our approach is to reduce the variance of the local stochastic gradient estimators via the optimal PAGE method (Li et al, ICML 2021).
>
> To define the PAGE estimator, let $v^{t+1}= \nabla f \left(x^{t+1}\right) $  with probability  $p$ and $v^{t+1} = v^{t}+\frac{1}{\tau} \sum_{i \in I_i^t }\left(\nabla f_{i}\left(x^{t+1}\right)-\nabla f_{i}\left(x^{t}\right)\right) $ with probability $1-p$, where $I_i^t$ is a subset of $\\{1, \dots, m\\}$ of size $\tau$ (independent uniform sampling).
>
> We make the following change to Algorithm 1 in the paper: replace the full gradient in line 4 with its PAGE estimator defined above, and call this modification EF21-PAGE. Our complexity result for EF21-PAGE is this:
>
>
> **Theorem 7.** Let Assumption 1 hold, and let the stepsize in EF21-PAGE be set as
> $$
> 0<\gamma \leq \frac{1}{L\left(1+\sqrt{\frac{6 \beta}{\theta}+\frac{2(1-p)}{p \tau}+\frac{4(1-p)}{\tau} \frac{\beta}{\theta}}\right)}.
> $$
> Fix $T \geq 1$ and let $\hat{x}^{T}$ be chosen from the iterates $x^{0}, x^{1}, \ldots, x^{T-1}$ uniformly at random. Then
> $$
> \mathbb{E}\left[|| \nabla f\left(\hat{x}^{T}\right) ||^{2}\right] \leq \frac{2\left(f\left(x^{0}\right)-f^{\mathrm{inf}}\right)}{\gamma T}.
> $$
>
> Notice that unlike the simple approach describe above (Algorithm 5), EF21-PAGE has the fast $O(1/T)$ rate, without any neighborhood, as claimed.
>
> Kind regards,
>
> Authors

---

### Official Review · Reviewer_HmR5 · 2021-07-15

**Rating:** 7
**Confidence:** 4

**Summary:**

This paper introduces a new simple method for error feedback (or error compensation) for enhancing distributed compressed gradient descent-based (CGD) algorithms. The paper is well written, and theoretical and experimental results are convincing.  Some minor clarifications on the methods and related works could improve the understanding of this work.

**Limitations And Societal Impact:**

This is mainly theoretical work. The Societal impact depends on how the proposed method is applied to solve real problems.


**Main Review:**

Strengths:
- The paper is well written, reads easily, and most central ideas are communicated clearly.
- Theoretical and empirical results showcase good performance gains.
- Assumptions underlying the proposed method are simpler compared to competing alternatives.

Weaknesses:
- Some parts of the paper could benefit from further clarification:
1. Why is $\Psi$ used as a metric for Polyak-Lojasiewicz functions?
1. Do Figs. 1, 2, and the experiments in the appendix indicate both uplink (from workers to server) and downlink (from server to workers) communication cost?

-  There is some related work on quantizing gradient differences or innovations for federated learning, referenced below. How is this work different from the referenced ones?

  - K. Mishchenko, E. Gorbunov, M. Takác, and P. Richtárik,“Distributed learning with compressed gradient differ-ences,”arXiv preprint:1901.09269, Jan 2019.

  - S. Magnússon, H. Shokri-Ghadikolaei and N. Li, "On Maintaining Linear Convergence of Distributed Learning and Optimization Under Limited Communication," in IEEE Transactions on Signal Processing, vol. 68, pp. 6101-6116, 2020, doi: 10.1109/TSP.2020.3031073.

  - J. Sun, T. Chen, G. B. Giannakis, Q. Yang and Z. Yang, "Lazily Aggregated Quantized Gradient Innovation for Communication-Efficient Federated Learning," in IEEE Transactions on Pattern Analysis and Machine Intelligence, doi: 10.1109/TPAMI.2020.3033286.


Other comments:
1. In line 24 of the abstract, there is a word missing in “can a large impact”
1. In line 198 “If we new” should be “If we knew”.


**Time Spent Reviewing:**

4

---

> ### Author Response · Authors · 2021-08-04
> **Addressing weakness 1: Lyapunov function $\Psi$**
>
> Dear reviewer,
>
> Thanks for the positive evaluation of our work!
>
> 1. Lyapunov function components:
>
> First, note that the class of PL functions contains the subclass of strongly convex functions, and that it is standard for analyses of gradient type methods applied to strongly convex functions to exhibit either convergence of the iterates, i.e., $||x^t-x^\star||^2\to 0$, or of the function suboptimality, i.e., $f(x^t)-f(x^\star) \to 0$. Our proof naturally contains the latter quantity since at one point we apply the PL inequality, which gives us a bound on function suboptimality. This explains why the Lyapunov functions contains the function suboptimality term.
>
> The inclusion of the second term, $G^t$, is new in our analysis of EF21 (to the best of our knowledge, it was never presented in any prior error feedback work), and plays a key role! It is included because we rely on our novel Markov compressor's ability to progressively diminish the error introduced by compression, and this is what drives/improves the convergence speed. Inclusion of this term in the Lyapunov function thus has two reasons:
> - It acts as a formal proof of the claim that Markov compressor indeed progressively diminishes the error introduced by compression when applied to the iterates of DCGD (recall that EF21 = DCGD with Markov compressor). Indeed, since the Lyapunov function converges to zero, so does $$\mathbb{E}[G^t] = \mathbb{E}\left[ \frac{1}{n} \sum_{i=1}^n || {\cal M}(\nabla f_i(x^t)) - \nabla f_i(x^t) ||^2 \right],$$
> where $\cal M$ is the Markov compressor. So, as a corollary, $\mathbb{E}\left[  || {\cal M}(\nabla f_i(x^t)) - \nabla f_i(x^t) ||^2 \right] \to 0$ for every worker $i$.
> - Our method is not necessarily monotonic in either the function suboptimality (unlike gradient descent, for example), nor in $\mathbb{E}[G^t]$. However, as our main theorem in the PL setting says,  it is monotonic in a linear combination of the two. So, in each iteration, at least one of these quantities must decrease. In particular, when one increases, the other must decrease even more dramatically. So, the theorem captures the actual discrete dynamics of EF21.

---

> ### Author Response · Authors · 2021-08-04
> **Addressing weakness 2: uplink vs downlink compression**
>
> 2. Uplink vs downlink compression
>
> In all theory and all experiments we only consider uplink compression, i.e., from workers to the server. This is standard in the EF literature. There are two reasons for this:
>
> 1) Often, uplink communication is much more expensive than downlink communication, and hence compression is applied where the bottleneck resides. This has been reported in many works (e.g., in the first paper you cite, where experiments were performed on a supercomputer). Of course, this is not universally true for all distributed compute systems, and may not be true in future systems. But it is often true at present, which makes this setup relevant.
>
> 2) It is difficult to prove any good rates for bidirectional compression even if one works with the much-simpler-and-much-more-understood-in-theory class of unbiased compressors $\cal C$ which satisfy the properties: i) $\mathbb{E}[{\cal C}(x)] = x$, and ii) $\mathbb{E}[ || {\cal C}(x) - x ||^2] \leq \omega || x ||^2.$ Indeed, the first papers with good rates in this regime appeared only recently, as this setup is harder to analyze. And in this class of compressors, error feedback is not necessary, nor effective, since there is a better mechanism there: the DIANA mechanism (introduced in the first paper you cite). The DIANA mechanism is not applicable to general contractive compressors.
>
> However, studying the effect of (a well designed) contractive compressor at the server side is an excellent topic for future research which our paper actually for the first time makes feasible. In fact, now that we finally have a tight analysis of an error feedback mechanism which (for the first time!) gives a tight analysis of gradient descent in a special case (when we use the identity compressor, EF21 reduces to GD), many similar questions pertaining to possible extensions to settings that were so far beyond the reach of error-feedback-based methods suddenly become much more tractable! We thus believe that our work should also be judged by its potential for accelerating further development in the field.
>
> For example, extensions to decentralized optimization, asynchronous communication (one reviewer pointed this out), local steps (as in local SGD), partial participation, adaptive stepsizes, and momentum, to list just a few possible directions of research which are out of scope of our work, now become feasible, thanks to the progress we've made in our work. We firmly believe that the potential for further development should be embraced and valued, and not be seen as a weakness of our work. We wanted to focus on the most basic error feedback setup on purpose, which is where our core contributions lie, so as not to dilute them with further bells and whistles. We already work on a few extensions, in fact, and have obtained some promising results. However, these will be the subject of future paper(s).

---

> ### Author Response · Authors · 2021-08-04
> **Addressing weakness 3: relation to Mishchenko et al (2019)**
>
> We will be concise. If any further questions arise, please ask. We'd be happy to respond.
>
> - First paper: K. Mishchenko, E. Gorbunov, M. Takác, and P. Richtárik, “Distributed learning with compressed gradient differences, arXiv preprint:1901.09269, Jan 2019.
>
> This work studies the class of unbiased compressors and introduces a variance reduction scheme called DIANA for reducing the variance introduced by these compressors. Loosely speaking, DIANA is to DCGD with unbiased compressors what SVRG is to SGD. DIANA is not applicable to the general class of contractive compressors (e.g., Top-K). As is well known, it does, however, apply to contractive compressors which arise as rescaled versions of unbiased compressors (e.g., properly rescaled Rand-K). In some sense, what DIANA achieves for the class of unbiased compressors, EF21 achieves for the class of contractive compressors.
>
> There is some similarity between how DIANA and EF21 work algorithmically, but there are also major differences. For example, the theoretical approach and proofs are completely different.
>
> DIANA compresses (using an unbiased compressor; actually, a ternary quantizer in the cited paper) the difference between the current gradient $\nabla f_i(x^t)$ and a vector $h^t$ which is "learned" using a separate procedure. In EF21, we compress (using any contractive compressor) the difference between the current gradient $\nabla f_i(x^t)$ and the compressed version of the previous gradient using the Markov compressor. So, there is no separate learning procedure since the iterates compressed by the Markov compressor play the role of the vectors $h^t$. The EF21 mechanism is more aggressive. A DIANA type proof does not work for EF21 as it crucially depends on both unbiasedness and on independence of the compressors applied on the different nodes. It was not previously known whether a DIANA-resembling method could work for the class of general contractive compressors. Our work resolves this in the affirmative, thus making a link between two different and hitherto disconnected techniques that were found effective in distributed optimization with compression: DIANA in the unbiased compression world, and EF in the contractive compression worlds.
>
> While DIANA was not the motivation for our work, it is conceivable, in hindsight, that it "could have been", and hence it does make perfect sense to comment on the similarities and differences in the paper. We will elaborate in the camera ready version of the paper as there will be more space there. This is actually a very good suggestion for a didactic improvement of the paper, thanks!

---

> ### Author Response · Authors · 2021-08-07
> **Addressing weakness 3: relation to Magnússon et al (2020)**
>
> We now remark on the connection of our work to the paper
>
> S. Magnússon, H. Shokri-Ghadikolaei and N. Li, "On Maintaining Linear Convergence of Distributed Learning and Optimization Under Limited Communication," in IEEE Transactions on Signal Processing, vol. 68, pp. 6101-6116, 2020, doi: 10.1109/TSP.2020.3031073.
>
> ---
>
> First of all, thanks for the reference. While this paper is not closely relevant (since it does not consider arbitrary contractive compressors, nor error feedback), it is certainly broadly relevant (since it deals with communication compression), and we will be happy to mention it in our work in the proper context. Hence, we consider the suggestion to mention connection to these works as a very minor suggestion.
>
> Magnússon et al (2020) propose a generic framework for analyzing distributed optimization methods in the strongly convex setting. They then extend their framework to allow for a more efficient communication. To achieve this, they propose a special type of a deterministic quantization function, and prove a linear rate.
>
> Key differences from our work:
>
> 1)  While the literature on error feedback, and our EF21 work as well, focus on distributed methods with communication compression via the general class of *contractive compression operators*, which includes the widely popular Top-K, or low rank approximation as in Power-SGD or FedNL, Magnússon et al (2020) do *not* work with this class of compressors at all. Instead, they focus on a very concrete adaptive quantization scheme which relies on the projection onto an adaptive grid. Unlike many contractive compressors, such as Top-1, their quantization scheme is *not* able to perform compression beyond O(d) bits for a d-dimensional gradient, and hence offers very limited compression ratio. Indeed, they say that "Theorem 1 states that we can maintain a linear convergence of the algorithms in Definition 1 with communicating only a fixed number of bits, bd, per iteration". So, while their work is relevant in the sense that it belongs to the literature on distributed/decentralized optimization with compressed communication, they only consider a single quantization compressor which is not able to compress to levels beyond O(d), which is very limiting in many applications. In this sense, their work does not belong to the literature on error feedback, which is concerned with the efficient use of general contractive compressors, such as Top-K.
>
> 2) Adaptive quantization schemes were already considered before, for example in the DIANA paper (which considers an adaptive ternary quantization scheme). DIANA is able to converge linearly with their quantization scheme, just like the method in Magnússon et al (2020) is. So, there is much more connection between these two works rather than Magnússon et al (2020) and our EF21 method. In some sense, one might have expected to see some discussion on the similarities and differences between Magnússon et al (2020) and DIANA in the Magnússon et al (2020) paper, since it appeared much later. While it would be interesting to see whether the method in Magnússon et al (2020) offers any theoretical benefits over DIANA (e.g., does it offer a better rate?), these are considerations that are not relevant to our work on EF21.
>
> 3) Magnússon et al (2020) consider the strongly convex setting only. In our work, we focus on the more challenging nonconvex regime.
>
> 4) Magnússon et al (2020) consider the problem of making a *general class* of linearly convergent methods efficient via their special (but very limited in terms of compression ration they can offer, as described above) quantization scheme. Our EF21 method, on the other hand, works in the opposite way. We focus on making a *single method* communication efficient: distributed gradient decent. But we can do so with *any* contractive compressor. So, these works are complementary rather than in competition.
>
> 5) The quantization compressor used Magnússon et al (2020) depends on some hyperparametres that are not known (see their Assumption 2; in one place a bound on the distance to the solution is needed). In contrast, EF21 does not depend on hyperparameters that can't be efficiently estimated.

---

> ### Author Response · Authors · 2021-08-07
> **Addressing weakness 3: relation to Sun et al (2020)**
>
> We now remark on the connection of our work to the paper
>
> J. Sun, T. Chen, G. B. Giannakis, Q. Yang and Z. Yang, "Lazily Aggregated Quantized Gradient Innovation for Communication-Efficient Federated Learning," in IEEE Transactions on Pattern Analysis and Machine Intelligence, doi: 10.1109/TPAMI.2020.3033286.
>
> ---
>
> First of all, thanks for the reference. The relevance of this work to our work is about the same as the relevance of the Magnússon et al (2020) work to our work. That is, while we are happy to cite this work and explain the differences and similarities to our work in the camera ready version our paper, the results therein are complementary rather than in competition to our results.
>
> The key differences between Sun et al (2020) and our work are:
>
> 1)  Sun et al (2020) focus on a very specific quantization scheme, similar to that one proposed by Magnússon et al (2020). Unlike compressors such as Top-K or low-rank approximation, this scheme is unable to compress gradients below $O(d)$. This is very limiting as in large scale learning, $d$ is enormous, and one always needs to use dramatic sparsification as well. In contrast, the literature on error compensation is characterized by its ability to provide guarantees and support for *arbitrary contractive compressors*, giving the user the flexibility to choose (aggressive) compressors capable of compressing $d$-dimensional vectors to $O(1)$ bits if needed. So, *in this sense*, and just like the work of Magnússon et al (2020), the work of Sun et al (2020) does not belong to the literature on error feedback.
>
> 2) Having said that, the way the quantization mechanism is applied in Sun et al (2020) leads to a linear rate under strong convexity, without strong assumptions on the functions (no assumption on data homogeneity or gradient boundedness is needed). Indeed, both Magnússon et al (2020) and Sun et al (2020) apply the "compression of gradient differences" approach first proposed in the Mishchenko et al (2019) work which introduced the DIANA method. In our work on EF21, we also compress the difference of two quantities, one of which is the gradient.
>
> 3) Moreover, Sun et al (2020) instead mainly focus on a complementary idea which we do not explore in our work: that of lazily aggregating gradients. That is, they sometimes skip communication altogether. We believe that skipping communication can actually be modeled by the design of an appropriate contractive compressor, and in that sense, our work on EF21 is more powerful/general. Indeed, if one defines a compressor $\cal C$ for the sequence of gradients $\{g^0, g^1, \dots\}$ as follows
>
> $${\cal C}(g^{k+1}) =  {\cal C}(g^{k})  \text{ if } || g^{k+1} - {\cal C}(g^{k}) ||^2 \leq (1-\alpha) || g^{k+1} ||^2, $$
> $${\cal C}(g^{k+1}) =  g^{k+1}  \text{otherwise},$$
>
> then, by definition, we get the contractive property
>
> $$ || {\cal C}(g^{k+1}) - g^{k+1} ||^2 \leq  (1-\alpha) || g^{k+1} ||^2.$$
>
> We can now use this compressor to define a Markov compressor, as explained in our paper, and we obtain a method that performs lazy aggregation, complete with its theoretical analysis in the smooth nonconvex and PL cases.
>
> So, while this was not our intention, EF21 is able to model new types of methods supporting lazy aggregation. In fact, we are already working on a follow-up paper in this direction.  The fact that EF21 can lead to such extensions should be seen positively. We believe EF21 will indeed lead to many follow up works in various directions.

---

### Official Review · Reviewer_emqQ · 2021-07-15

**Rating:** 7
**Confidence:** 3

**Summary:**

This paper proposes a new EF mechanism to solve nonconvex distributed optimization problem. The new proposed algorithm has the following advantages：1）Theoretical analysis relies on standard assumptions only as compared to others assumptions used in previous works, and leads to better an more meaningful rates. In specific, the convergence rate is improved from O(1/T^{2/3}) to O(1/T). 2) This paper provides the first linear convergence result for an error feedback method not relying on unbiased compressors. The successful results relies on the proposed recursive Markov compressor. Experimental results are provided to verify their theoretical observations.

**Limitations And Societal Impact:**

Yes

**Main Review:**

This paper is well-written, and the idea is well-presented. As mentioned in this paper, the core idea of the paper is to use the propose recursive Markov compressor. In the theoretically sides, the recursive Markov compressor actually reduces the gradient approximation error, and thus provides better theoretically  results. This idea may share the same spirit of the stochastic variance reduction method in classical nonconvex optimization that reducing the approximation error by  using the previous gradient.  Although I do not go through the proof in details, I believe the proposed method works in theory. Correct me if I am wrong.

The drawback of the proposed method is about  practical usage. First, the nonconvex distributed optimization setting used in this paper is simple Synchronous SGD with distributed data.  Synchronous setting is time-consuming, and not that popular in practice.
The more interesting setting should give to the asynchronous setting.  It will be interesting to extend this idea to asynchronous setting.

Overall, I think the authors in this paper propose a interesting idea by using recursive  trick to solve the  nonconvex distributed optimization problem. In specific, the convergence rate is improved from O(1/T^{2/3}) to O(1/T) ). Moreover, This paper provides the first linear convergence result for an error feedback method not relying on unbiased compressors.  Therefore, I tend to accept this paper at this point for its theoretical contribution.

**Time Spent Reviewing:**

3hours

---

> ### Author Response · Authors · 2021-08-04
> **What do you mean by "Correct me if I am wrong"?**
>
> Thanks for the positive evaluation of our paper!
>
> Please can we ask what do you mean when you say "Although I do not go through the proof in details, I believe the proposed method works in theory. Correct me if I am wrong."? We do not understand what you are trying to say as insufficient detail is given for us to be able to do so.

---

> ### Author Response · Authors · 2021-08-04
> **Drawback: focus on synchronous vs asynchronous setting**
>
> We respectfully disagree.
>
> Not relying in asynchronous communication is as much a drawback of our method as it is a drawback of distributed version of Nesterov accelerated gradient descent, ADAM, and Federated Averaging. The mentioned methods are completely fine, and were transformational to the field of optimization even without any theoretical support of asynchronous communication. In other words, this is by no means a drawback of our method.
>
> Please note that we resolve a major problem in the theory of error feedback, unsolved for 7 years since the development of error feedback in 2014. Error feedback is a synchronous method. So, this can't be an issue with our method.
>
> Perhaps what you want to say is that many distributed methods can benefit from asynchronous communication. We agree! But many methods can also benefit from momentum, adaptive stepsizes, second order information, local steps, stochastic approximation and so on. There are dozens of tricks, more or less understood in isolation, and more rarely understood in limited combinations, which are orthogonal in the sense that they are mutually combinable for an overall better practical performance.
>
> However, this does not invalidate research on methods which study only some techniques (such as error feedback) in isolation from others. Quite the opposite is true: we need to study techniques in isolation first so as to go deep, to really understand what is going on, before going broad, before working on extensions, combinations, applications, software and so on. That is science at its best, one may say. And that is precisely what we do with error feedback. We do not focus on extra orthogonal tricks in this work, such as stochastic approximation (even though we know how to do this - and outline one approach in the appendix), momentum (we do not know how to do it; this seems very challenging - we tried several approaches in fact, but failed), or asynchronous communication (we welcome the community to study this combination; it's a good suggestion!).
>
> Instead, we would want to suggest that the fact that our method can be combined with other tricks is a plus (and indeed, after our work, this is much easier to do than before), as this points to a potential for impact on the field.

---

> ### Author Response · Authors · 2021-08-04
> **Thanks!**
>
> Thanks again for the positive comments and evaluation! We hope you will be able to increase your score given our response to your concern and the concerns of other reviewers.

---

### Author Response · Authors · 2021-08-10
**To All Reviewers**

Dear reviewers,

Thanks a lot for all your reviews and the work you put into reading and evaluating our paper!

Due to reasons which we believe are apparent to experts in the area of error feedback (e.g., EF21 is the first error feedback mechanism for general contractive compressors which has the same rate as gradient descent - $O(1/T)$ and $O(e^{-T})$ for smooth nonconvex and smooth PL problems, respectively; and moreover achieves this using the weakest assumptions in the literature, which are at the same time the first standard and reasonable assumptions used in theory of distributed error feedback), we believe that our work is arguably the most significant theoretical contribution to the literature on error feedback since it was first proposed 7 years ago in 2014.

In the light of this, and since error feedback is widely studied and widely used in the practice of communication efficient distributed training, we believe this is a major theoretical breakthrough, and we are of the opinion that a score in the range 8-10 would be fully appropriate. We do not think a score of 7 or less does justice to our work.

We have submitted our work to a workshop associated with a leading ML/AI conference (this is not against NeurIPS rules as the workshop does not produce proceedings), and our paper received the scores top 15%, top 15%, top 15%, and top 40% out of nearly 100 submitted papers. In our experience, reviewers in tightly focused workshops are selected more carefully, and are more likely to be experts in the area. We believe these scores were appropriate.

We truly hope it is not too improper to voice our view on what we believe is a more appropriate score for our own paper, but in the light of the discrepancy between what we believe our paper brings, and the preliminary scores we have received, we wanted to explain how we feel about our work.

We will soon respond to all detailed comments and issues raised.

Authors

---

### Decision · Program_Chairs · 2021-09-27

**Decision:**

Accept (Oral)

**Comment:**

The paper's strength and appeal is in proposing a simple modification to the Error Feedback mechanism that has both theoretical and empirical benefits.  The idea and mechanism proposed in the paper could have significant implications in important problems include distributed optimization and in optimization under various constraints (communication restricted, memory restricted, privacy restricted, etc).  The proposed EF21 mechanism is also nicely and clearly motivated and presented.  For these reason, I think the paper can make for an interesting oral presentation.

The ACs and myself were disappointed by the the authors choosing to restrict the scope of the paper to presenting the core idea only, without much applications and extensions.  It is much more useful to the community and makes for a much stronger paper to include these in the paper, so as to justify importance of the core idea.  While the paper is certainly "acceptable" without them, in order to be a milestone paper and a foundational reference, these further applications and extensions are appropriate.  This is certainly much preferred to splitting the contribution into multiple papers each with 1 MPU (minimum publishable unit) worth of content.  In a sense the recommendation for an oral presentation is on credit, and based on the expectation that this will urge the authors to make the paper more comprehensive and complete.

I should also say that I found the comment regarding reviewer scores at a workshop entirely inappropriate.  While I appreciate the author's frustration with scores they disagree with, the author response is meant to point out factual issues with the reviews and address reviewer questions, and the authors also have the option of commenting on review quality.  But the paper should speak for it's own merit, and including recommendation letters for the paper, let alone vague and anonymous scores without justification, is not part of the reviewing process.  Furthermore, the criteria in a broad conference such as NeurIPS and in a technical workshop are different.  In particular, for a broad conference, it would be appropriate to include a better discussion and examples of applications and extensions so as to help show the significance of the results to the reviewers, as representatives of the general audience.